# Contrastive Learning with Data Misalignment: Feature Purity, Training Dynamics and Theoretical Generalization Guarantees

**Jiawei Sun**
Rensselaer Polytechnic Institute
sunj11@rpi.edu

**Shuai Zhang**
New Jersey Institute of Technology
sz457@njit.edu

**Hongkang Li**
University of Pennsylvania
lihk@seas.upenn.edu

**Meng Wang**
Rensselaer Polytechnic Institute
wangm7@rpi.edu

## Abstract

Contrastive learning is a powerful framework for learning discriminative representations from image-text pairs. Despite its success, its theoretical foundations, especially when the image-text pair exhibits misalignment, remain underexplored. This paper provides the first theoretical analysis of contrastive learning under data misalignment, proving how the ground-truth modality-paired features are amplified while spurious features are suppressed through the training dynamics analysis. Specifically, we study two nonlinear encoders trained jointly with a contrastive loss and demonstrate that noisy (or misaligned) data pairs result in mixed representations and degrade the model's generalization ability. In contrast, recaptioning and filtering improve the data alignment, which in turn purifies the features learned by neurons and subsequently enhances generalization. Our analysis identifies feature purity as a key factor in the success of contrastive learning and offers insights into how data quality and training procedures impact representation learning and downstream generalization. Theoretical insights are supported by experiments on standard benchmarks.

## 1 Introduction

Vision-language models (VLMs) have achieved strong performance across diverse multimodal tasks such as vision-language understanding and generation. State-of-the-art methods like CLIP [37] and SimVLM [50] use contrastive learning to train dual encoders on large-scale image-text pairs scraped from the web, aligning embeddings by pulling paired samples closer in a shared space. These models excel in zero-shot scenarios, requiring no task-specific fine-tuning.

However, web-sourced captions are often noisy or misaligned, containing irrelevant or spurious details that hinder cross-modal alignment and reduce representation quality. For example, [34] cites an image of a blue Mercedes-Benz in a parking lot paired with the caption: "2003 Mercedes-Benz C240 sedan, Leather, MUST BE SEEN – $6199." The price information in this caption is only superficially correlated with the image and does not contribute meaningfully to understanding the image context. To mitigate this issue, many works [13, 34, 46, 3, 38, 16, 45] adopt text generation methods during VLM training to produce high-quality synthetic captions more faithful to the corresponding images. Models like LaCLIP [13] and BLIP [25] show that such recaptioning improves both the quality and diversity of training data, leading to significantly better performance. Further, [34] demonstrates that the cosine similarities between BLIP2 generated captions [24] and their paired images is higher than

39th Conference on Neural Information Processing Systems (NeurIPS 2025).

Table 1: Comparison with existing theoretical works on contrastive learning.

| Work | Train Dyn. | Nonlinear | Zero-shot Gen. | Recaption | Multi-modal | Joint Encoder |
|---|---|---|---|---|---|---|
| (Wen & Li, 2021) [51] | ✓ | ✓ | | | | |
| (Nakada et al., 2023) [33] | ✓ | | | | ✓ | ✓ |
| (Chen et al., 2024) [10] | ✓ | | ✓ | | ✓ | |
| (Lee et al., 2021) [21] | | ✓ | | | ✓ | |
| (Zhang et al., 2023a) [60] | | | | | ✓ | |
| (Pareek et al., 2025) [35] | ✓ | | | ✓ | ✓ | ✓ |
| **This paper** | ✓ | ✓ | ✓ | ✓ | ✓ | ✓ |

that of raw captions. [22] analyzes conformity on MSCOCO and finds that it correlates with how common or rare an image–caption embedding is, reflecting its degree of alignment within the dataset.

Despite the impressive success of VLMs and the practical advancements driven by recaptioned texts, their theoretical foundations remain relatively underdeveloped. Several critical questions remain mostly open:

> How do contrastively pre-trained VLMs align modalities, extract feature representations, and achieve zero-shot capabilities? How does text recaptioning on noisy image-text pairs provably enhance generalization performance?

Notably, even the theory of vanilla multimodal contrastive learning is still incomplete. For instance, [15, 60] extend spectral contrastive loss to multimodal settings, showing that the objective can be related to matrix factorization. [35] provides a theoretical characterization of when data filtering improves multimodal contrastive learning, offering a complementary, data-centric perspective to objective-level analyses. Also, [17, 21, 59] show that, under certain conditions, multimodal models outperform unimodal ones with better representations. However, these works assume an optimal solution to the non-convex problem without analyzing the training dynamics that lead to strong generalization. The zero-shot ability of VLMs also lacks full theoretical study. To the best of our knowledge, only [10] analyzes CLIP's zero-shot performance, showing it learns shared features while ignoring modality-specific ones. Yet, their setup does not consider real-world issues like misalignment between image and text. Beyond standard contrastive learning, [33] proposes a modified loss using unpaired data to detect ground-truth pairs and improve results, but only for linear models. So far, no work has theoretically studied the effect of text recaptioning on VLMs.

**Contributions:** To the best of our knowledge, this is the first theoretical work explaining why text recaptioning improves zero-shot generalization in VLMs, especially under image-text misalignment, where text may include spurious or missing features. We analyze the training dynamics of stochastic gradient descent (SGD) in multimodal contrastive learning and derive the generalization behavior of the learned model. Our analysis uses a one-hidden-layer ReLU network, which remains the state-of-the-art model in theoretical studies of contrastive [51] and supervised learning [2, 61]. All findings are validated empirically on practical VLMs like CLIP. A comparison to prior theory works is shown in Table 1. Key contributions include:

**1. Theoretical training dynamics and generalization analysis of contrastive learning in nonlinear VLMs.** We provide a theoretical analysis of jointly training two nonlinear encoders with contrastive loss. Prior works on training dynamics in contrastive learning [51, 10, 33] either analyze a single encoder or are restricted to linear neural networks. In contrast, our analysis captures the joint learning behavior of both nonlinear encoders with ReLU activation functions.

**2. Theoretical characterization of the impact of misaligned image-text pairs on pre-training performance.** We analyze a data model with modality misalignment, where some texts may contain features spuriously correlated with the image and others may omit relevant features. We show that spurious and missing features cause neurons to entangle true and irrelevant representations, which hinders the ability of the vision-language model to disentangle semantic components, ultimately degrading generalization performance.

**3. Theoretical justification of enhanced out-of-domain generalization through pre-training with text recaptioning.** We first analyze the training dynamics of the text generation process and formally prove that the resulting text after recaptioning has reduced spurious correlation and enhanced semantic relevance with the corresponding images. When these filtered texts are used for contrastive pre-training, the resulting model exhibits improved feature purity and succeeds in out-of-domain zero-shot classification, whereas the model trained on raw data provably fails.

## 1.1  Related Works

**Vision-Language Models:** VLMs [58, 48, 37, 19, 28, 26] are trained via contrastive learning on large web-sourced image-text pairs. Following CLIP, later models [30, 1, 56] aim to boost zero-shot performance. Data quality has become a key bottleneck, leading to recent filtering efforts [13, 24, 49, 20, 27]. For example, LaCLIP [13] uses LLM-generated caption rewrites as augmentation, and BLIP [25] leverages synthetic captions to drop noisy pairs, enhancing feature quality and robustness.

**Theoretical Exploration on Contrastive Learning.** Recent studies explore why contrastive learning yields effective representations. [47] identifies alignment and uniformity as key properties of contrastive loss. [15] shows that solving auxiliary prediction tasks improves contrastive representations. [44] highlights the role of inductive biases in shaping learning dynamics. [29] proves that multimodal contrastive learning can recover shared latent factors under a generative model.

**Generalization analyses of Neural Networks (NNs).** Various approaches have been developed to analyze the generalization of feedforward NNs. The neural tangent kennel (NTK) approach shows that overparameterized networks can be approximated by kernel methods in the limiting case [18, 2]. The model estimation approach assumes the existence of a ground-truth one-hidden-layer model with desirable generalization and estimates the model parameters using the training data [63]. The feature learning approach analyzes how a shallow NN learns important features during training and thus achieves desirable generalization [31, 23, 43].

## 2   Problem Formulation and Algorithm

VLMs leverage large-scale web-based datasets containing paired visual and textual data to pre-train two separate encoders: an image encoder $f$ and a text encoder $h$, parameterized by weights $\mathbf{W}$ and $\mathbf{V}$, respectively. Contrastive learning serves as the core framework, ensuring the learned embeddings of matching pairs are closer while separating mismatched pairs.

Specifically, let $S$ be the indices of the image-text pairs, e.g., $(x_p, y_p)$ with $p \in S$. $(x_p, y_p)$ is referred to as a positive pair, while $(x_p, y_n)$ with $p \neq n$ is referred to as a negative pair. We minimize the following spectral loss function:

$$L(f,h) = \sum_{p \in S} \left[ -\langle f(x_p), h(y_p) \rangle + \sum_{n \in S \setminus \{p\}} \frac{(\langle f(x_n), h(y_p) \rangle)^2}{2\tau} + \sum_{n \in S \setminus \{p\}} \frac{(\langle f(x_p), h(y_n) \rangle)^2}{2\tau} \right] \tag{1}$$

where the hyper-parameter $\tau > 0$ is referred as the temperature. The spectral contrastive loss $L$ in (1) has been extensively utilized in recent theoretical works [15, 42, 60]. Although it differs from the commonly used SimCLR [9] in practice, the spectral contrastive loss closely resembles SimCLR numerically, as shown in [15].

### 2.1   Training Framework

Let $S = S_h \cup S_w$ include human-annotated high-quality image-text pairs with indices in $S_h$ and noisy web low-quality dataset with indices in $S_w$. Due to the inherently noisy nature of web data, the learned embeddings from (1) may be suboptimal. To mitigate this, many practical training methods [25, 13] incorporate recaptioned text to improve the quality and diversity of image-text pairs. While specific implementations vary, most frameworks follow a similar four-stage approach:

(S1) **Image-text contrastive pre-training (ITCP) on raw data:** The image encoder $f$ and text encoder $h$ are trained using the image-text pairs $\{(x_p, y_p)\}_{p \in S}$ by minimizing the contrastive loss as in (1). Let $\overline{\mathbf{W}}$ and $\overline{\mathbf{V}}$ denote the learned weights in $f$ and $h$. We then estimate the image and text

embeddings of $(x_p, y_p)$ by $z'_{x_p} = f_{\overline{\mathbf{W}}}(x_p)$ and $z'_{y_p} = h_{\overline{\mathbf{V}}}(y_p)$. Due to the low-quality data in $S_w$ when training the encoders, these estimations might not be accurate.

(S2) **Generating text captions:** The high-quality data pairs in $S_h$ are used to finetune an image-grounded text decoder $G$, which maps an image $x_p$ to text through $G(x_p)$. Then, the learned $G$ is applied to every image $x_p$ in $S_w$ to generate a synthetic caption $\hat{y}_p = G(x_p)$. Next, the estimated text embedding of $\hat{y}_p$ is computed as $\hat{z}_{y_p} = h_{\overline{\mathbf{V}}}(\hat{y}_p) = h_{\overline{\mathbf{V}}}(G(x_p))$, where $\overline{\mathbf{V}}$ represents the weights of $h$ learned from Stage (S1).

(S3) **Filtering:** For every $(x_p, y_p)$ in $S_w$, we compute the cosine similarity between the image embedding $z'_{x_p}$ and the text embeddings of the original caption $z'_{y_p}$ and the synthetic caption $\hat{z}_{y_p}$, respectively. If the pair $(z'_{x_p}, \hat{z}_{y_p})$ has higher similarity to each other than the pair $(z'_{x_p}, z'_{y_p})$, $(x_p, y_p)$ is replaced with $(x_p, \hat{y}_p)$. Let $\tilde{S}_w$ denote the index set of the resulting data pairs. By filtering noisy captions in $S_w$ with synthetic captions that better align with image embeddings, $\tilde{S}_w$ becomes a cleaner dataset.

(S4) **ITCP on filtered data:** The image encoder $f$ and text encoder $h$ are trained by minimizing the contrastive loss in (1), repeating the procedure from Stage (S1) with the only difference being that the original dataset $S$ is replaced by $\tilde{S} = S_h \cup \tilde{S}_w$. The resulting loss is denoted by $\tilde{L}(f, h)$. Let $\widetilde{\mathbf{W}}$ and $\widetilde{\mathbf{V}}$ denote the resulting learned weights. $f_{\widetilde{\mathbf{W}}}$ and $g_{\widetilde{\mathbf{V}}}$ can produce improved embeddings compared with $f_{\overline{\mathbf{W}}}$ and $g_{\overline{\mathbf{V}}}$.

We employ stochastic gradient descent (SGD) with step size $\eta$ and batch size $B$, following standard practice. Despite the non-convexity of (1), we present a detailed analysis of the resulting training dynamics and establish convergence guarantees in Section 4. This stands in contrast to existing works [39, 53, 10] that assume the attainability of a global optimum.

## 2.2 Downstream Tasks

As a demonstration of the performance of the learned model $(f_{\widetilde{\mathbf{W}}}, g_{\widetilde{\mathbf{V}}})$, we consider a downstream image classification task in a zero-shot setting. Unlike the regression and binary classification tasks to evaluate the uni-modal contrastive learning in [51], we consider a $K$-classification problem for any constant $K \geq 2$. Each class label is associated with a given text prompt $y_k$, where $k \in [K]$. For any image $x$ with its ground-truth label $l_x \in [K]$, the zero-shot predicted label by the pre-trained models $(f_{\widetilde{\mathbf{W}}}, g_{\widetilde{\mathbf{V}}})$ is computed as $\arg\max_{k \in [K]} \langle f_{\widetilde{\mathbf{W}}}(x), g_{\widetilde{\mathbf{V}}}(y_k) \rangle$. This approach follows the typical setting of zero-shot image classification using VLMs [10, 19, 25]. The prediction is considered accurate if and only if $\arg\max_{k \in [K]} \langle f_{\widetilde{\mathbf{W}}}(x), g_{\widetilde{\mathbf{V}}}(y_k) \rangle = l_x$.

# 3 Technical Assumptions and Setups

We introduce a set of assumptions that are either derived conceptually from the real data distribution or follow existing approaches in contrastive learning theory.

## 3.1 Backbone of the Encoders

We use a two-layer neural network with ReLU activation functions as the image and text encoders, respectively. Formally, we have

**Definition 3.1.** *The image encoder $f_{\mathbf{W}} : \mathbb{R}^{d_1} \to \mathbb{R}^m$ and text encoder $h_{\mathbf{V}} : \mathbb{R}^{d_1} \to \mathbb{R}^m$ is*

$$f(x) = (f_1(x), \ldots, f_m(x))^\top \in \mathbb{R}^m, \quad with \quad f_i(x) = \sigma\left(\langle w_i, x \rangle - b_i\right) - \sigma\left(-\langle w_i, x \rangle - b_i\right), \quad (2)$$

$$h(y) = (h_1(y), \ldots, h_m(y))^\top \in \mathbb{R}^m, \quad with \quad h_i(y) = \sigma\left(\langle v_i, y \rangle - b_i\right) - \sigma\left(-\langle v_i, y \rangle - b_i\right), \quad (3)$$

*where $\sigma$ is ReLU function, and $\mathbf{W} = [w_1, w_2, \ldots, w_m]^\top$, $\mathbf{V} = [v_1, v_2, \ldots, v_m]^\top \in \mathbb{R}^{m \times d_1}$.*

Because deep neural networks are highly nonlinear, analyzing the training dynamics and resulting generalization performance of learned models remains challenging. As a result, existing theoretical studies are largely limited to one-hidden-layer neural networks [2, 61, 51, 33], where the learning problem is already nonconvex. In this paper, we extend this line of research to a more complex setting, where two such encoders are jointly trained for image and text modalities.

## 3.2 Data Model for ITCP

Our data model in Assumption 3.1 builds on the sparse coding framework, which has been widely used in both uni-modal contrastive learning for images [2, 51] and multi-modal image-text contrastive learning [10]. This sparse coding model has been employed in theoretical analyses [4, 7, 14] because it effectively models the practical NLP [5, 6, 36, 11] and image data [55, 52, 54].

**Assumption 3.1** (Sparse coding model for image-text pairs). *Each image-text pair $(x_p, y_p)$, $p \in S$, is generated i.i.d. from the following sparse coding form:*

$$x_p = \mathbf{M}z_{x_p} + \xi_{x_p}, \quad y_p = \mathbf{H}z_{y_p} + \xi_{y_p}, \tag{4}$$

*where $x_p, y_p \in \mathbb{R}^{d_1}$, $z_{x_p}, z_{y_p} \in \mathbb{R}^d$, and $d_1 = \text{poly}(d)$. We assume:*

*(a) Image dictionary: $\mathbf{M} = [\mathbf{M}_1, \ldots, \mathbf{M}_d] \in \mathbb{R}^{d_1 \times d}$ is column-orthonormal.*

*(b) Text dictionary: $\mathbf{H} = [\mathbf{H}_1, \ldots, \mathbf{H}_d] \in \mathbb{R}^{d_1 \times d}$ is column-orthonormal.*

*(c) Additive noise: $\xi_{x_p}, \xi_{y_p} \sim \mathcal{N}(0, \sigma_\xi^2 \mathbf{I}_{d_1})$ with $\omega(1/d_1) \leq \sigma_\xi^2 \leq O\big(\sqrt{\log d / d^{1+c_0}}\big)$.*

*(d) Sparse latent vector: $z_{x_p} = (z_{x_p}^1, \ldots, z_{x_p}^d)$ with $z_{x_p}^j \in \{0, \pm 1\}$, where $|z_{x_p}^j| \sim \text{Bernoulli}(C_z/d)$.*

Notably, we operate in a regime where the noise magnitude can dominate the signal: since $\omega(1/d_1) < \sigma_\xi^2 \leq O\big(\sqrt{\log d / d^{1+c_0}}\big)$[1], we have $\|\xi\|_2^2 \gg \Theta(1) \gg \|\mathbf{M}z\|_2$, indicating that the overall noise energy significantly exceeds that of the signal. Nevertheless, we will show that contrastive learning remains effective even under such high-noise conditions, due to the encoders' ability to extract denoised and purified features, as characterized in Theorem 4.4. An intuitive explanation for why feature recovery is still possible lies in the different alignment properties of the signal and noise: for any active feature $z_j \neq 0$, the signal aligns well with its corresponding basis: $|\langle \mathbf{M}z, \mathbf{M}_j \rangle| = \Theta(1)$, while the noise contribution remains small, $|\langle \xi, \mathbf{M}_j \rangle| \leq O(1/\sqrt{d})$.

We introduce Assumptions 3.2 and 3.3 to capture the characteristics of the dataset $S = S_h \cup S_w$. Notably, the number of high-quality pairs in $S_h$ may be significantly fewer than that of low-quality pairs in $S_w$, with $|S_h| = \Theta(d^2)$ and $|S_w| = \text{poly}(d) \gg \omega(d^2)$.

**Assumption 3.2** (High-quality image-text pairs). *Every high-quality image-text pair $(x_p, y_p)$ with $p \in S_h$ satisfies $z_{x_p} = z_{y_p}$, i.e., the image and text have the same latent vector.*

Compared to high-quality pairs in $S_h$, low-quality pairs in $S_w$ show modality misalignment due to spurious image-text correlations and missing descriptions of key visual features.

**Assumption 3.3** (Low-quality misaligned image-text pairs). *There exists a constant $C_s \in (\omega(1/\log d), 1/2)$ such that for every low-quality pair $(x_p, y_p)$ in $S_w$ and every image feature $\mathbf{M}_j$ ($j \in [d]$) in $x_p$, we have*

$$\Pr\left(z_{y_p}^{j'} = z_{x_p}^j \mid |z_{x_p}^j| = 1\right) = C_s, \quad \Pr\left(z_{y_p}^j = 0 \mid |z_{x_p}^j| = 1\right) = C_s, \tag{5}$$

*where the first term in (5) is the probability that a text feature $\mathbf{H}_{j'}$ ($j' \neq j$) is spuriously correlated to the image feature $\mathbf{M}_j$, and the second term is the probability that $\mathbf{H}_j$ is missing in the text while the image feature $\mathbf{M}_j$ exists.*

Consider the blue Mercedes-Benz example from [34]. Here, $\mathbf{M}_j$ denotes the car's visual feature, while $\mathbf{H}_{j'}$ refers to unrelated price information spuriously correlated with $\mathbf{M}_j$, illustrating the first term in (5). The correct text feature "Mercedes-Benz" is $\mathbf{H}_j$; its absence reflects the omission of a relevant feature, as captured by the second term in (5). We focus on a single spurious pair $(j, j')$ for simplicity. Since our analysis depends on the total spurious feature probability (bounded by $C_s$), the results extend to multiple spurious features as long as their total probability stays within $C_s$.

## 3.3 Image-Grounded Text Decoder $G$ in Stage (S2)

Recall that $G$ is employed in Stage (S2) to generate synthetic text captions. In practice, the core idea behind the widely adopted approaches [25, 58, 48] is to train the encoder-decoder model $G$ and

---

[1]The columns $\mathbf{M}_j$ and $\mathbf{H}_j$ are column-orthonormal with each entry bounded by $\widetilde{O}(1/\sqrt{d_1})$, ensuring small inner products with isotropic noise.

leverage the high-quality image-text pairs $S_h$ to improve its performance. In this paper, we consider a simplified form of $G$, given by:

$$G(x_p) = \mathbf{V}^\top \sigma(\mathbf{W} x_p), \tag{6}$$

where $\sigma$ denotes the ReLU function. The parameters $\mathbf{W}$ and $\mathbf{V}$ are learned by solving

$$\min_{\mathbf{W},\mathbf{V}} L_C = \sum_{p \in S_h} \frac{1}{2} \left\| \mathbf{V}^\top \sigma(\mathbf{W} x_p) - y_p \right\|_2^2, \tag{7}$$

initialized at $\overline{\mathbf{W}}$ and $\overline{\mathbf{V}}$, using SGD with step size $\eta$. Although $G$ in (6) is a conceptual simplification, where $\sigma(\mathbf{W} x_p)$ acts as the encoder and $\mathbf{V}^\top$ as the decoder, it serves as a realistic abstraction to illustrate the underlying advantages of synthetic text caption generation.

### 3.4 Zero-Shot Generalization on Image Classification

We consider an out-of-domain (OOD) setting for testing images and text prompts as follows.

**Image:** Each test image $x$ can be approximated by a sparse coding model with dictionary $\mathbf{M}'$,

$$x = \mathbf{M}' z_x' + \xi_x, \quad \|z_x'\|_0 = \Theta(1), \quad \|z_x'\|_{\max} = \Theta(1), \tag{8}$$

where $\mathbf{M}' = \mathbf{M} \mathbf{P}_1$, and $\max_{i,j} |(\mathbf{P}_1)_{ij} - \delta_{ij}| \leq O(1/\sqrt{d})$ . The noise $\xi_x$ matches the training distribution (Assumption 3.1(d)) and $\delta_{ij}$ denotes the Kronecker delta function.

**Text:** Each class $k \in [K]$ has a prompt that has a sparse decomposition

$$y_k = \mathbf{H} z_{y_k}' + \xi_{y_k}, \quad \|z_{y_k}'\|_0 = \Theta(1), \quad \|z_{y_k}'\|_{\max} = \Theta(1). \tag{9}$$

If $x$ belongs to class $k$, then among all $K$ binary vectors $z_{y_{k'}}'$, $z_x'$ is maximally aligned with $z_{y_k}'$,

$$\|(z_x')^\top z_{y_k}'\|_2 > \|(z_x')^\top z_{y_{k'}}'\|_2, \quad \forall k' \neq k \tag{10}$$

This formulation reflects the intuition that $x$ belongs to class $k$ if its sparse representation is most similar to the sparse representation of class $k$'s text prompt.

## 4 Main Results

### 4.1 Intuition and Informal Insights

Before presenting our main results, we first offer an intuitive explanation of the encoder-learner's success. To learn the latent representation $z$ from input pair $(x, y)$, a well-trained image encoder $f$ and text encoder $g$ must ensure that each feature pair $(\mathbf{M}_j, \mathbf{H}_j)$ is captured by at least one neuron pair $(w_i, v_i)$, without interference from spurious signals. We call this a *purified feature*, meaning the neuron pair encodes only one true feature with no contamination. In this case, $\langle w_i, x \rangle \approx z_x^j$ and $\langle v_i, y \rangle \approx z_y^j$, so $f$ and $g$ recover the full latent space $z$. But in real data, where high-quality pairs in $S_h$ are rare and noisy pairs with misaligned image-text pairs in $S_w$ dominate, achieving this is difficult. See Appendix B.1 for proof sketches and we summarize main findings below:

**(I) SGD provably solves the nonconvex training problems (1).** The existing training dynamics and convergence analyses are limited to either single-modal contrastive learning [51] or linear networks [10, 33]. Theorem 4.1 provides a convergence analysis of SGD for solving the nonconvex ITCG problem when the network contains nonlinear activations for both modalities.

**(II) Failure of learning due to spurious correlations.** Theorem 4.2 provides a negative result: if $f$ and $g$ are directly trained on the raw data $S$, the model inevitably learns $\mathbf{M}_j$ and $\mathbf{M}_{j'}$ together via some $w_i$, and $\mathbf{H}_j$ and $\mathbf{H}_{j'}$ together via some $v_i$. As a result, the model fails to distinguish between these spuriously correlated features.

**(III) Successful learning with recaptioning and filtering**. Theorem 4.3 demonstrates that recaptioned texts significantly suppress spurious features and enhance relevant feature alignment. Building on this, Theorem 4.4 states that training $f$ and $g$ on the recaptioned data $\tilde{S}$ enables the resulting encoder pair to learn purified representations of $\mathbf{M}_j$ and $\mathbf{H}_j$ accurately, as if trained solely on sufficient high-quality data. This highlights the advantage of leveraging the recaptioned data $\tilde{S}_w$.

**(IV) Enhanced zero-shot image classification accuracy due to text recaptioning**. The advantage of using synthetic text captions is further validated in downstream tasks. As shown in Theorem 4.5, for a zero-shot out-of-domain multi-class image classification task, ITCP trained using $\tilde{S}$ achieves high accuracy, whereas ITCP directly using $S$ fails to generalize accurately.

## 4.2 Feature Purity Improvements in Converged Models via Recaptioned Data

We first characterize the training dynamics and convergence of solving (1) using SGD in Stage (S1) and (S4) in Section 2.1. Let $L^*$ and $\tilde{L}^*$ denote the optimal values of the contrastive loss on the raw dataset $S$ and the filtered dataset $\tilde{S}$, respectively. Note that $(\overline{\mathbf{W}}, \overline{\mathbf{V}})$ and $(\widetilde{\mathbf{W}}, \widetilde{\mathbf{V}})$ are the converged weights from contrastive training on $S$ and $\tilde{S}$ in Stage (S1) and (S4), respectively.

**Theorem 4.1** (**Convergence of ITCP**). *Suppose Assumptions 3.1 to 3.3 hold. Let the model complexity be $m = d^{1.01}$, initialized at $w_i^{(0)}, v_i^{(0)} \sim \mathcal{N}(0, \sigma_0^2 \mathbf{I}_{d_1})$, where $\sigma_0^2 = \Theta\left(\frac{1}{d_1 poly(d)}\right)$. After $T = \Theta\left(d^2 \log d\right)$ iterations with batch size $B = \Omega(d)$ and $\eta = O(1)$, the returned weights achieve a loss that is sufficiently close to the optimal loss in Stage (S1) and (S4), respectively, i.e.,*

$$(L(f_{\overline{\mathbf{W}}}, h_{\overline{\mathbf{V}}}) - L^*)/ |L^*| \le o(1), \quad (\tilde{L}(f_{\widetilde{\mathbf{W}}}, h_{\widetilde{\mathbf{V}}}) - \tilde{L}^*)/ \left|\tilde{L}^*\right| \le o(1). \tag{11}$$

**Remark 4.1.** Theorem 4.1 demonstrates that SGD iterations can converge to weights that achieve a near optimal loss of (1), respectively. This result is of independent interest, as existing training dynamics and convergence analyses for contrastive loss are limited to linear networks. Here, we extend such analysis to nonconvex optimization settings where the network contains nonlinear ReLU activations. Next, we characterize the feature purity of the learned models.

**Theorem 4.2** (**Unsuccessful learning of ITCP on raw data $S$ with low feature purity**). *For each neuron pair $(\bar{w}_i, \bar{v}_i)$ in $(\overline{\mathbf{W}}, \overline{\mathbf{V}})$, there exists a spurious feature pair $(j, j') \in [d]$ such that*

$$\bar{w}_i = \alpha_{i,j}\mathbf{M}_j + \alpha_{i,j'}\mathbf{M}_{j'} + \mathbf{r}_i, \quad \bar{v}_i = \alpha_{i,j}\mathbf{H}_j + \alpha_{i,j'}\mathbf{H}_{j'} + \mathbf{s}_i \tag{12}$$

*where $\alpha_{i,j}^2, \alpha_{i,j'}^2 = \Theta\left(\|\bar{w}_i\|_2^2 + \|\bar{v}_i\|_2^2\right)$ and $\|\mathbf{r}_i\|_2^2, \|\mathbf{s}_i\|_2^2 \le O((\|\bar{w}_i\|_2^2 + \|\bar{v}_i\|_2^2)/d)$. Moreover, for every spuriously correlated pair $(j, j')$, there exist at least $\Omega(1)$ neuron pairs $(\bar{w}_i, \bar{v}_i)$ that primarily learn the mixed feature pair $(\mathbf{M}_j, \mathbf{M}_{j'}, \mathbf{H}_j, \mathbf{H}_{j'})$.*

**Remark 4.2.** Theorem 4.2 indicates that the model learned by ITCP on raw data achieves only limited feature purity. Specifically, a neuron pair $(\bar{w}_i, \bar{v}_i)$ learns a mixture of image and text features, respectively. $\mathbf{M}_j$ and $\mathbf{M}_{j'}$ are always mixed together, as are $\mathbf{H}_j$ and $\mathbf{H}_{j'}$. As a result, the learned weights $\overline{\mathbf{W}}$ and $\overline{\mathbf{V}}$ fail to produce purified representations, making it difficult to distinguish between features $j$ and $j'$, which ultimately degrades downstream performance shown in (15).

**Theorem 4.3** (**Spurious feature suppression and relevant feature preservation by recaptioned texts**). *After $T = \Theta(d \log d)$ steps of SGD, the decoder $G$ in (6), finetuned by solving (7), converges to weights $(\hat{\mathbf{W}}, \hat{\mathbf{V}})$ with expected loss $L_C \le \Theta(1/d)$. The recaptioned texts in $\tilde{S}_w$ are computed by $\hat{y}_p = G(x_p)$. Then for any index $j \in [d]$ such that $|z_{x_p}^j| = 1$, the decoder output satisfies:*

$$\Pr(z_{\hat{y}_p}^j = 1 \mid |z_{x_p}^j| = 1) \ge 1 - \Theta(1/d), \quad \Pr(z_{\hat{y}_p}^{j'} = 1 \mid |z_{x_p}^j| = 1) \le \Theta(1/d), \quad \forall j' \neq j. \tag{13}$$

**Remark 4.3.** After captioning and filtering, the resulting text contains fewer spurious features and more aligned feature pairs than raw data. Compared with Assumption 3.3, the probability of spurious features can be reduced from a constant $C_s$ in $S_w$ to $\Theta(1/d)$ in $\tilde{S}_w$, while the probability of retaining all aligned feature pairs increases from $C_s$ in $S_w$ to $1 - \Theta(1/d)$ in $\tilde{S}_w$. The resulting dataset $\tilde{S} = S_h \cup \tilde{S}_w$ has better-aligned image-text pairs, enabling higher feature purity in contrastive training. We next show how ITCP trained on $\tilde{S}$ improves feature purity.

**Theorem 4.4** (**Successful learning of ITCP on filtered data $\tilde{S}$ with high feature purity**). *For each neuron pair $(\tilde{w}_i, \tilde{v}_i)$ in $(\widetilde{\mathbf{W}}, \widetilde{\mathbf{V}})$, there exists $j \in [d]$ such that $(\tilde{w}_i, \tilde{v}_i)$ primarily learns $(\mathbf{M}_j, \mathbf{H}_j)$*

$$\tilde{w}_i = \tilde{\alpha}_{i,j}\mathbf{M}_j + \tilde{\mathbf{r}}_i, \quad \tilde{v}_i = \tilde{\alpha}_{i,j}\mathbf{H}_j + \tilde{\mathbf{s}}_i \tag{14}$$

*where $\tilde{\alpha}_{i,j}^2 = \Theta(\|\tilde{w}_i\|_2^2 + \|\tilde{v}_i\|_2^2)$ and $\|\tilde{\mathbf{r}}_i\|_2^2, \|\tilde{\mathbf{s}}_i\|_2^2 \le O\left((\|\tilde{w}_i\|_2^2 + \|\tilde{v}_i\|_2^2)/d\right)$. Moreover, for every feature $j \in [d]$, there exist at least $\Omega(1)$ neuron pairs $(\tilde{w}_i, \tilde{v}_i)$ that primarily learn purified feature pair $(\mathbf{M}_j, \mathbf{H}_j)$.*

**Remark 4.4.** Theorem 4.4 indicates that the model learned by ITCP on filtered data achieves a purified representation. Specifically, a neuron pair $(\tilde{w}_i, \tilde{v}_i)$ learns one single feature pair $(\mathbf{M}_j, \mathbf{H}_j)$, respectively. As a result, $\widetilde{\mathbf{W}}$ and $\widetilde{\mathbf{V}}$ yield purified representations that effectively separate individual features, enabling improved downstream performance shown in (16).

### 4.3 Performance Comparison on Downstream Tasks

We next compare the performance of the models $(f_{\overline{\mathbf{W}}}, g_{\overline{\mathbf{V}}})$ and $(f_{\widetilde{\mathbf{W}}}, g_{\widetilde{\mathbf{V}}})$ on the zero-shot image classification problem with out-of-domain data described in Sections 2.2 and 3.4.

**Theorem 4.5** (Zero-Shot Image Classification). *For the OOD zero-shot $K$-class image classification problem, the model $(f_{\overline{\mathbf{W}}}, g_{\overline{\mathbf{V}}})$ from ITCP using raw data has a constant failure probability:*

$$\Pr\left(\arg\max_{k\in[K]}\langle f_{\overline{\mathbf{W}}}(x), g_{\overline{\mathbf{V}}}(y_k)\rangle = l_x\right) = 1 - \Theta(1); . \tag{15}$$

*In contrast, the model $(f_{\widetilde{\mathbf{W}}}, g_{\widetilde{\mathbf{V}}})$ from ITCP using filtered caption succeeds with high probability:*

$$\Pr\left(\arg\max_{k\in[K]}\langle f_{\widetilde{\mathbf{W}}}(x), g_{\widetilde{\mathbf{V}}}(y_k)\rangle = l_x\right) = 1 - o(1). \tag{16}$$

**Remark 4.5.** Theorem 4.5 first demonstrates that the zero-shot performance of $(f_{\overline{\mathbf{W}}}, g_{\overline{\mathbf{V}}})$ is unsatisfactory, resulting from the low feature purity in $(f_{\overline{\mathbf{W}}}, g_{\overline{\mathbf{V}}})$, as established in Theorem 4.2. Theorem 4.5 further shows that $(f_{\widetilde{\mathbf{W}}}, g_{\widetilde{\mathbf{V}}})$ achieves accurate classification. This success is attributed to high feature purity in $(f_{\widetilde{\mathbf{W}}}, g_{\widetilde{\mathbf{V}}})$, as described in Theorem 4.4. Note that Theorem 4.5 holds for image data with a distribution shift from the training data.

## 5 Experiment

### 5.1 Simulated Experiment

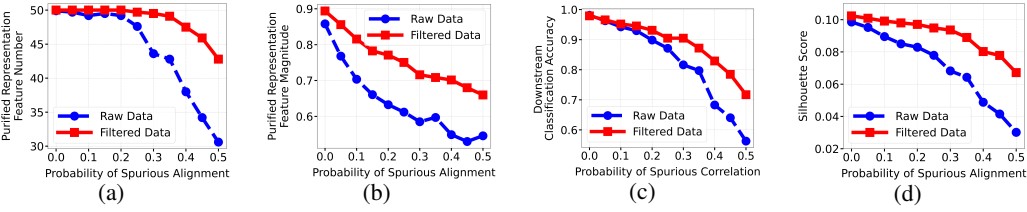

Figure 1: Performance comparison of ITCP on raw data and filtered (recaptioned) data when the probability of spurious correlation $C_s$ changes. (a) Number of features that have purified representation in the model (b) Average magnitude of purified presentations (c) Zero-shot out-of-domain classification accuracy (d) Silhouette Score with cosine distance.

**Experiment Setup.** We first validate our results via simulated experiments, using the same framework from Section 2.1. We adopt a more general spurious correlation model than Assumption 3.3, allowing each $\mathbf{M}_j$ to be spuriously linked with multiple $\mathbf{H}_{j'}$ ($j' \neq j$), while keeping the total spurious correlation probability at $C_s$. We set $d_1 = 2500$, $d = 50$, $|S_w| = 5000$, $|S_h| = 1000$, and use $m = 80$ neurons. Matrices $\mathbf{M}, \mathbf{H}$ are drawn from standard Gaussians and orthonormalized via QR decomposition. Sparse codes $z_x$ follows Bernoulli(0.1) Noise variance $\sigma_\xi^2 = 1/d$. SGD runs with batch size 500 and step size 0.001. Downstream evaluation uses 5-way classification with test $z_x \sim$ Bernoulli(0.2); class codes $z_{y_k}$ partition the $d$-dim space. Results are averaged over 20 trials. Models $(\overline{\mathbf{W}}, \overline{\mathbf{V}})$ and $(\widetilde{\mathbf{W}}, \widetilde{\mathbf{V}})$ are trained on raw and filtered data, respectively.

**Improved feature representation using filtered (recaptioned) data.** We say a weight $\bar{w}_i$ learn a *purified representation* of $\mathbf{M}_j$ if its projection along $\mathbf{M}_j$ achieves the largest magnitude and satisfies $|\langle \bar{w}_i, \mathbf{M}_j\rangle|/\|\bar{w}_i\| > 0.5$. The same applies to $(\widetilde{\mathbf{W}}, \widetilde{\mathbf{V}})$. Figure 1(a) shows the number of features $\mathbf{M}_j$ (out of $d = 50$ total features) for which at least one neuron in $\overline{\mathbf{W}}$ (or $\widetilde{\mathbf{W}}$, respectively) learns a purified

Table 2:  of CLIP and LaCLIP on Accuracy (%) and Silhouette Score.

| Model | Food-101 | | CIFAR-10 | | Caltech-101 | | CIFAR-100 | | Pets | | STL-10 | |
|---|---|---|---|---|---|---|---|---|---|---|---|---|
| | Acc | SS | Acc | SS | Acc | SS | Acc | SS | Acc | SS | Acc | SS |
| CC12M CLIP | 50.8 | 0.034 | 64.9 | 0.113 | 77.4 | 0.225 | 38.5 | 0.005 | 64.1 | 0.069 | 91.0 | 0.195 |
| CC12M LaCLIP | **60.7** | **0.038** | **75.1** | **0.157** | **83.3** | **0.276** | **43.9** | **0.029** | **72.4** | **0.070** | **95.1** | **0.273** |
| RedCaps CLIP | 81.5 | 0.125 | 70.4 | 0.100 | 72.8 | 0.210 | 39.9 | $-0.002$ | **82.7** | **0.091** | **92.8** | 0.226 |
| RedCaps LaCLIP | **85.0** | **0.175** | **74.8** | **0.107** | **76.4** | **0.233** | **40.7** | **0.011** | 78.2 | 0.074 | 91.4 | **0.275** |
| LAION CLIP | 85.5 | 0.116 | 93.0 | 0.181 | 91.2 | 0.258 | 71.7 | 0.078 | 90.1 | 0.122 | 97.3 | 0.223 |
| LAION LaCLIP | **86.5** | **0.148** | **93.5** | **0.215** | **92.4** | **0.306** | **73.9** | **0.108** | **90.9** | **0.152** | **98.4** | **0.260** |

representation. The results show that ITCP trained on filtered data learns purified representations for nearly all features, even at high spurious correlation levels ($C_s = 0.3$). In contrast, ITCP on raw data degrades significantly, with purity dropping faster as $C_s$ increases. Moreover, Figure 1(b) shows the average of the largest projection magnitudes among neurons that learn purified features. The magnitude from $\widetilde{\mathbf{W}}$ (ITCP on filtered data) is consistently higher than that from $\overline{\mathbf{W}}$, indicating stronger purified representations. This aligns with Theorems 4.2, 4.4 and Remark 4.4.

**Improved zero-shot out-of-domain performace using filtered (recaptioned) data.** Figure 1(c) compares the classification accuracy of both models on zero-shot out-of-domain data. The model trained on filtered data consistently outperforms the one trained on raw data, with the performance gap widening as spurious correlations in the raw data increase. We also adopt the widely used Silhouette Score (SS) with cosine distance [57, 32, 62] to evaluate feature embedding quality in different clusters, as shown in Figure 1(d). A higher SS indicates better intra-class alignment and inter-class orthogonality, reflecting more purified representations. These results verify Theorem 4.5.

**Impact of feature purity.** When $C_s$ reaches 0.35 in Figure 1, even the filtered data fails to maintain full feature purification: the number of neurons learning disentangled representations of all $d = 50$ features drops significantly (Figure 1(a)), and the SS (Figure 1(d)) and classification accuracy (Figure 1(c)) both decline sharply. This highlights that *feature purity*—the extent to which each neuron aligns to a single semantic direction—is a key bottleneck in contrastive pretraining and downstream generalization. We provide extra results in Appendix A.1.

## 5.2 Experiments on Practical Data and Models

**LaCLIP improves generalization over CLIP via recaption.** Tables 2 compare CLIP [37] and LaCLIP [13], which share the same architecture and datasets, except LaCLIP replaces part of the original captions with LLM-generated rewrites. "CC12M CLIP" denotes a CLIP model pretrained on raw CC12M [8], while "CC12M LaCLIP" uses the same model and data but with LLM-rewritten captions. Other models are obtained similarly using RedCaps [12] and LAION [40] datasets. We evaluate their zero-shot classification accuracy and Silhouette Scores on various downstream datasets. LaCLIP generally outperforms CLIP in both metrics, empirically validating that higher-quality captions improve zero-shot generalization. Additional ImageNet results are reported in Table 3 of Appendix A.2.

Next, we study the feature purity using a CLIP model pretrained on CC3M [41]. Both the image and text encoders are 12-layer transformers that produce features in $\mathbb{R}^{768}$, which are subsequently projected into a shared embedding space of $\mathbb{R}^{512}$ through final linear projection layers, as illustrated in Figure 6 of Appendix A.2. The final linear projection layer has 512 neurons and is functionally aligned with $\mathbf{V}$ in our theoretical model. We now present two key findings from this setting:

**Purified neurons enhance generalization.** To investigate the effect of feature purity on generalization, we prune the neurons in the final linear layer in different ways and evaluate the resulting zero-shot classification performance. Specifically, we rank the 512 neurons by their average pairwise absolute cosine similarity to all other neurons, from lowest to highest. The absolute cosine similarity of neurons $v_j$, $v_{j'}$ is computed as $|\langle v_j, v_{j'} \rangle| / \|v_j\| \|v_{j'}\|$ for all $j, j' \in \{1, 2, \ldots, 512\}$. A lower average indicates higher feature purity (i.e., more orthogonal representations), while a higher value suggests feature mixing. We evaluate three pruning strategies: (1) retaining **high-purity** neurons, i.e., with lowest similarity, (2) retaining **low-purity** neurons, i.e., with highest similarity, and (3) retaining a **random** subset of neurons. The number of retained neurons is varied from 200 to 500. As shown in Figure 2 (a-c,e-g), downstream performance is the best when retaining high-purity neurons,

followed by random selection, with low-purity neurons performing the worst. These results highlight the critical role of purified features in downstream generalization.

**Data misalignment reduces feature purity.** To study how image-text misalignment affects feature purity, we randomly shuffling texts across image-text pairs in CC3M with probability $C_m$, as illustrated in Figure 7 of Appendix A.2, thereby introducing a controlled probability of modality misalignment. We then use the shuffled dataset to fine-tune the last linear projection layer only of the pretrained CLIP model, freezing other layers. We then compute the cosine similarities of all 512 neuron weight vectors $v_j \in \mathbb{R}^{768}$ of the fine-tuned model. Figure 2 (d) reports the average absolute cosine similarity of all neuron pairs, while (h) presents a histogram of cosine similarity $\langle v_j, v_{j'} \rangle / (\|v_j\| \|v_{j'}\|)$. One can see that as $C_m$ increases, the average absolute cosine similarity increases, and the neurons become less orthogonal to each other and tend to encode mixed representations, resulting in lower feature purity. This coincides with the decreases classification accuracy in downstream tasks, as shown in Table 4 of Appendix A.2.

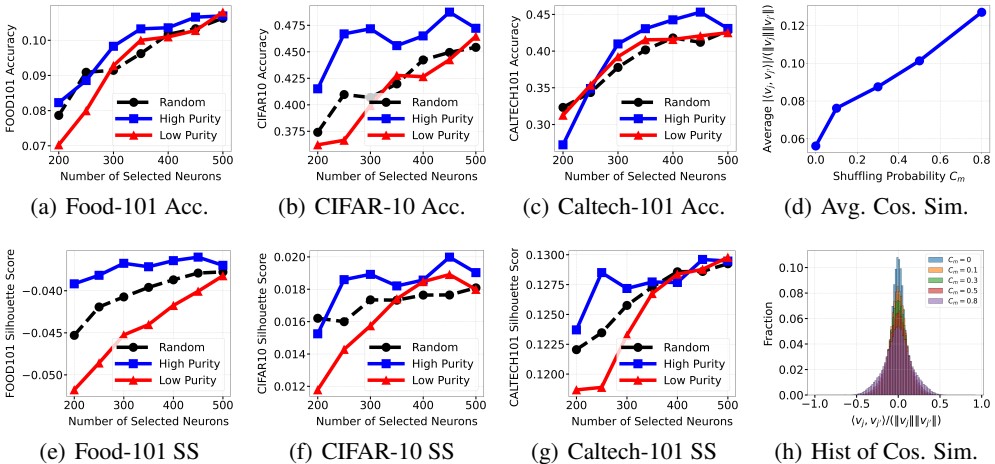

Figure 2: Left (a–c,e-g): Retaining high-purity neurons outperform random and low-purity neurons in downstream tasks. More datasets shown in Figure 8. Right(d,h): When $C_m$ increases, the neurons have higher cosine similarity and reduced feature purity.

# 6 Conclusion

This paper provides a theoretical analysis of contrastive learning with nonlinear networks, linking training dynamics to generalization. We identify feature purity as central to generalization and show that text recaptioning enhances purity and zero-shot performance. The theory is empirically validated on benchmarks. Future work includes extending to Transformer models and tasks like retrieval and visual question answering.

### Acknowledgments

This work was supported by National Science Foundation (NSF) #2430223, Army Research Office (ARO) W911NF-25-1-0020, and the Rensselaer-IBM Future of Computing Research Collaboration (http://airc.rpi.edu). The work of Shuai Zhang was supported by NSF #2349879. We also thank all anonymous reviewers for their constructive comments.

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

The overall structure of the appendix is as follows. Each appendix provides supplementary information that supports the main content of this document but is not included in the main body to maintain clarity and flow.

- **Appendix A: Extra Experiments**
  Additional experiments including both synthetic simulations and CLIP/LaCLIP evaluations on omitted datasets.
    - **A.1 Extra Simulated Experiment**
      Complements Section 5.1 with further analysis of neuron behavior trained on simulated data.
    - **A.2 Extra CLIP/LaCLIP Experiment**
      Complements Section 5.2 by evaluating on datasets omitted due to space.
- **Appendix B: Preliminaries**
  Mathematical preliminaries and notation used throughout the paper. A *proof sketch* is also provided to outline the key ideas behind the main results.
- **Appendix C: Technical Lemmas**
  Full statements and proofs of supporting lemmas used in the theoretical analysis.
- **Appendix D–J: Proofs and Theoretical Analysis**
    - **Appendix D–F: ITCP on Raw Data (Phase I–III)**
      Theoretical proof of ITCP across three training phases on raw data.
    - **Appendix G: Captioning**
      Theoretical proof of reception using high quality data.
    - **Appendix H: Filtering**
      Theoretical proof of filtering noisy caption-text pairs.
    - **Appendix I: ITCP on Synthetic (Recaptioned) Data**
      Theoretical proof of training dynamics when using synthetic recaptions.
    - **Appendix J: Downstream Task Evaluation**
      Theoretical implications for performance on downstream tasks.
- **Checklist**

# A Extra Experiment

All experiments were conducted on an internal compute cluster using 8 NVIDIA A5000 GPUs with 24 GB memory each, and each run completed within 50 GPU-hours. No large-scale pretraining or resource-intensive tuning was performed beyond the reported experiments.

## A.1 Extra Simulated Experiment

This section extends the analysis in Section 5.1 by providing additional simulated experiments on neuron behavior under synthetic data training.

**Neurons trained on filtered data exhibit a more concentrated distribution.** Figure 3 visualizes the histograms of $|\langle \bar{v}_i, \mathbf{H}_j \rangle|/\|\bar{v}_i\|$ and $|\langle \tilde{v}_i, \mathbf{H}_j \rangle|/\|\tilde{v}_i\|$ for all $i \in [m]$ and $j \in [d]$. The values of $|\langle \tilde{v}_i, \mathbf{H}_j \rangle|/\|\tilde{v}_i\|$ are more concentrated, typically around 0.05 and 0.7. In contrast, the values for $|\langle \bar{v}_i, \mathbf{H}_j \rangle|/\|\bar{v}_i\|$ are less concentrated. This phenomenon is consistent with Theorem 4.4, which indicates that for every $\mathbf{H}_j$, certain neurons $\tilde{v}_i$ in $\widetilde{\mathbf{V}}$ predominately learns $\mathbf{H}_j$. In such cases, $|\langle \tilde{v}_i, \mathbf{H}_j \rangle|$ approaches 1, while $|\langle \tilde{v}_i, \mathbf{H}_{j'} \rangle|/\|\tilde{v}_i\|$ approaches 0 for $j' \neq j$. The concentrated values of 0.05 and 0.7 observed in Figure 3 are due to noise in the data. In contrast, feature alignment is less significant for $\overline{\mathbf{V}}$, leading to less concentration of the corresponding values. Similar results are obtained for image encoder $|\langle w_i, \mathbf{M}_j \rangle|$, deferred to Figure 4.

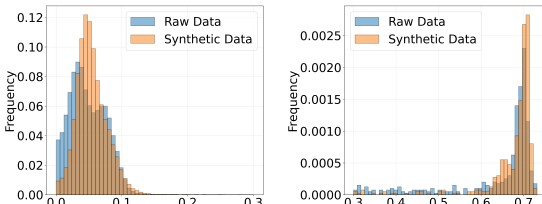

Figure 3: Histogram of $|\langle \bar{v}_i, \mathbf{H}_j \rangle|/\|\bar{v}_i\|$ for ITCP on raw data and $|\langle \tilde{v}_i, \mathbf{H}_j \rangle|/\|\tilde{v}_i\|$ for ITCP on filtered data (split into two figures to highlight the significant differences in the value distributions).

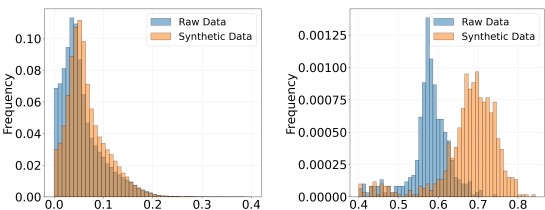

Figure 4: Histogram of $|\langle \bar{w}_i, \mathbf{M}_j \rangle|/|\bar{w}_i|$ for ITCP on raw data and $|\langle \tilde{w}_i, \mathbf{M}_j \rangle|/\tilde{w}_i$ for ITCP on filtered data (split into two figures to highlight the significant differences in the value distributions).

**Enhanced class separation of downstream tasks by ITCP with recaptioned data**. Figure 5 visualizes the t-distributed stochastic neighbor embedding (t-SNE) of the feature embeddings generated by the two models, computed as $f_{\overline{\mathbf{W}}}(x_p)$ and $f_{\widetilde{\mathbf{W}}}(x_p)$ for each $x_p$, respectively. The t-SNE method projects the high-dimensional embeddings onto a two-dimensional map. One can see that the embeddings from different groups are more distinctly separated in the model trained using ITCP on recaptioned data, indicating that this approach achieves better feature alignment.

## A.2 Extra Experiment on CLIP and LaCLIP

To complement the results in Section 5.2, we report additional experiments on CLIP and LaCLIP using datasets omitted from the main text due to space constraints.

**ImageNet Results.** The LaCLIP variants consistently surpass their CLIP counterparts on both Top-1 and Top-5 accuracy. Higher silhouette scores further indicate cleaner feature separation after recaptioning, in line with our theoretical predictions.

**CLIP architecture.** Figure 6 illustrates the CLIP architecture used in our experiments. Both image and text inputs are independently encoded by 12-layer transformer backbones, each producing a

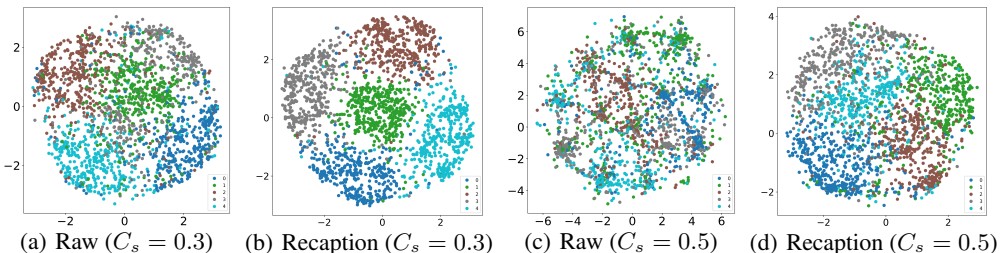

(a) Raw ($C_s = 0.3$)   (b) Recaption ($C_s = 0.3$)   (c) Raw ($C_s = 0.5$)   (d) Recaption ($C_s = 0.5$)

Figure 5: t-SNE visualization of text embedding with spurious correlation probability $C_s$.

Table 3: Comparison of CLIP and LaCLIP on ImageNet: Top-1 (%), Top-5 (%), and Silhouette Score.

| Model | Top-1 (%) | Top-5 (%) | Silhouette |
|---|---|---|---|
| CC12M CLIP | 35.04 | 62.10 | -0.014639 |
| CC12M LaCLIP | **42.62** | **70.17** | $\mathbf{-0.008141}$ |
| LAION-400M CLIP | 58.34 | 84.73 | -0.029893 |
| LAION-400M LaCLIP | **62.27** | **86.34** | $\mathbf{-0.056593}$ |
| RedCaps CLIP | 37.66 | 63.31 | -0.022045 |
| RedCaps LaCLIP | **39.66** | **66.06** | $\mathbf{-0.012269}$ |

768-dimensional feature vector. These features are then projected into a shared 512-dimensional embedding space through learned linear projection matrices $\mathbf{W} \in \mathbb{R}^{768 \times 512}$ and $\mathbf{V} \in \mathbb{R}^{768 \times 512}$, corresponding to the image and text encoders in our theorem, defined in Eq. (2). The resulting embeddings are aligned via a contrastive loss that maximizes similarity for matched image-text pairs while minimizing similarity for unmatched pairs. This architecture forms the foundation for our analyses on neuron selection and feature purity in the shared embedding space.

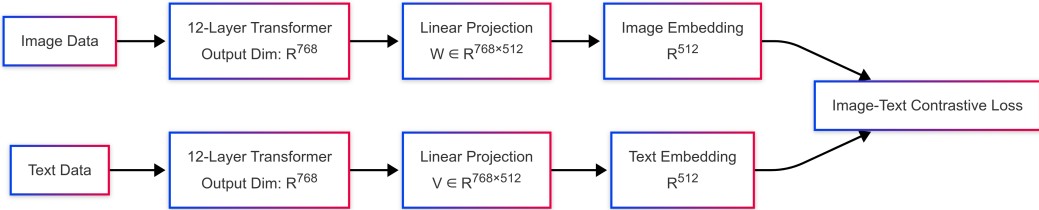

Figure 6: Architecture of CLIP used in our experiments. Both image and text encoders are 12-layer transformers that output features in $\mathbb{R}^{768}$, which are then projected into a shared $\mathbb{R}^{512}$ embedding space via final linear projection layers $\mathbf{W}$ and $\mathbf{V}$, corresponding to Eq. (2) and Eq. (3) in our theoretical analysis. Contrastive loss is computed between the resulting image and text embeddings.

**Simulating Modality Misalignment via Caption Shuffling.** Figure 7 illustrates how modality misalignment is introduced by randomly shuffling text captions across image-text pairs with probability $C_m$, resulting in noisy supervision for contrastive learning.

**Purified neuron selection enhances generalization.** Figure 8 presents additional experimental results on CIFAR-100, Pets, and STL-10, complementing the main results reported in Figure 2. Due to space constraints, we include only Food-101, CIFAR-10, and Caltech-101 in the main text. All experiments follow the same protocol, evaluating zero-shot classification accuracy and Silhouette Score under different neuron selection strategies. These results consistently support our core finding: selecting high-purity neurons leads to improved downstream performance across diverse datasets.

**Higher shuffling probability leads to reduced generalization and feature purity.** Table 4 presents additional experimental results on CLIP models finetuned with different levels of randomly shuffling probability $C_m$ to simulate spurious correlation, showing that both accuracy and Silhouette Score consistently decrease as $C_m$ increases.

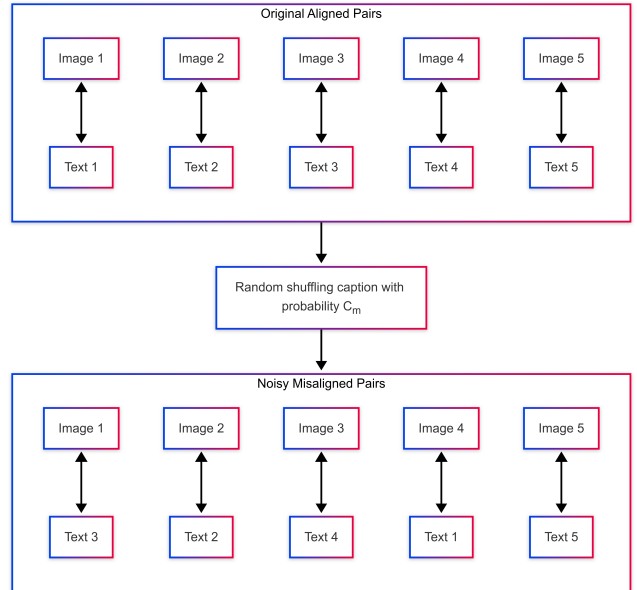

Figure 7: Simulating Modality Misalignment via Caption Shuffling. Starting from original aligned image-text pairs, a controlled probability $C_m$ of misalignment is introduced by randomly shuffling the text captions. This results in noisy pairs that reflect varying levels of spurious correlations.

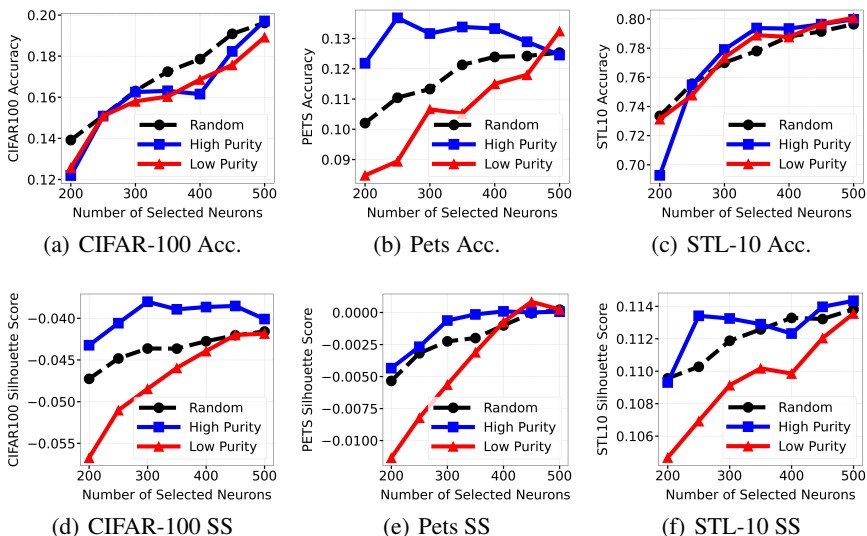

Figure 8: Zero-shot classification accuracy (top) and Silhouette Score (bottom) under different neuron selection strategies for CIFAR-100, Pets, and STL-10 datasets.

## B    Preliminaries

We first restate some important notations used in the Appendix, which are summarized in Table 5.

### B.1    Proof Scratch

Theorem 4.1 is proven by integrating the convergence analyses in Appendix F and Appendix I. Appendix F establishes convergence for ITCP on raw data, while Appendix I extends the convergence result to ITCP on synthetic data. Together, they verify that SGD with ReLU networks achieves near-optimal contrastive loss on both datasets.

Table 4: Accuracy (%) and Silhouette Score of CLIP models finetuned with varying $C_m$ on six datasets.

| **Dataset** | $C_m = 0$ | | $C_m = 0.1$ | | $C_m = 0.3$ | | $C_m = 0.5$ | | $C_m = 0.8$ | |
|---|---|---|---|---|---|---|---|---|---|---|
| | Acc | SS | Acc | SS | Acc | SS | Acc | SS | Acc | SS |
| Caltech101 | 59.7 | 0.160 | 48.2 | 0.124 | 47.9 | 0.121 | 43.6 | 0.117 | 44.5 | 0.115 |
| CIFAR-10 | 57.9 | 0.030 | 50.7 | 0.012 | 49.5 | 0.013 | 46.5 | 0.013 | 44.1 | 0.011 |
| CIFAR-100 | 26.4 | $-0.038$ | 19.5 | $-0.042$ | 17.8 | $-0.043$ | 17.4 | $-0.044$ | 16.2 | $-0.048$ |
| Food-101 | 12.9 | $-0.073$ | 10.9 | $-0.052$ | 10.9 | $-0.056$ | 11.1 | $-0.057$ | 11.1 | $-0.059$ |
| Pets | 13.9 | $-0.005$ | 13.3 | $-0.006$ | 13.2 | $-0.009$ | 13.4 | $-0.011$ | 12.6 | $-0.012$ |
| STL-10 | 86.3 | 0.164 | 79.8 | 0.103 | 79.2 | 0.102 | 78.8 | 0.100 | 78.3 | 0.097 |

Table 5: Summary of Notations

| Notations | Annotation |
|---|---|
| $\mathbf{M} \in \mathbb{R}^{d_1 \times d}, \mathbf{H} \in \mathbb{R}^{d_1 \times d}$ | $\mathbf{M}$ is the image dictionary matrix, $\mathbf{H}$ is the text dictionary matrix. |
| $\mathbf{W} \in \mathbb{R}^{m \times d_1}, \mathbf{V} \in \mathbb{R}^{m \times d_1}$ | $\mathbf{W}$ is the weight of image encoder, $\mathbf{V}$ is the weight of text encoder. |
| $x_p \in \mathbb{R}^{d_1}, y_p \in \mathbb{R}^{d_1}$ | $x_p$ and $y_p$ represent an image and a text data, respectively. |
| $z_{x_p}, z_{y_p} \in \mathbb{R}^d$ | $z_{x_p}$ and $z_{y_p}$ are the sparse signals of image and text, respectively. $z_{y_k}$ is the sparse signal for the text prompt $y_k$. |
| $z_{x_p}^j, z_{y_p}^j$ | $z_{x_p}^j$ is the $j$-th coordinate of $z_{x_p}$; $z_{y_p}^j$ is the $j$-th coordinate of $z_{y_p}$. |
| $L, L_C$ | $L$ is the loss for ITCP; $L_C$ is the loss for Image-grounded Text Decoding. |
| $S = S_h \cup S_w$ | $S_w$ is the noisy web low-quality dataset; $S_h$ is the human-annotated high-quality dataset. |
| $\tilde{S} = S_h \cup \tilde{S}_w$ | $\tilde{S}_w$ replaces noisy captions in $S_w$ with synthetic captions. |
| $T_1$ | Phase I of ITCP with $b_i^{(t)} = 0$. |
| $T_2$ | Phase II of ITCP with $b_i^{(t+1)} = (1 + \frac{\eta}{d})b_i^{(t)}$. |
| $T_3$ | Phase III of ITCP with $b_i^{(t+1)} = b_i^{(T_2)}$. |
| $T_C$ | Stage of training caption generators. |
| $\mathcal{S}_{j,\text{sure}}$ | The set of well-initialized neurons $(w_i, v_i)$ on features $(\mathbf{M}_j, \mathbf{H}_j)$. |

Theorem 4.2 is proven across Appendix D, Appendix E, and Appendix F. Specifically, Appendix D models Phase I training ($t \leq T_1$) and proves that neurons simultaneously align with true features and spuriously correlated features due to comparable gradient contributions, preventing pure feature separation. Appendix E analyzes Phase II training ($T_1 < t \leq T_2$) and shows that this spurious alignment continues to strengthen, as neurons with initial mixed alignment further amplify their entanglement during continued SGD updates. Appendix F establishes the convergence behavior during Phase III ($T_2 < t \leq T_3$), showing that the network stabilizes into mixed solutions where each neuron represents a combination of multiple features. These detailed stages collectively prove the failure of purified feature alignment as formalized in Theorem 4.2.

Theorem 4.3 is proven across Appendix G and Appendix H. Specifically, Appendix G analyzes the captioning stage, where the decoder is fine-tuned on clean data to generate synthetic captions. It proves that for neurons aligned with true features, the alignment towards the true features grows exponentially while the alignment towards spurious features remains negligible. This ensures that the synthetic captions preserve relevant features and suppress spurious ones. Appendix H then formalizes the filtering process, demonstrating that after replacing noisy captions with synthetic ones, the resulting dataset satisfies much stronger feature purity conditions, with spurious correlations

suppressed to $\Theta(1/d)$ and true features preserved with probability $1 - \Theta(1/d)$. These results directly support the purified feature learning described in Theorem 4.3.

Theorem 4.4 is proven in Appendix I, which integrates the proofs of Phase I, Phase II, and Phase III for ITCP on synthetic data. Specifically, Appendix I first establishes in Phase I that purified training pairs allow neurons aligned with true features to grow exponentially without spurious interference. It then shows in Phase II that these alignments continue to strengthen while suppressing non-informative neurons, leading to clear feature separation. Finally, it proves in Phase III that the model converges, achieving a bounded final loss and dominant true feature alignment. Since the overall proof structure closely mirrors that of Theorem 4.2 (which was proven separately across Appendix D, Appendix E, and Appendix F), we consolidate all stages into a single appendix for brevity and clarity.

Theorem 4.5 is proven in Appendix J, which analyzes the downstream zero-shot classification. Appendix J shows that for ITCP on raw data, spurious features cause a constant classification error, while for ITCP on synthetic data, true and spurious features become separable with high probability, leading to an $o(1)$ error rate. This directly supports the main text conclusion on downstream generalization.

## B.2 Feature Coupling and Expected Values in $S_w$

The following Assumption B.1 corresponds to the more specific forms of Assumptions 3.2 and 3.3 discussed earlier.

**Assumption B.1** (High and low quality pairs). *The high-quality image-text pairs in $S_h$ have size $|S_h| = \Theta(d^2)$. The low-quality image-text pairs in $S_w$ have size $|S_w| = poly(d)| \gg \omega(d^2)$.*

*In $S_h$, for a positive pair $(x_p, y_p)$, we assume perfect alignment, meaning $z_{x_p} = z_{y_p}$. Consequently, the following holds:*

$$\mathbb{E}\left[z_{x_p}^j z_{y_p}^j\right] = \frac{C_z}{d}, \quad \mathbb{E}\left[z_{x_p}^j z_{y_p}^{j'}\right] = \Theta\left(\frac{1}{d^2}\right), \quad j' \neq j \tag{17}$$

*To model the misaligned features in low-quality pairs in $S_w$, where spurious misalignment occurs at a non-negligible level, we assume $[d]$ can be divided into $d/2$ disjoint sets, each containing exactly two entries. Let $(j, j') \subset [d]$ denote one such set, referred to as a "spuriously correlated set." The following assumptions capture the nature of spurious and true alignments:*

$$\Pr(|z_{y_p}^{j'}| = 1 \mid |z_{x_p}^j| = 1) = \Theta(1) < \frac{1}{2},$$
$$\Pr(|z_{y_p}^{j'}| = 1 \mid |z_{x_p}^j| = 1) + \Pr(|z_{y_p}^j| = 1 \mid |z_{x_p}^j| = 1) = 1. \tag{18}$$

These assumptions imply that true alignment dominates, with $\Pr(|z_{y_p}^j| = 1 \mid |z_{x_p}^j| = 1) > \frac{1}{2}$, while spurious alignment exists at a constant percentage level, making it non-negligible. The intuition behind this assumption is that each feature $j$ is paired with exactly one spuriously correlated feature $j'$, ensuring that $j$ is not associated with any other feature $j'' \neq j'$. This design simplifies the analysis while effectively capturing the key challenges posed by low-quality data.

Then, for a positive pair $(x_p, y_p)$ with $p$ in $S_w$, we have:

$$\mathbb{E}\left[z_{x_p}^j z_{y_p}^j\right] + \mathbb{E}\left[z_{x_p}^j z_{y_p}^{j'}\right] = \frac{C_z}{d},$$
$$\mathbb{E}\left[z_{x_p}^j z_{y_p}^{j'}\right] = \Theta\left(\frac{1}{d}\right) < \frac{C_z}{2d}. \tag{19}$$

where $(j, j')$ is a spuriously correlated set.

For negative pairs $(x_p, y_q)$, where $p \neq q$, and $p, q \in S$, we have:

$$\mathbb{E}\left[z_{x_p}^j z_{y_q}^{j'}\right] = \Theta\left(\frac{1}{d^2}\right), \quad \forall j, j' \in [d]. \tag{20}$$

In $S_w$, mismatched text and image pairs are prevalent compared to $S_h$. For a postive pair $(x_p, y_p)$, we assume $\frac{\log(1/c_0)}{2\log d} < \Pr(|z_{y_p}^{j'}| = 1 \mid |z_{x_p}^j| = 1) < \frac{1}{2}$. To model this, we assume that for each primary

feature $j \in [d]$, there exists exactly one spurious feature $j'$ such that $j$ and $j'$ are uniquely coupled. This implies that $j$ cannot be associated with any other feature $j'' \neq j'$. Mathematically, the coupling is defined as:

$$\Pr(|z_{y_p}^{j'}| = 1 \mid |z_{x_p}^{j}| = 1) + \Pr(|z_{y_p}^{j}| = 1 \mid |z_{x_p}^{j}| = 1) = 1. \tag{21}$$

For a positive pair $(x_p, y_p)$ in $S_w$, the probabilities of spurious and aligned features are further constrained:

$$\frac{\log(1/c_0)}{2 \log d} < \Pr(|z_{y_p}^{j'}| = 1 \mid |z_{x_p}^{j}| = 1) < \frac{1}{2}, \tag{22}$$

The lower bound is established in Lemma C.8.

and:

$$\Pr(|z_{y_p}^{j}| = 1 \mid |z_{x_p}^{j}| = 1) = 1 - \Pr(|z_{y_p}^{j'}| = 1 \mid |z_{x_p}^{j}| = 1). \tag{23}$$

Under these assumptions, the expected values for the aligned and spurious features are calculated as follows:

For the aligned feature $j$, we have:

$$\begin{aligned}
\mathbb{E}\left[z_{x_p}^{j} z_{y_p}^{j}\right] &= \Pr(|z_{y_p}^{j}| = 1, |z_{x_p}^{j}| = 1) \\
&= \Pr(|z_{y_p}^{j}| = 1 \mid |z_{x_p}^{j}| = 1) \cdot \Pr(|z_{x_p}^{j}| = 1) \\
&= \Pr(|z_{y_p}^{j}| = 1 \mid |z_{x_p}^{j}| = 1) \cdot \frac{C_z}{d}.
\end{aligned} \tag{24}$$

For the spurious feature $j'$, we have:

$$\begin{aligned}
\mathbb{E}\left[z_{x_p}^{j} z_{y_p}^{j'}\right] &= \Pr(|z_{y_p}^{j'}| = 1, |z_{x_p}^{j}| = 1) \\
&= \Pr(|z_{y_p}^{j'}| = 1 \mid |z_{x_p}^{j}| = 1) \cdot \Pr(|z_{x_p}^{j}| = 1) \\
&= \Pr(|z_{y_p}^{j'}| = 1 \mid |z_{x_p}^{j}| = 1) \cdot \frac{C_z}{d}
\end{aligned} \tag{25}$$

The total expected value across both aligned and spurious features satisfies:

$$\mathbb{E}\left[z_{x_p}^{j} z_{y_p}^{j}\right] + \mathbb{E}\left[z_{x_p}^{j} z_{y_p}^{j'}\right] = \frac{C_z}{d} \tag{26}$$

Here, $j'$ denotes the spurious feature associated with $j$.

## B.3   Gradient

The contrastive loss in vision-language models (VLM) is defined as follows:

$$\begin{aligned}
L(f^{(t)}, h^{(t)}) = \sum_{p \in S} \Bigg[ &- \langle f^{(t)}(x_p), h^{(t)}(y_p) \rangle + \sum_{x_n \in \mathfrak{N}'} \frac{\left( \langle f^{(t)}(x_n), h^{(t)}(y_p) \rangle \right)^2}{2\tau} \\
&+ \sum_{y_n \in \mathfrak{N}'} \frac{\left( \langle f^{(t)}(x_p), h^{(t)}(y_n) \rangle \right)^2}{2\tau} \Bigg],
\end{aligned} \tag{27}$$

where $\tau > 0$ is a temperature parameter.

We perform stochastic gradient descent (SGD) on this contrastive loss. Let $f^{(t)}$ and $h^{(t)}$ be the image encoder and text encoder networks at iteration $t$, respectively. The network parameters are updated as follows:

$$w_i^{(t+1)} \leftarrow w_i^{(t)} - \eta \nabla_{w_i} L(f^{(t)}, h^{(t)}), \tag{28}$$

$$v_i^{(t+1)} \leftarrow v_i^{(t)} - \eta \nabla_{v_i} L(f^{(t)}, h^{(t)}), \tag{29}$$

where $b_i^{(t)}$, the bias term, is manually tuned during training and thus excluded from gradient updates.

The gradient of $L(f^{(t)}, h^{(t)})$ with respect to $w_i^{(t)}$ at iteration $t$ is given by:

$$\nabla_{w_i} L(f^{(t)}, h^{(t)}) = - \langle v_i^{(t)}, y_p \rangle x_p \cdot \mathbf{1}_{\left|\langle w_i^{(t)}, x_p \rangle\right| \geq b_i^{(t)}} \cdot \mathbf{1}_{\left|\langle v_i^{(t)}, y_p \rangle\right| \geq b_i^{(t)}}$$

$$+ \sum_{x_n \in \mathfrak{N}} \frac{\langle f^{(t)}(x_n), h^{(t)}(y_p) \rangle \langle v_i^{(t)}, y_p \rangle x_n}{\tau} \cdot \mathbf{1}_{\left|\langle w_i^{(t)}, x_n \rangle\right| \geq b_i^{(t)}} \cdot \mathbf{1}_{\left|\langle v_i^{(t)}, y_p \rangle\right| \geq b_i^{(t)}}$$

$$+ \sum_{y_n \in \mathfrak{N}} \frac{\langle f^{(t)}(x_p), h^{(t)}(y_n) \rangle \langle v_i^{(t)}, y_n \rangle x_p}{\tau} \cdot \mathbf{1}_{\left|\langle w_i^{(t)}, x_p \rangle\right| \geq b_i^{(t)}} \cdot \mathbf{1}_{\left|\langle v_i^{(t)}, y_n \rangle\right| \geq b_i^{(t)}}.$$

$$(30)$$

Similarly, the empirical gradient of $L(f^{(t)}, h^{(t)})$ with respect to $v_i^{(t)}$ is:

$$\nabla_{v_i} L(f^{(t)}, h^{(t)}) = - \langle w_i^{(t)}, x_p \rangle y_p \cdot \mathbf{1}_{\left|\langle w_i^{(t)}, x_p \rangle\right| \geq b_i^{(t)}} \cdot \mathbf{1}_{\left|\langle v_i^{(t)}, y_p \rangle\right| \geq b_i^{(t)}}$$

$$+ \sum_{x_n \in \mathfrak{N}} \frac{\langle f^{(t)}(x_n), h^{(t)}(y_p) \rangle \langle w_i^{(t)}, x_n \rangle y_p}{\tau} \cdot \mathbf{1}_{\left|\langle w_i^{(t)}, x_n \rangle\right| \geq b_i^{(t)}} \cdot \mathbf{1}_{\left|\langle v_i^{(t)}, y_p \rangle\right| \geq b_i^{(t)}}$$

$$+ \sum_{y_n \in \mathfrak{N}} \frac{\langle f^{(t)}(x_p), h^{(t)}(y_n) \rangle \langle w_i^{(t)}, x_p \rangle y_n}{\tau} \cdot \mathbf{1}_{\left|\langle w_i^{(t)}, x_p \rangle\right| \geq b_i^{(t)}} \cdot \mathbf{1}_{\left|\langle v_i^{(t)}, y_n \rangle\right| \geq b_i^{(t)}}.$$

$$(31)$$

### B.4 Alignment Updates

We analyze how each neuron $i \in [m]$ aligns with the feature $\mathbf{M}_j$ during each iteration of SGD. The alignment can be described by the following update rule:

$$\langle w_i^{(t+1)}, \mathbf{M}_j \rangle = \langle w_i^{(t)}, \mathbf{M}_j \rangle - \langle \nabla_{w_i} L(f^{(t)}, h^{(t)}), \mathbf{M}_j \rangle$$
$$= \langle w_i^{(t)}, \mathbf{M}_j \rangle + \eta z_x^j z_y^j \langle v_i^{(t)}, \mathbf{H}_j \rangle + \eta z_x^j z_y^{j'} \langle v_i^{(t)}, \mathbf{H}_{j'} \rangle \pm Err_t. \tag{32}$$

Similarly, for $\langle v_i^{(t+1)}, \mathbf{H}_j \rangle$, the update rule becomes:

$$\langle v_i^{(t+1)}, \mathbf{H}_j \rangle = \langle v_i^{(t)}, \mathbf{H}_j \rangle - \langle \nabla_{v_i} L(f^{(t)}, h^{(t)}), \mathbf{H}_j \rangle$$
$$= \langle v_i^{(t)}, \mathbf{H}_j \rangle + \eta z_x^j z_y^j \langle w_i^{(t)}, \mathbf{M}_j \rangle + \eta z_x^j z_y^{j'} \langle w_i^{(t)}, \mathbf{M}_{j'} \rangle \pm Err_t. \tag{33}$$

Using Lemma C.6, we know that with high probability, $\sum_{x_n \in \mathfrak{N}} \frac{\langle f^{(t)}(x_n), h^{(t)}(y_p) \rangle}{\tau} \leq O(\frac{1}{d})$, so in Eq (30) the sum of second term and third term is always less than the first term, until $\langle f^{(t)}(x_n), h^{(t)}(y_p) \rangle = \Theta(d)$.

The updates for the components $\langle w_i^{(t+1)}, \mathbf{M}_j \rangle$, $\langle v_i^{(t+1)}, \mathbf{H}_j \rangle$, $\langle w_i^{(t+1)}, \mathbf{M}_{j'} \rangle$, and $\langle v_i^{(t+1)}, \mathbf{H}_{j'} \rangle$ (where $j'$ represents the spurious aligned feature corresponding to $j$) can be expressed concisely in matrix form as follows:

$$\begin{bmatrix} \langle w_i^{(t+1)}, \mathbf{M}_j \rangle \\ \langle v_i^{(t+1)}, \mathbf{H}_j \rangle \\ \langle w_i^{(t+1)}, \mathbf{M}_{j'} \rangle \\ \langle v_i^{(t+1)}, \mathbf{H}_{j'} \rangle \end{bmatrix} = \begin{bmatrix} a & b & 0 & c \\ b & a & c & 0 \\ 0 & c & a & b \\ c & 0 & b & a \end{bmatrix} \begin{bmatrix} \langle w_i^{(t)}, \mathbf{M}_j \rangle \\ \langle v_i^{(t)}, \mathbf{H}_j \rangle \\ \langle w_i^{(t)}, \mathbf{M}_{j'} \rangle \\ \langle v_i^{(t)}, \mathbf{H}_{j'} \rangle \end{bmatrix} \pm Err_t, \tag{34}$$

where the coefficients are defined as:

$$a = 1, \quad b = z_x^j z_y^j \cdot \mathbf{1}_{\left|\langle w_i^{(t)}, x_p \rangle\right| \geq b_i^{(t)}} \cdot \mathbf{1}_{\left|\langle v_i^{(t)}, y_p \rangle\right| \geq b_i^{(t)}},$$

$$c = z_x^j z_y^{j'} \cdot \mathbf{1}_{\left|\langle w_i^{(t)}, x_p \rangle\right| \geq b_i^{(t)}} \cdot \mathbf{1}_{\left|\langle v_i^{(t)}, y_p \rangle\right| \geq b_i^{(t)}}.$$

Therefore, we have

$$\langle w_i^{(t)}, \mathbf{M}_j \rangle = \langle v_i^{(t)}, \mathbf{H}_j \rangle = \frac{(a+b+c)^t + (a+b-c)^t}{4} \left( \langle w_i^{(0)}, \mathbf{M}_j \rangle + \langle v_i^{(0)}, \mathbf{H}_j \rangle \right)$$
$$+ \frac{(a+b+c)^t - (a+b-c)^t}{4} \left( \langle w_i^{(0)}, \mathbf{M}_{j'} \rangle + \langle v_i^{(0)}, \mathbf{H}_{j'} \rangle \right) \tag{35}$$

and

$$\langle w_i^{(t)}, \mathbf{M}_{j'} \rangle = \langle v_i^{(t)}, \mathbf{H}_{j'} \rangle = \frac{(a+b+c)^t + (a+b-c)^t}{4} \left( \langle w_i^{(0)}, \mathbf{M}_{j'} \rangle + \langle v_i^{(0)}, \mathbf{H}_{j'} \rangle \right)$$
$$+ \frac{(a+b+c)^t - (a+b-c)^t}{4} \left( \langle w_i^{(0)}, \mathbf{M}_j \rangle + \langle v_i^{(0)}, \mathbf{H}_j \rangle \right) \tag{36}$$

This matrix representation highlights the interactions between the alignment of true and spurious features during SGD updates. The diagonal elements $a$ dominate the contribution from existing alignments, while the off-diagonal terms $b, c$ capture the mutual influence between paired features and spurious alignments. Note that if $c$ is very small, it indicates that the spurious alignment ($j'$) has minimal influence, allowing $w_i$ to focus on learning purified features. Conversely, if $c$ is large, the spurious alignment could significantly interfere with the learning process, hindering the purification of features. The error term $Err_t$ accounts for higher-order noise or unmodeled effects in the update process.

Assuming a single spurious feature is a simplification for presentation that was made for ease of presentation in the proof and can be extended to a more general setting without altering the underlying insights. If each feature $j$ has $K-1$ spurious correlates, (34) becomes a $2K \times 2K$ matrix, and $N_i = j, j'$ in the last sentence of Theorem 4.2 contains $j$ and other $K-1$ features. Our analysis relies on the total spurious feature probability (bounded by $C_s$), not the number of correlated features, so **as long as the sum of all spurious feature probabilities is upper bounded by $C_s$, the core mechanism and insights of the theorem remain unchanged.**

## C  Technical Lemmas

**Definition C.1** (Neuron Characterization). *Let us define a few notations to characterize each neuron $w_i^{(t)}$'s behavior. For every constant $c_0 \in (0,1)$ and $\gamma \in (0, 0.1)$, by choosing $c_1 = 2 + 2(1-\gamma)c_0$ and $c_2 = \gamma c_0$, we define:*

*1. Let $\mathcal{S}_{j,sure}^{(t)} \subseteq [m]$ be those neurons $i \in [m]$ satisfying*

- $(\frac{1}{n} \sum_{i=1}^n \langle w_i^{(t)}, \mathbf{M}_j \rangle)^2 \geq \frac{(c_1+c_2) \log d}{d} \|\mathbf{M}\mathbf{M}^\top w_i^{(t)}\|_2^2$

- $(\frac{1}{n} \sum_{i=1}^n \langle w_i^{(t)}, \mathbf{M}_{j'} \rangle)^2 < \frac{(c_1-c_2) \log d}{d} \|\mathbf{M}\mathbf{M}^\top w_i^{(t)}\|_2^2$

*2. Let $\mathcal{S}_{j,pot}^{(t)} \subseteq [m]$ be those neurons $i \in [m]$ satisfying*

- $\langle w_i^{(t)}, \mathbf{M}_j \rangle^2 \geq \frac{(c_1-c_2) \log d}{d} \|\mathbf{M}\mathbf{M}^\top w_i^{(t)}\|_2^2$

**Lemma C.1** (Geometry at initialization). *We initialize the parameters by $w_i^{(0)} \sim \mathcal{N}(0, \sigma_0^2 \mathbf{I}_{d_1})$, where $\sigma_0^2 = \Theta\left(\frac{1}{d_1 poly(d)}\right)$. We have with probability $\geq 1 - o(1/d^3)$ over the random initialization, for all $j \in [d]$:*

$$\left| \mathcal{S}_{j,sure}^{(0)} \right| = \Omega\left( d^{\frac{\gamma}{4} c_0} \right) =: \Xi_1$$
$$\left| \mathcal{S}_{j,pot}^{(0)} \right| \leq O\left( d^{2\gamma c_0} \right) =: \Xi_2$$

*Proof.* If $g$ is standard Gaussian, then for every $t > 0$,

$$\frac{1}{\sqrt{2\pi}} \frac{(t)}{t^2+1} e^{-t^2/2} < \Pr_{g \sim \mathcal{N}(0,1)}[g > t] < \frac{1}{\sqrt{2\pi}} \frac{1}{(t)} e^{-t^2/2}. \tag{37}$$

We initialize the parameters by $w_i^{(0)} \sim \mathcal{N}(0, \sigma_0^2 \mathbf{I}_{d_1})$, where $\sigma_0^2 = \Theta\left(\frac{1}{d_1 \text{poly}(d)}\right)$. We have $\frac{1}{n}\sum_{i=1}^{n}\langle w_i^{(0)}, \mathbf{M}_i \rangle \sim \mathcal{N}\left(0, \frac{\sigma_0^2}{n}\right)$.

Therefore, for every $i \in m$ and $j \in d$, we have

$$
\begin{aligned}
p_1 = \Pr\left[\left(\frac{1}{n}\sum_{i=1}^{n}\langle w_i^{(0)}, \mathbf{M}_j \rangle\right)^2 \geq (c_1 + c_2)\frac{\sigma_0^2}{n}\log d\right] \\
= \Theta\left(\frac{1}{\log d}\right) \cdot \frac{1}{d^{(c_1+c_2)/2}} \\
= \Theta\left(\frac{1}{\sqrt{\log d}}\right) \cdot \frac{1}{d \cdot d^{(1-\gamma/2)c_0}}
\end{aligned}
\tag{38}
$$

$$
\begin{aligned}
p_2 = \Pr\left[\left(\frac{1}{n}\sum_{i=1}^{n}\langle w_i^{(0)}, \mathbf{M}_{j'} \rangle\right)^2 \geq (c_1 - c_2)\frac{\sigma_0^2}{n}\log d\right] \\
= \Theta\left(\frac{1}{\log d}\right) \cdot \frac{1}{d^{(c_1-c_2)/2}} \\
= \Theta\left(\frac{1}{\sqrt{\log d}}\right) \cdot \frac{1}{d \cdot d^{(1-3\gamma/2)c_0}}
\end{aligned}
\tag{39}
$$

Let $\mathcal{S}_{j,\text{sure}}^{(0)} \subseteq [m]$ be those neurons $i \in [m]$ satisfying

- $\left(\frac{1}{n}\sum_{i=1}^{n}\langle w_i^{(0)}, \mathbf{M}_j \rangle\right)^2 \geq \frac{(c_1+c_2)\log d}{d}\|\mathbf{MM}^\top w_i^{(0)}\|_2^2$
- $\left(\frac{1}{n}\sum_{i=1}^{n}\langle w_i^{(0)}, \mathbf{M}_{j'} \rangle\right)^2 < \frac{(c_1-c_2)\log d}{d}\|\mathbf{MM}^\top w_i^{(0)}\|_2^2$

By concentration with respect to all $m$ choices of $i \in [m]$, we know with probability at least $1 - o\left(\frac{1}{d^3}\right)$ it satisfies $\left|\mathcal{S}_{j,\text{sure}}^{(0)}\right| = \Omega\left(d^{\frac{\gamma}{4}c_0}\right)$.

Let $\mathcal{S}_{j,\text{pot}}^{(0)} \subseteq [m]$ be those neurons $i \in [m]$ satisfying

- $\langle w_i^{(0)}, \mathbf{M}_j \rangle^2 \geq \frac{(c_1-c_2)\log d}{d}\|\mathbf{MM}^\top w_i^{(0)}\|_2^2$

By concentration with respect to all $m$ choices of $i \in [m]$, we know with probability at least $1 - o\left(\frac{1}{d^3}\right)$ it satisfies $\left|\mathcal{S}_{j,\text{pot}}^{(0)}\right| = O\left(d^{2\gamma c_0}\right)$.

More details of the proof can be found in Lemma B.2 of [2]. $\qquad\square$

**Lemma C.2.** *With high probability* $1 - \frac{1}{poly(d)}$*, for every* $i \in [m]$*, the following holds:*

$$
\Pr\left[\left(\frac{1}{2n}\sum_{i=1}^{n}\langle w_i^{(0)}, \mathbf{M}_j \rangle - \langle w_i^{(0)}, \mathbf{M}_{j'} \rangle\right)^2 \geq \frac{1}{d}\frac{\sigma_0^2}{2n}\log d\right] \geq 1 - O\left(\frac{1}{\sqrt{d}}\right)
\tag{40}
$$

**Lemma C.3.** *With high probability* $1 - \frac{1}{poly(d)}$*, for every* $i \in [m]$*, the following holds:*

$$
\|\mathbf{MM}^\top w_i^{(0)}\|_2^2 + \|\mathbf{HH}^\top v_i^{(0)}\|_2^2 \in 2d\sigma_0^2\left[1 - \tilde{O}\left(\frac{1}{\sqrt{d}}\right), 1 + \tilde{O}\left(\frac{1}{\sqrt{d}}\right)\right].
\tag{41}
$$

*Proof.* Let $X \sim \chi_n^2$. By standard properties of the chi-squared distribution, we know that with probability at least $1 - \delta$,

$$
|X - n| \leq 2\sqrt{n\log(1/\delta)}.
\tag{42}
$$

In our case, we consider $\frac{\|\mathbf{M}\mathbf{M}^\top w_i^{(0)}\|_2^2 + \|\mathbf{H}\mathbf{H}^\top v_i^{(0)}\|_2^2}{\sigma_0^2} \sim \chi_{2d}^2$. Setting $\delta = \frac{1}{\text{poly}(d)}$, we have $n = 2d$, and thus, with high probability $1 - \frac{1}{\text{poly}(d)}$, the following holds:

$$\left| \frac{\|\mathbf{M}\mathbf{M}^\top w_i^{(0)}\|_2^2 + \|\mathbf{H}\mathbf{H}^\top v_i^{(0)}\|_2^2}{\sigma_0^2} - 2d \right| \leq 2\sqrt{2d \log(\text{poly}(d))}. \tag{43}$$

Rearranging and incorporating the scaling factor $\sigma_0^2$, we get:

$$\|\mathbf{M}\mathbf{M}^\top w_i^{(0)}\|_2^2 + \|\mathbf{H}\mathbf{H}^\top v_i^{(0)}\|_2^2 \in 2d\sigma_0^2 \left[ 1 - \widetilde{O}\left(\frac{1}{\sqrt{d}}\right), 1 + \widetilde{O}\left(\frac{1}{\sqrt{d}}\right) \right]. \tag{44}$$

$\square$

**Lemma C.4** (Noise Projection Bound). *For the spurious dense noise $\xi_{x_p} \sim \mathcal{N}(0, \sigma_\xi^2 \mathbf{I}_{d_1})$, where the variance satisfies $\omega\left(\frac{1}{d_1}\right) \leq \sigma_\xi^2 \leq O\left(\frac{1}{d}\right)$, the following holds with high probability $1 - e^{-\Omega(d_1)}$:*

$$|\langle w_i, \xi \rangle|^2 \leq O\left(\frac{\|w_i\|_2^2}{d^{1+c_0}}\right), \quad \forall i \in [m]. \tag{45}$$

*Proof.* For all $j \in [d_1]$, by the properties of the Gaussian distribution, we have:

$$\Pr_\xi \left[ \langle \mathbf{M}_j, \xi \rangle^2 \leq O\left(\frac{1}{d^{1+c_0}}\right) \right] \geq 1 - e^{-\Omega(d_1)}. \tag{46}$$

Now, consider the term $|\langle w_i, \xi \rangle|^2$. We decompose it as:

$$|\langle w_i, \xi \rangle|^2 = \sum_{j \in [d]} |\langle w_i, \mathbf{M}_j \rangle|^2 \cdot |\langle \mathbf{M}_j, \xi \rangle|^2 + \sum_{j \in [d_1] \setminus [d]} |\langle w_i, \mathbf{M}_j^\perp \rangle|^2 \cdot |\langle \mathbf{M}_j^\perp, \xi \rangle|^2. \tag{47}$$

For the first term, since $|\langle \mathbf{M}_j, \xi \rangle|^2 \leq O\left(\frac{1}{d^{1+c_0}}\right)$ with high probability, we have:

$$\sum_{j \in [d]} |\langle w_i, \mathbf{M}_j \rangle|^2 \cdot |\langle \mathbf{M}_j, \xi \rangle|^2 \leq \sum_{j \in [d]} O\left(\frac{|\langle w_i, \mathbf{M}_j \rangle|^2}{d^{1+c_0}}\right). \tag{48}$$

Similarly, for the second term:

$$\sum_{j \in [d_1] \setminus [d]} |\langle w_i, \mathbf{M}_j^\perp \rangle|^2 \cdot |\langle \mathbf{M}_j^\perp, \xi \rangle|^2 \leq \sum_{j \in [d_1] \setminus [d]} O\left(\frac{|\langle w_i, \mathbf{M}_j^\perp \rangle|^2}{d^{1+c_0}}\right). \tag{49}$$

Combining these, we have:

$$|\langle w_i, \xi \rangle|^2 \leq O\left(\frac{\|\mathbf{M}\mathbf{M}^\top w_i\|_2^2}{d^{1+c_0}} + \frac{\|\mathbf{M}^\perp \mathbf{M}^{\perp \top} w_i\|_2^2}{d^{1+c_0}}\right). \tag{50}$$

Since $\|\mathbf{M}\mathbf{M}^\top w_i\|_2^2 + \|\mathbf{M}^\perp \mathbf{M}^{\perp \top} w_i\|_2^2 = \|w_i\|_2^2$, we conclude:

$$|\langle w_i, \xi \rangle|^2 \leq O\left(\frac{\|w_i\|_2^2}{d^{1+c_0}}\right). \tag{51}$$

Thus, the lemma holds. $\square$

**Lemma C.5** (Tail Bound for Matrix Product). *Let $\mathbf{Q} \in \mathbb{R}^{n \times n}$ be a symmetric matrix, and let $w, v$ be independent zero-mean Gaussian random vectors with covariance matrix $\mathbf{I}_n$. Define*

$$Z := \sum_{i,j=1}^n Q_{ij} w_i v_j. \tag{52}$$

*Then, for any $\delta > 0$, the following tail bound holds:*

$$\Pr[|Z| \geq \delta] \leq 4 \exp\left(-\frac{\delta^2}{4\|\mathbf{Q}\|_F^2 + 4\delta\|\mathbf{Q}\|_{op}}\right). \tag{53}$$

**Lemma C.6** (Bound Inner Product). *Consider the inner product between the feature vectors at initialization:*

$$\langle f(x), h(y) \rangle = \langle \mathbf{W}x, \mathbf{V}y \rangle = \sum_{l=1}^{m} w_l^\top x y^\top v_l = \sum_{l=1}^{m} \sum_{i,j=1}^{d_1} (x_i^\top y_j) w_l^\top v_l. \tag{54}$$

*Here, using Lemma C.5, $\mathbf{Q} = xy^\top$, with $\|\mathbf{Q}\|_{op} = \Theta(1)$, $\|\mathbf{Q}\|_F = \Theta(1)$ and $\sigma_0^2 = \Theta\left(\frac{1}{d_1 poly(d)}\right)$. Then, at initialization ($t = 0$), the following holds:*

$$\Pr[|\langle f^{(t)}(x), h^{(t)}(y) \rangle| \geq \Omega(1)] \leq e^{-poly(d)}, \tag{55}$$

**Lemma C.7** (Concentration bound for empirical loss and gradients). *There exist $N \geq poly(d)$ for some sufficiently large polynomial and all $\|w_i\|_2 \leq O(d)$, $i \in [m]$, it satisfies*

$$\left| \frac{1}{N} \sum_{p \in [N]} L(f^{(t)}, h^{(t)}; (x_p, y_p)) - \mathbb{E}_{(x_p, y_p) \in D}[L(f^{(t)}, h^{(t)}; (x_p, y_p))] \right| \leq O(\frac{1}{d}) \tag{56}$$

$$\left\| \frac{1}{N} \sum_{p \in [N]} \nabla_{w_i} L(f^{(t)}, h^{(t)}; (x_p, y_p)) - \mathbb{E}_{(x_p, y_p) \in D}[\nabla_{w_i} L(f^{(t)}, h^{(t)}; (x_p, y_p))] \right\|_2 \leq O(\frac{1}{d}) \tag{57}$$

*Proof.* The proof can be done by trivial VC dimension or Rademacher complexity arguments similarly to Lemma A.2. [2]. $\qquad\square$

**Lemma C.8** (Misalignment Probability Bound). *The probability of spurious alignment satisfies:*

$$\frac{\log\left(\frac{1}{2\gamma c_0}\right)}{2\log\frac{d_1}{d}} < \Pr(|z_{y_p}^j| = 1 \mid |z_{x_p}^{j'}| = 1) < \frac{1}{2}. \tag{58}$$

*Proof.* By concentration over all $m$ choices of $i \in [m]$, we find that with probability at least $1 - o\left(\frac{1}{d^3}\right)$, the number of neurons satisfying:

$$\left( \frac{1}{n} \sum_{i=1}^{n} \langle w_i, \mathbf{M}_j \rangle \right)^2 < (c_1 + 4c_2) \frac{\sigma_0^2}{n} \log d \tag{59}$$

is $o(1)$.

In addition, for all neurons, we have:

$$\max\left( \langle w_i^{(T_1)}, \mathbf{M}_{j'} \rangle^2 \right) \leq \frac{c_1 + 3c_2}{2} \frac{\log d}{d} \cdot \frac{\|w_i^{(T_1)}\|_2^2 + \|v_i^{(T_1)}\|_2^2}{2}. \tag{60}$$

Define:

$$\Delta^{(T_1)} = \frac{(a+b-c)^{T_1}}{4} \left| \langle w_i^{(0)}, \mathbf{M}_j \rangle + \langle v_i^{(0)}, \mathbf{H}_j \rangle - \langle w_i^{(0)}, \mathbf{M}_{j'} \rangle - \langle v_i^{(0)}, \mathbf{H}_{j'} \rangle \right|. \tag{61}$$

Thus:

$$\langle w_i^{(T_1)}, \mathbf{M}_{j'} \rangle^2 = \left| \max\left( \langle w_i^{(T_1)}, \mathbf{M}_{j'} \rangle \right) - \Delta^{(T_1)} \right|^2 \geq \frac{c_1 - c_2}{2} \frac{\log d}{d} \cdot \frac{\|w_i^{(T_1)}\|_2^2 + \|v_i^{(T_1)}\|_2^2}{2}. \tag{62}$$

We begin by expressing $a + b - c$ and $a + b + c$ as functions of $P_1 = \Pr(|z_{y_p}^j| = 1 \mid |z_{x_p}^{j'}| = 1)$ and $P_2 = \Pr(|z_{y_p}^j| = 1 \mid |z_{x_p}^j| = 1)$, where $P_1 + P_2 = 1$:

$$a + b - c = 1 - \eta\lambda + \eta \frac{(P_1 - P_2)C_z \log\log d}{d}, \tag{63}$$

$$a + b + c = 1 - \eta\lambda + \eta\frac{(P_1 + P_2)C_z \log\log d}{d}. \tag{64}$$

Using Eq (62), Eq (35) and Eq (36), we derive:

$$\frac{(a+b-c)^{2T_1}}{(a+b+c)^{2T_1}} \leq \left(\sqrt{\frac{c_1 + 3c_2}{2}} - \sqrt{\frac{c_1 - c_2}{2}}\right)^2 \leq 2c_2^2. \tag{65}$$

Substituting back, we find:

$$\frac{\log\left(\frac{1}{2\gamma c_0}\right)}{2\log\frac{d_1}{d}} < P_1 < \frac{1}{2}. \tag{66}$$

For example, setting $c_0 = 0.1$, $\gamma = 0.005$, $d = 100$, and $d_1 = 10000$, we calculate:

$$\frac{1}{4} \leq \Pr(|z_{y_p}^j| = 1 \mid |z_{x_p}^{j'}| = 1) < \frac{1}{2}. \tag{67}$$

This concludes the proof by bounding $\Pr(|z_{y_p}^j| = 1 \mid |z_{x_p}^{j'}| = 1)$ under the given conditions. $\qquad\square$

## D  ITCP on Raw Data I

In this section we analyze Phase I of ITCP on Raw Data as the training iterations $t \leq T_1$, where $T_1 = \Theta\left(\frac{d\log d}{\eta}\right)$ is the iteration when all $\frac{\|w_i^{(T_1)}\|_2^2 + \|v_i^{(T_1)}\|_2^2}{2} \geq \|w_i^{(0)}\|_2^2 + \|v_i^{(0)}\|_2^2$. When $t \leq T_1$, we set $b_i^{(t)} = 0$. For every neuron $i \in [m]$, the weights $w_i$ and $v_i$ exhibit an increase in alignment along the direction of informative features $\mathbf{M}$ and $\mathbf{H}$, while showing negligible increase in alignment along the direction of noise features $\mathbf{M}^\perp$ and $\mathbf{H}^\perp$.

Based on subsection B.2, we have $\Pr(|z_{y_p}^j| = 1 \mid |z_{x_p}^{j'}| = 1) = \Theta(1)$, so $\mathbb{E}\left[z_x^j z_y^j\right]$ and $\mathbb{E}\left[z_x^j z_y^{j'}\right]$ both in $\Theta\left(\frac{1}{d}\right)$. In this case, $w_i^{(t+1)}$ is jointly influenced by $\mathbf{M}_j$ and $\mathbf{M}_{j'}$, with both features contributing comparably to the updates.

To simplify our analysis, we consider the worse case where $\Pr(|z_{y_p}^{j'}| = 1 \mid |z_{x_p}^j| = 1) = \Pr(|z_{y_p}^j| = 1 \mid |z_{x_p}^j| = 1) = \frac{1}{2}$ such that $\mathbb{E}\left[z_x^j z_y^j\right] = \mathbb{E}\left[z_x^j z_y^{j'}\right] = \frac{C_z}{2d}$, so using Eq (35), Eq (36) and $b_i^{(t)} = 0$, we have

$$\langle w_i^{(t)}, \mathbf{M}_j\rangle = \frac{(a+b+c)^t}{4}\left(\langle w_i^{(0)}, \mathbf{M}_j\rangle + \langle v_i^{(0)}, \mathbf{H}_j\rangle + \langle w_i^{(0)}, \mathbf{M}_{j'}\rangle + \langle v_i^{(0)}, \mathbf{H}_{j'}\rangle\right) \tag{68}$$

$$\langle w_i^{(t)}, \mathbf{M}_{j'}\rangle = \frac{(a+b+c)^t}{4}\left(\langle w_i^{(0)}, \mathbf{M}_j\rangle + \langle v_i^{(0)}, \mathbf{H}_j\rangle + \langle w_i^{(0)}, \mathbf{M}_{j'}\rangle + \langle v_i^{(0)}, \mathbf{H}_{j'}\rangle\right) \tag{69}$$

This represents the worst-case scenario as the contributions of the aligned feature $\mathbb{E}\left[z_x^j z_y^j\right]$ and the spurious feature $\mathbb{E}\left[z_x^j z_y^{j'}\right]$ are identical. Under real circumstances, we expect $\mathbb{E}\left[z_x^j z_y^j\right] < \mathbb{E}\left[z_x^j z_y^{j'}\right]$, which would result in $\langle w_i^{(t+1)}, \mathbf{M}_j\rangle > \langle w_i^{(t+1)}, \mathbf{M}_{j'}\rangle$. However, in this worst-case scenario, the equality of contributions prevents the network from prioritizing purified features, resulting in equal magnitudes for $\langle w_i^{(t+1)}, \mathbf{M}_j\rangle$ and $\langle w_i^{(t+1)}, \mathbf{M}_{j'}\rangle$, thereby hindering effective feature separation.

We first provide a lower bound for $\|\mathbf{MM}^\top w_i^{(t)}\|_2^2$ for iterations $t \leq t_1$. From Eq (122) and Eq (69) we have:

$$\|\mathbf{MM}^\top w_i^{(t)}\|_2^2 = \sum_{i=1}^d \left[\frac{(a+b+c)^t}{4}\left(\langle w_i^{(0)}, \mathbf{M}_j\rangle + \langle v_i^{(0)}, \mathbf{H}_j\rangle\right) + \frac{(a+b+c)^t}{4}\left(\langle w_i^{(0)}, \mathbf{M}_{j'}\rangle + \langle v_i^0, \mathbf{H}_{j'}\rangle\right)\right]^2$$
$$= \left(1 + \frac{\eta C_z}{d}\right)^{2t}\frac{\|\mathbf{MM}^\top w_i^{(0)}\|_2^2 + \|\mathbf{HH}^\top v_i^{(0)}\|_2^2}{8} \tag{70}$$

$$\|\mathbf{M}^{\perp}(\mathbf{M}^{\perp})^{\top}w_i^{(t)}\|_2^2 \le \left(1 + \frac{1}{\text{poly}(d)}\right)\|\mathbf{M}^{\perp}(\mathbf{M}^{\perp})^{\top}w_i^{(0)}\|_2^2. \tag{71}$$

The detailed proof of Eq (71) can be found in Hypothesis C.4 of [51].

A similar result holds for $\|\mathbf{H}\mathbf{H}^{\top}v_i^{(t)}\|_2^2$ and $\|\mathbf{H}^{\perp}(\mathbf{H}^{\perp})^{\top}v_i^{(t)}\|_2^2$.

Eq (70) and Eq (71) shows that the image and text dictionary features $\mathbf{M}, \mathbf{H}$ can grow exponentially, while the noisy features $\mathbf{M}^{\perp}, \mathbf{H}^{\perp}$ remain almost unchanged when $t \le T_1$.

For $\mathbf{M}_j^{\perp}$ where $j \in [d_1] \setminus [d]$, using Eq (71), we obtain:

$$|\langle w_i^{(t+1)}, \mathbf{M}_j^{\perp}\rangle|^2 \le O\left(\frac{1}{d_1}\right)\|w_i^{(0)}\|_2^2 \le O\left(\frac{1}{d_1}\right) \cdot \frac{\|w_i^{(T_1)}\|_2^2 + \|v_i^{(T_1)}\|_2^2}{2}. \tag{72}$$

This result demonstrates that the noisy features $\mathbf{M}_j^{\perp}$ experience nearly no increase during this phase, remaining insignificant in their contribution to the alignment of $w_i$.

## D.1  Lower Bound of Alignment for $i \in S_{j,\text{sure}}$

This section provides a analysis of the alignment growth for neurons $i \in S_{j,\text{sure}}$. Specifically, we demonstrate that for every $j \in [d]$, if $i \in S_{j,\text{sure}}$, the alignment $\langle \mathbf{M}_j, w_i^{(t)}\rangle^2$ and its spurious alignment $\langle \mathbf{M}_j', w_i^{(t)}\rangle^2$ increase exponentially when $t \le T_1$.

We now prove the lower bound of $|\langle w_i^{(T_1)}, \mathbf{M}_j\rangle|^2$ for $i \in S_{j,\text{sure}}$:

$$
\begin{aligned}
|\langle w_i^{(T_1)}, \mathbf{M}_j\rangle|^2 &= \left(1 + \eta\frac{C_z}{d}\right)^{2T_1}\left(\frac{\langle w_i^{(0)}, \mathbf{M}_j\rangle + \langle v_i^{(0)}, \mathbf{H}_{j'}\rangle + \langle w_i^{(0)}, \mathbf{M_{j'}}\rangle + \langle v_i^{(0)}, \mathbf{H}_j\rangle}{4}\right)^2 \\
&\overset{\diamondsuit}{\ge} \left(1 + \eta\frac{C_z}{d}\right)^{2T_1} \cdot \frac{(c_1 + c_2)\log d}{d} \cdot \frac{\|\mathbf{M}\mathbf{M}^{\top}w_i^{(0)}\|_2^2 + \|\mathbf{H}\mathbf{H}^{\top}v_i^{(0)}\|_2^2}{8} \\
&\overset{\heartsuit}{=} \frac{(c_1 + c_2)\log d}{d} \cdot \frac{\|\mathbf{M}\mathbf{M}^{\top}w_i^{(T_1)}\|_2^2 + \|\mathbf{H}\mathbf{H}^{\top}v_i^{(T_1)}\|_2^2}{2} \\
&\overset{\clubsuit}{\ge} \frac{(c_1 + c_2)\log d}{d} \cdot \frac{\|w_i^{(T_1)}\|_2^2 + \|v_i^{(T_1)}\|_2^2 - \|w_i^{(0)}\|_2^2 - \|v_i^{(0)}\|_2^2}{2} \\
&\overset{\spadesuit}{>} \frac{(1 + c_0 - \gamma c_0)\log d}{d} \cdot \frac{\|w_i^{(T_1)}\|_2^2 + \|v_i^{(T_1)}\|_2^2}{2}
\end{aligned}
\tag{73}
$$

In $\diamondsuit$ we use Definition C.1. In $\heartsuit$ we use Eq (70). In $\clubsuit$ we use $\frac{\|w_i^{(T_1)}\|_2^2 + \|v_i^{(T_1)}\|_2^2}{2} \ge \|w_i^{(0)}\|_2^2 + \|v_i^{(0)}\|_2^2$. In $\spadesuit$ we use $c_1 + c_2 > 2(1 + c_0 - \gamma c_0)$.

Similarly, $|\langle w_i^{(T_1)}, \mathbf{M}_{j'}\rangle|^2$ have the same lower bound.

## D.2  Upper Bound of Alignment for $i \notin S_{j,\text{pot}}$

In this subsection, we analyze the alignment of neuron $i \notin S_{j,\text{pot}}$ with the feature $\mathbf{M}_j$ and provide an upper bound for $|\langle w_i^{(T_1)}, \mathbf{M}_j\rangle|^2$. While neurons $i \notin S_{j,\text{pot}}$ still exhibit exponential growth in their alignment, their weaker initialization results in significantly smaller alignment compared to neurons in $S_{j,\text{sure}}$, limiting their contribution to learning the feature $\mathbf{M}_j$.

To establish the bound, we begin with the following expression:

$$|\langle w_i^{(T_1)}, \mathbf{M}_j \rangle|^2 = \left(1 + \eta \frac{C_z}{d}\right)^{2T_1} \left(\frac{\langle w_i^{(0)}, \mathbf{M}_j \rangle + \langle v_i^{(0)}, \mathbf{H}_{j'} \rangle + \langle w_i^{(0)}, \mathbf{M}_{\mathbf{j'}} \rangle + \langle v_i^{(0)}, \mathbf{H}_j \rangle}{4}\right)^2$$

$$\overset{\diamond}{\leq} \left(1 + \eta \frac{C_z}{d}\right)^{2T_1} \cdot \frac{(c_1 - c_2) \log d}{d} \cdot \frac{\|\mathbf{M}\mathbf{M}^\top w_i^{(0)}\|_2^2 + \|\mathbf{H}\mathbf{H}^\top v_i^{(0)}\|_2^2}{8}$$

$$= \frac{(c_1 - c_2) \log d}{d} \cdot \frac{\|\mathbf{M}\mathbf{M}^\top w_i^{(T_1)}\|_2^2 + \|\mathbf{H}\mathbf{H}^\top v_i^{(T_1)}\|_2^2}{2}. \tag{74}$$

Here, in $\diamond$, we use Lemma C.1, which captures the reduced alignment for neurons outside $S_{j,\text{pot}}$.

Similar to the analysis for $i \in S_{j,\text{sure}}$, the alignment strength for $i \notin S_{j,\text{pot}}$ is weaker, as $c_1 - c_2$ is less than $2(1 + c_0 - \gamma c_0)$, leading to:

$$|\langle w_i^{(T_1)}, \mathbf{M}_j \rangle|^2 < \frac{(1 + c_0 - 3\gamma c_0) \log d}{d} \cdot \frac{\|w_i^{(T_1)}\|_2^2 + \|v_i^{(T_1)}\|_2^2}{2}. \tag{75}$$

This inequality highlights the slower alignment for neurons outside $S_{j,\text{pot}}$, distinguishing their behavior from neurons in $S_{j,\text{sure}}$. Consequently, $i \notin S_{j,\text{pot}}$ contributes less significantly to the alignment of $\mathbf{M}_j$, reinforcing the importance of initial affinity for effective alignment.

### D.3 Summary

At this stage ($t \leq T_1$), we do not consider the worst-case scenario where the probability bounds for feature coupling satisfy

$$\frac{\log(1/c_0)}{2 \log d} < \Pr(|z_{y_p}^{j'}| = 1 \mid |z_{x_p}^j| = 1) < \frac{1}{2} < \Pr(|z_{y_p}^j| = 1 \mid |z_{x_p}^j| = 1) < 1$$

(as assumed in SubSection B.2). Thus, we summarize the results when $t \leq T_1$ as follows:

1. For $i \in S_{j,\text{sure}}$, the alignment strength satisfies:

$$|\langle w_i^{(T_1)}, \mathbf{M}_j \rangle|^2 > |\langle w_i^{(T_1)}, \mathbf{M}_{j'} \rangle|^2 > \frac{(1 + c_0 - \gamma c_0) \log d}{d} \cdot \frac{\|w_i^{(T_1)}\|_2^2 + \|v_i^{(T_1)}\|_2^2}{2}, \tag{76}$$

where $j'$ represents the corresponding spurious alignment feature.

2. For $i \notin S_{j,\text{pot}}$, the alignment strength satisfies:

$$|\langle w_i^{(T_1)}, \mathbf{M}_j \rangle|^2 < \frac{(1 + c_0 - 3\gamma c_0) \log d}{d} \cdot \frac{\|w_i^{(T_1)}\|_2^2 + \|v_i^{(T_1)}\|_2^2}{2}. \tag{77}$$

3. For $\mathbf{M}_j^\perp$ where $j \in [d_1] \setminus [d]$, we have:

$$|\langle w_i^{(t+1)}, \mathbf{M}_j^\perp \rangle|^2 < O\left(\frac{1}{d_1}\right) \cdot \frac{\|w_i^{(T_1)}\|_2^2 + \|v_i^{(T_1)}\|_2^2}{2}. \tag{78}$$

These results demonstrate that when $t \leq T_1$, all features in $\mathbf{M}$ increase, but the alignment for $i \in S_{j,\text{sure}}$, including the corresponding spurious alignment, grows significantly larger due to favorable initialization. In contrast, noisy features $\mathbf{M}^\perp$ remain unchanged.

## E  ITCP on Raw Data II

The Phase II of ITCP on Raw Data is defined as the training iterations $T_1 < t \leq T_2$, where $T_2 - T_1 = \Theta\left(\frac{d \log d}{\eta}\right)$.

At the beginning of this phase, we set the bias threshold as:

$$b_i^{(T_1)} = \sqrt{\frac{(1 + c_0 - 2\gamma c_0) \log d}{d} \cdot \frac{\|w_i^{(T_1)}\|_2^2 + \|v_i^{(T_1)}\|_2^2}{2}}. \tag{79}$$

During training, the bias threshold is iteratively updated as:

$$b_i^{(t+1)} = \left(1 + \frac{\eta}{d}\right) b_i^{(t)}, \tag{80}$$

until all neurons satisfy:

$$\|w_i^{(T_2)}\|_2^2 \geq \Omega(d)\|w_i^{(T_1)}\|_2^2. \tag{81}$$

In this phase, the dynamics of alignment vary depending on whether a neuron belongs to $S_{j,\text{sure}}$ or not:

- For $i \notin S_{j,\text{pot}}$: The weights $w_i$ and $v_i$ show negligible alignment growth with both the informative features $\mathbf{M}_j$, $\mathbf{H}_j$ and the noise features $\mathbf{M}^\perp$, $\mathbf{H}^\perp$. This is due to their weaker initialization, as shown in Phase I, and the effect of the indicator function when $t \geq T_1$ which prevents them from being activated. As a result, their capacity to learn meaningful alignments during this phase is significantly limited.

- For $i \in S_{j,\text{sure}}$: The weights $w_i$ and $v_i$ exhibit continued alignment growth with the informative features $\mathbf{M}_j$, $\mathbf{H}_j$. Additionally, their alignment with the corresponding spurious features $\mathbf{M}_{j'}$, $\mathbf{H}_{j'}$ also increases due to their strong initialization, as shown in Phase I, and the effect of the indicator function when $t \geq T_1$, which ensures they are always activated.

By the end of this stage ($t = T_2$), the weights $w_i$, $v_i$ will predominantly focus on the features $\mathbf{M}_j$, $\mathbf{H}_j$ if $i \in S_{j,\text{sure}}$, while largely ignoring the features $\mathbf{M}_j$, $\mathbf{H}_j$ if $i \notin S_{j,\text{pot}}$, as well as the noise features $\mathbf{M}^\perp$, $\mathbf{H}^\perp$. This separation lays the foundation for the Phase II of ITCP on Raw Data, where spurious alignments are expected to further diminish due to the dominance of true feature alignments.

Similarly to the proof of $t \leq T_1$ To simplify our analysis, we still consider the worse case where $\Pr(|z_{y_p}^{j'}| = 1 \mid |z_{x_p}^j| = 1) = \Pr(|z_{y_p}^j| = 1 \mid |z_{x_p}^j| = 1) = \frac{1}{2}$ such that $\mathbb{E}\left[z_x^j z_y^j\right] = \mathbb{E}\left[z_x^j z_y^{j'}\right] = \frac{C_z}{2d}$.

### E.1   Alignment for $i \in S_{j,\text{sure}}$

This section provides a analysis of the alignment growth for neurons $i \in S_{j,\text{sure}}$. Specifically, we demonstrate that for every $j \in [d]$, if $i \in S_{j,\text{sure}}$, the alignment $\langle \mathbf{M}_j, w_i^{(t)} \rangle^2$ and its spurious alignment $\langle \mathbf{M}_j', w_i^{(t)} \rangle^2$ increase exponentially when $T_1 < t \leq T_2$.

For $i \in S_{j,\text{sure}}$, using Lemma C.4, the following holds with high probability $1 - e^{-\Omega(d_1)}$ when $T_1 < t \leq T_2$ :

$$\left|\langle w_i^{(t)}, \xi \rangle\right|^2 \leq O\left(\frac{\left\|w_i^{(t)}\right\|_2^2}{d^{1+c_0}}\right) < b_i^{(t)} \tag{82}$$

Therefore, with high probability $1 - e^{-\Omega(d_1)}$, using Eq (76) and Eq (79) the indicator function satisfies the condition when $t = T_1$:

$$\mathbf{1}_{\left|\left\langle w_i^{(t)}, x_p \right\rangle\right| \geq b_i^{(t)}} \cdot \mathbf{1}_{\left|\left\langle v_i^{(t)}, y_p \right\rangle\right| \geq b_i^{(t)}} = 1, \tag{83}$$

we can ensure that:

$$\mathbb{E}\left[z_x^j z_y^j \cdot \mathbf{1}_{\left|\left\langle w_i^{(t)}, x_p \right\rangle\right| \geq b_i^{(t)}} \cdot \mathbf{1}_{\left|\left\langle v_i^{(t)}, y_p \right\rangle\right| \geq b_i^{(t)}}\right] = \frac{C_z}{d}. \tag{84}$$

Using Eq (116) we know that $\left(1 + \eta \frac{C_z}{2d}\right) > \left(1 + \frac{\eta}{d}\right)$ and using Eq (34) we have

$$|\langle w_i^{(t+1)}, \mathbf{M}_j \rangle| > (1 + \frac{\eta}{d}) b_i^{(t)} = b_i^{(t+1)}. \tag{85}$$

This implies that when $t > T_1$, the alignment strength of informative features surpasses the updated bias threshold $b_i^{(t)}$. Consequently, the indicator functions become consistently activated $T_1 < t \leq T_2$ such that

$$\mathbf{1}_{\left|\left\langle w_i^{(t)}, x_p \right\rangle\right| \geq b_i^{(t)}} \cdot \mathbf{1}_{\left|\left\langle v_i^{(t)}, y_p \right\rangle\right| \geq b_i^{(t)}} = 1, \tag{86}$$

Using Eq (34), the weight dynamics for $|\langle w_i^{(t+1)}, \mathbf{M}_j \rangle|$ can be expressed as when $T_1 < t \le T_2$:

$$|\langle w_i^{(t+1)}, \mathbf{M}_j \rangle| = \left(1 + \eta\frac{C_z}{d}\right)\left(\frac{\langle w_i^{(t)}, \mathbf{M}_j \rangle + \langle v_i^{(t)}, \mathbf{H}_{j'} \rangle + \langle w_i^{(t)}, \mathbf{M}_{\mathbf{j}}^{\perp} \rangle + \langle v_i^{(t)}, \mathbf{H}_j \rangle}{4}\right). \quad (87)$$

Similarly, $|\langle w_i^{(T_1)}, \mathbf{M}_{j'} \rangle|^2$ have the same result.

## E.2  Alignment for $i \notin S_{j,\textbf{pot}}$

In this section, we analyze the alignment behavior for neurons $i \notin S_{j,\text{pot}}$. Specifically, we demonstrate that for every $j \in [d]$, if $i \notin S_{j,\text{pot}}$, the alignment $\langle \mathbf{M}_j, w_i^{(t)} \rangle^2$ exhibits negligible growth during the interval $T_1 < t \le T_2$.

For $i \notin S_{j,\text{pot}}$, using Eq (156), Eq (79) and Eq (76), we have with high probability $1 - e^{-\Omega(d_1)}$, similarly to the proof of $i \in S_{j,\text{sure}}$, the indicator function satisfies the condition when $t = T_1$:

$$\mathbf{1}_{\left|\langle w_i^{(t)}, x_p \rangle\right| \ge b_i^{(t)}} \cdot \mathbf{1}_{\left|\langle v_i^{(t)}, y_p \rangle\right| \ge b_i^{(t)}} = 0, \quad (88)$$

We can ensure that:

$$\mathbb{E}\left[z_x^j z_y^j \cdot \mathbf{1}_{\left|\langle w_i^{(t)}, x_p \rangle\right| \ge b_i^{(t)}} \cdot \mathbf{1}_{\left|\langle v_i^{(t)}, y_p \rangle\right| \ge b_i^{(t)}}\right] \le o\left(\frac{1}{d^2}\right). \quad (89)$$

Using Eq (116) we know that $\left(1 + o(\frac{\eta}{d^2})\right) < \left(1 + \frac{\eta}{d}\right)$ and using Eq (34) we have

$$|\langle w_i^{(t+1)}, \mathbf{M}_j \rangle| < (1 + \frac{\eta}{d})b_i^{(t)} = b_i^{(t+1)}. \quad (90)$$

This implies that when $t > T_1$, the alignment strength of informative features does not surpass the updated bias threshold $b_i^{(t)}$. Consequently, the indicator functions become consistently not activated $T_1 < t \le T_2$ such that

$$\mathbf{1}_{\left|\langle w_i^{(t)}, x_p \rangle\right| \ge b_i^{(t)}} \cdot \mathbf{1}_{\left|\langle v_i^{(t)}, y_p \rangle\right| \ge b_i^{(t)}} = 0, \quad (91)$$

Using Eq (34), the weight dynamics for $|\langle w_i^{(t+1)}, \mathbf{M}_j \rangle|$ can be expressed as when $T_1 < t \le T_2$:

$$|\langle w_i^{(t+1)}, \mathbf{M}_j \rangle| \le \left(1 + o\left(\frac{\eta}{d^2}\right)\right)^t \left(\frac{\langle w_i^{(T_1)}, \mathbf{M}_j \rangle + \langle v_i^{(T_1)}, \mathbf{H}_{j'} \rangle + \langle w_i^{(T_1)}, \mathbf{M}_{\mathbf{j}} \rangle + \langle v_i^{(T_1)}, \mathbf{H}_j \rangle}{4}\right)$$
$$(92)$$

Because $\left(1 + o\left(\frac{\eta}{d^2}\right)\right)^{T_2} \le 1 + o\left(\frac{1}{d}\right)$, the growth in $|\langle w_i^{(T_2)}, \mathbf{M}_j \rangle|$ is negligible. Consequently, we have:

$$|\langle w_i^{(T_2)}, \mathbf{M}_j \rangle|^2 \le \left(1 + o\left(\frac{1}{d}\right)\right)|\langle w_i^{(T_1)}, \mathbf{M}_j \rangle|^2. \quad (93)$$

## E.3  Summary

When $T_2 = \Theta\left(\frac{d\log d}{\eta}\right)$, we know $\left(1 + \eta\frac{C_z}{d}\right)^{T_2} = poly(d)$. Using Eq (76), we can ensure that when all neurons satisfy the following condition:

$$\|w_i^{(T_2)}\|_2 \ge \Omega(d)\|w_i^{(T_1)}\|_2, \quad (94)$$

we terminate the training process at $T_2 = \Theta\left(\frac{d\log d}{\eta}\right)$. This ensures that the alignment has sufficiently progressed for effective learning.

Thus, using Eq (93) and Eq (71) we have

$$|\langle w_i^{(T_2)}, \mathbf{M}_j \rangle|^2 + |\langle w_i^{(T_2)}, \mathbf{M}_{j'} \rangle|^2 = \|w_i^{(T_2)}\|_2^2 - \sum_{j \in [d], j \notin \mathcal{N}_i} \langle w_i^{(T_2)}, \mathbf{M}_j \rangle^2 - \sum_{j \in [d_1]\setminus[d]} \langle w_i^{(T_2)}, \mathbf{M}_j^{\perp} \rangle^2$$

$$\ge \|w_i^{(T_2)}\|_2^2 - (1 + o(\frac{1}{d}))(\|w_i^{(T_1)}\|_2^2 - |\langle w_i^{(T_1)}, \mathbf{M}_j \rangle|^2 - |\langle w_i^{(T_1)}, \mathbf{M}_{j'} \rangle|^2)$$

$$\ge \|w_i^{(T_2)}\|_2^2 - \|w_i^{(T_1)}\|_2^2 - o(\frac{\|w_i^{(T_1)}\|_2^2}{d})$$

$$(95)$$

Thus, at this stage ($T_1 < t \leq T_2$), we do not consider the worst-case scenario where the probability bounds for feature coupling satisfy

$$\frac{\log(1/c_0)}{2\log d} < \Pr(|z_{y_p}^{j'}| = 1 \mid |z_{x_p}^{j}| = 1) < \frac{1}{2} < \Pr(|z_{y_p}^{j}| = 1 \mid |z_{x_p}^{j}| = 1) < 1$$

We summarize the results when $T_1 < t \leq T_2$ as follows:

1. For $i \in S_{j,\text{sure}}$, the alignment strength satisfies:

$$|\langle w_i^{(T_2)}, \mathbf{M}_j \rangle|^2 > |\langle w_i^{(T_2)}, \mathbf{M}_{j'} \rangle|^2 \geq \frac{1}{4} \frac{\|w_i^{(T_2)}\|_2^2 + \|v_i^{(T_2)}\|_2^2}{2} \tag{96}$$

where $j'$ represents the corresponding spurious alignment feature.

2. For $i \notin S_{j,\text{pot}}$, the alignment strength satisfies:

$$|\langle w_i^{(T_1)}, \mathbf{M}_j \rangle|^2 \leq O(\frac{1}{d}) \cdot \frac{\|w_i^{(T_2)}\|_2^2 + \|v_i^{(T_2)}\|_2^2}{2} \tag{97}$$

3. For $\mathbf{M}_j^\perp$ where $j \in [d_1] \setminus [d]$, we have:

$$|\langle w_i^{(t+1)}, \mathbf{M}_j^\perp \rangle|^2 < O\left(\frac{1}{d_1}\right) \cdot \frac{\|w_i^{(T_2)}\|_2^2 + \|v_i^{(T_2)}\|_2^2}{2}. \tag{98}$$

These results demonstrate that when $T_1 < t \leq T_2$, the alignment for $i \in S_{j,\text{sure}}$, including the corresponding spurious alignment, grows significantly larger. In contrast, the alignment strength for $i \notin S_{j,\text{pot}}$ and noisy features $\mathbf{M}^\perp$ remains unchanged. Similar results also hold for $v_i$.

# F    ITCP on Raw Data III Convergence

In the previous section, we demonstrated that for $t \leq T_2$, the neurons $(w_i, v_i)$ are sparsely activated and remain consistently activated for $i \in S_{j,\text{sure}}$. Building on this result, this section establishes the convergence of these neurons to sparse solutions, providing a detailed analysis of their behavior during Phase III of ITCP on Raw Data. The following theorem outlines the convergence guarantees under these conditions.

The Phase III of ITCP on Raw Data is defined as the training iterations $T_2 < t \leq T_3$, where $T_3 - T_2 = \Theta(d)$. At the beginning of this phase, we fix the bias threshold as $b_i^{(t)} = b_i^{T_2}$ for $T_2 < t \leq T_3$. Because $b_i^{(T_2)} = \left(1 + \frac{\eta}{d}\right)^{\Theta(d \log d/\eta)} b_i^{(T_1)}$, it is easy to know that for $t \geq T_2$, only when $(x_p, y_p)$ and $(x_n, y_n)$ contain the true feature $j$ and its corresponding spurious feature $j'$, the indicator functions remain consistently activated for $i \in S_{j,\text{sure}}$.

Consequently, using Eq (27), Eq (30), and Eq (31), the loss function $L$ becomes convex with respect to $w_i$ and $v_i$ independently when $(x_p, y_p)$ and $(x_n, y_n)$ contain the true feature $j$ and its corresponding spurious feature $j'$.

At the end of Phase II, using Eq (81), we know that $\|w_i^{(T_2)}\|_2 \geq \Omega(d)$. Consequently, we cannot only consider $-\langle f^{(t)}(x_p), h^{(t)}(y_p) \rangle$, and the error term $Err_t$ becomes non-negligible.

Specifically, based on Eq (27), it can be observed that the term $-\langle f^{(t)}(x_p), h^{(t)}(y_p) \rangle$ is convex and $l_{i,j,1} = \|x_p\|_2 \|y_p\|_2 = \Theta(1)$-smooth. This ensures that the true features contribute consistently to the optimization process.

Additionally, $L_{i,j,2} = \frac{\left(\langle f^{(t)}(x_n), h^{(t)}(y_p) \rangle\right)^2}{2\tau}$ is also convex, and we further establish its smoothness to provide a rigorous understanding of its behavior.

To analyze the $l_{i,j,2}$-smoothness, we aim to find an upper bound that satisfies:

$$\|\nabla_{w_i, v_i} L_2(w_{i,1}, v_{i,1}) - \nabla_{w_i, v_i} L_2(w_{i,2}, v_{i,2})\|_2 \leq l_{i,j,2} \|(w_{i,1} - w_{i,2}, v_{i,1} - v_{i,2})\|_2. \tag{99}$$

The gradient difference for $w_i$ is given by:

$$\|\nabla_{w_i} L_{i,j,2}(w_{i,1}, v_{i,1}) - \nabla_{w_i} L_{i,j,2}(w_{i,2}, v_{i,2})\|_2 = \frac{\left\| \left(x^\top W_1^\top V_1 y\right) x(v_{i,1} y)^\top - \left(x^\top W_2^\top V_2 y\right) x(v_{i,2} y)^\top \right\|_2}{2\tau}$$

$$\leq \frac{l_{w_i,1}}{2\tau}\|w_{i,1} - w_{i,2}\|_2 + \frac{l_{w_i,2}}{2\tau}\|v_{i,1} - v_{i,2}\|_2,$$

(100)

where $l_{w_i,1} = \|x_n\|_2^2\|y_p\|_2^2\|v_{i,1}\|_2\|v_{i,2}\|_2 \leq O(d)$ and $l_{w_i,2} = \|x_n\|_2^2\|y_p\|_2^2\left(\|v_{i,1}\|_2\|w_{i,2}\|_2 + \|w_{i,1}\|_2\|v_{i,1}\|_2\right) \leq O(d)$.

Similarly, the gradient difference for $v_i$ is:

$$\|\nabla_{v_i} L_{i,j,2}(w_{i,1}, v_{i,1}) - \nabla_{v_i} L_{i,j,2}(w_{i,2}, v_{i,2})\|_2 \leq \frac{l_{v_i,1}}{2\tau}\|w_{i,1} - w_{i,2}\|_2 + \frac{l_{v_i,2}}{2\tau}\|v_{i,1} - v_{i,2}\|_2,$$

(101)

where $l_{v_i,1} \leq O(d)$ and $l_{v_i,2} \leq O(d)$.

Combining the results, we find:

$$l_{i,j,2} = \frac{\sqrt{l_{w_i,1}^2 + l_{w_i,2}^2 + l_{v_i,1}^2 + l_{v_i,2}^2}}{2\tau} \leq O(1).$$

(102)

Thus, the total smoothness constant is:

$$l_{i,j} = l_{i,j,1} + l_{i,j,2} = \Theta(1).$$

(103)

These results demonstrate that the loss function $L$ remains convex and $l_{i,j}$-smooth for neurons $(w_i, v_i)$ when $(x_p, y_p)$ and $(x_n, y_n)$ contain the true feature $j$ and its corresponding spurious feature $j'$ during Phase III of ITCP on Raw Data, ensuring their convergence to sparse solutions while maintaining consistency in their activation patterns.

We verify that the following inequality holds

$$L_j(w_i^{(t+1)}, v_i^{(t+1)}) \leq L_j(w_i^{(t)}, v_i^{(t)})$$
$$+ \left\langle \nabla L_j(w_i^{(t)}, v_i^{(t)}), \left(w_i^{(t+1)} - w_i^{(t)}, v_i^{(t+1)} - v_i^{(t)}\right) \right\rangle$$
$$+ \frac{l_{i,j}}{2} \left\| \left(w_i^{(t+1)} - w_i^{(t)}, v_i^{(t+1)} - v_i^{(t)}\right) \right\|^2$$

(104)

Let $L = \max_{i \in m}(l_{i,j}/(2\tau)) = \Theta(1)$ and $\eta = \frac{1}{L}$ to ensure a monotonic decrease, plug Eq (28) and Eq (29) into Eq (178), we have

$$L_j(w_i^{(t+1)}, v_i^{(t+1)}) \leq L_j(w_i^{(t)}, v_i^{(t)}) - \frac{\eta}{2}\|\nabla L_j(w_i^{(t)}, v_i^{(t)})\|^2.$$

(105)

Under our data assumptions for $S_w$ and conclusion in Eq (96), we define $w_i^* = \alpha_{i,j}^* \mathbf{M}_j + \alpha_{i,j'}^* \mathbf{M}_{j'}, v_i^* = \alpha_{i,j}^* \mathbf{H}_j + \alpha_{i,j'}^* \mathbf{H}_{j'}$. Thus, $L_j(w_i^*, v_i^*)$ captures both the alignment with the true feature $\mathbf{M}_j, \mathbf{H}_j$ and the spurious feature $\mathbf{M}_{j'}, \mathbf{H}_{j'}$, representing the minimal loss achievable under the influence of both true and spurious features in the optimization process. Using Eq (81), we know $w_i^{(T_2)} = \Theta(d)$, so $L_j(w_i^*, v_i^*) = -\Theta(d)$.

By the property of smoothness, we have

$$\|\nabla L_j(w_i^{(t)}, v_i^{(t)})\|_2^2 \geq \frac{2}{L}\left(L_j(w_i^{(t)}, v_i^{(t)}) - L_j(w_i^*, v_i^*)\right)$$

(106)

Take the telescope sum of from $T_2$ to $T_3$, we have

$$\frac{1}{T_3 - T_2}\sum_{t=T_2}^{T_3} L_j(w_i^{(t)}, v_i^{(t)}) \leq L_j(w_i^*, v_i^*) + \frac{L^2\Delta_0}{T_3 - T_2}$$

$$\overset{\diamondsuit}{\leq} L_j(w_i^*, v_i^*) + \Theta(1)$$

(107)

where $\Delta_0 = L_j(w_i^{(T_1)}, v_i^{(T_1)}) - L_j(w_i^*, v_i^*) = \Theta(d)$. In $\Diamond$, we use $T_3 - T_2 = \Theta(d)$, and $L = \Theta(1)$
.

Generalized to every $j \in d$, the same convergence holds for all $i \in S_{j,\text{sure}}$ when $(x_p, y_p)$ and $(x_n, y_n)$ contain feature $j, j'$. For all $(x_p, y_p)$ and $(x_n, y_n)$ in $S_w$, the following inequality holds:

$$\frac{1}{T_3 - T_2} \sum_{t=T_2}^{T_3} L(f^{(T_3)}, h^{(T_3)}) \leq L(f^*, h^*) + \Theta(1). \tag{108}$$

As a result, the relative difference is bounded by:

$$\frac{L(f^{(T_3)}, h^{(T_3)}) - L(f^*, h^*)}{|L(f^*, h^*)|} \leq \Theta\left(\frac{1}{d}\right). \tag{109}$$

### F.1 Summary

ITCP trained on raw data $S$ undergoes Stages D–F. After $T = \Theta(d^2 \log d)$ SGD iterations with batch size $B = \Omega(d)$ and learning rate $\eta = O(1)$, the resulting weights $(\overline{\mathbf{W}}, \overline{\mathbf{V}})$ minimize the contrastive loss in Eq. (1) up to a vanishing relative error:

$$\frac{L(f_{\overline{\mathbf{W}}}, h_{\overline{\mathbf{V}}}) - L^*}{|L^*|} \leq o(1). \tag{110}$$

However, each neuron pair $(\bar{w}_i, \bar{v}_i)$ in $(\overline{\mathbf{W}}, \overline{\mathbf{V}})$, for $i \in [m]$, predominantly encodes a mixture of features indexed by a subset $N_i \subseteq [d]$, with $|N_i| \geq 2$. Specifically, we have:

$$
\begin{aligned}
\bar{w}_i &= \sum_{j \in N_i} \alpha_{i,j} \mathbf{M}_j + \sum_{j \in [d] \setminus N_i} \beta_{i,j} \mathbf{M}_j + \sum_{j \in [d_1] \setminus [d]} \gamma_{i,j} \mathbf{M}_j^\perp, \\
\bar{v}_i &= \sum_{j \in N_i} \alpha_{i,j} \mathbf{H}_j + \sum_{j \in [d] \setminus N_i} \beta_{i,j} \mathbf{H}_j + \sum_{j \in [d_1] \setminus [d]} \gamma_{i,j} \mathbf{H}_j^\perp,
\end{aligned}
\tag{111}
$$

where $\alpha_{i,j}^2 = \Theta(\|\bar{w}_i\|_2^2 + \|\bar{v}_i\|_2^2)$, and the interference from other features is small: $\beta_{i,j}/\alpha_{i,j} \leq O(1/\sqrt{d}), \gamma_{i,j}/\alpha_{i,j} \leq O(1/\sqrt{d_1})$.

Moreover, for every spuriously correlated feature pair $(j, j')$ satisfying Assumption 3.3, there exists at least an $\Omega(1)$ many of neurons $i \in [m]$ with $N_i = \{j, j'\}$, indicating the prevalence of feature mixing due to data misalignment.

## G  Captioning

In this stage, the model fine-tunes the pre-trained encoder parameters $\mathbf{W}$ and $\mathbf{V}$ to obtain the updated parameters $\hat{\mathbf{W}}$ and $\hat{\mathbf{V}}$ through Image-Text Contrastive Pre-training (ITCP) on raw data.

Given an image-text pair $(x_p, y_p)$ in $S_w$, the decoder generates synthetic captions $\hat{y}_p = \hat{\mathbf{V}}^T \sigma(\hat{\mathbf{W}} x_p)$, where $\sigma(\cdot)$ denotes the activation function. The Image-Grounded Text Decoder, initialized with $\mathbf{W}$ and $\mathbf{V}$ from the pre-trained encoders, is fine-tuned on $S_h$ by minimizing the following loss function:

$$L_C = \mathbb{E}_{(x_p, y_p) \in S_h} \left[ \frac{1}{2} \left\| \mathbf{V}^T \sigma(\mathbf{W} x_p) - y_p \right\|_2^2 \right], \tag{112}$$

where $\| \cdot \|_2$ denotes the Euclidean norm. This fine-tuning process refines the model to generate captions that are more closely aligned with the target text data in $S_h$.

During the captioning, we sample a batch of image-text pairs $S_h^{(t)} = \{(x_p, y_p)\}_{p=1}^B \subseteq S_h$. We perform stochastic gradient descent on $L_C$. At each iteration, we update as

$$w_i^{(t+1)} \leftarrow w_i^{(t)} - \eta \nabla_{w_i} L_C^{(t)} \tag{113}$$

$$v_i^{(t+1)} \leftarrow v_i^{(t)} - \eta \nabla_{v_i} L_C^{(t)} \tag{114}$$

At the beginning of this phase, we set the bias threshold as:

$$b_i^{(0)} = \sqrt{\frac{\|w_i^{(T_2)}\|_2^2 - \|w_i^{(T_1)}\|_2^2}{2}} \tag{115}$$

During training, the bias threshold is iteratively updated as:

$$b_i^{(t+1)} = \left(1 + \frac{\eta}{d}\right) b_i^{(t)}, \tag{116}$$

The gradient of $L_C$ with respect to $w_i^{(t)}$, $v_i^{(t)}$, $\mathbf{W}$, and $\mathbf{V}$ at iteration $t$ is given by:

$$\nabla_{w_i^{(t)}} L_C = v_i^{(t)}(y_p - \mathbf{V}^T\mathbf{W}x_p)x_p^T \cdot \mathbf{1}_{\left|\left\langle w_i^{(t)}, x_p\right\rangle\right| \geq b_i^{(t)}} \tag{117}$$

$$\nabla_{v_i^{(t)}} L_C = w_i^{(t)}x_p(y_p - \mathbf{V}^T\mathbf{W}x_p)^T \cdot \mathbf{1}_{\left|\left\langle w_i^{(t)}, x_p\right\rangle\right| \geq b_i^{(t)}} \tag{118}$$

The alignment can be described by the following update rule:

$$\begin{aligned}
\langle w_i^{(t+1)}, \mathbf{M}_j\rangle &= \langle w_i^{(t)}, \mathbf{M}_j\rangle - \langle \nabla_{w_i} L_C, \mathbf{M}_j\rangle \\
&= \langle w_i^{(t)}, \mathbf{M}_j\rangle + \eta \cdot \mathrm{tr}(v_i^{(t)\top}(y_p - \mathbf{V}^T\mathbf{W}x_p)x_p^T\mathbf{M}_j \cdot \mathbf{1}_{|\langle w_i, x_p\rangle| \geq b_i^{(t)}})
\end{aligned} \tag{119}$$

$$\begin{aligned}
\langle v_i^{(t+1)}, \mathbf{H}_j\rangle &= \langle v_i^{(t)}, \mathbf{H}_j\rangle - \langle \nabla_{v_i} L_C, \mathbf{H}_j\rangle \\
&= \langle v_i^{(t)}, \mathbf{H}_j\rangle + \eta \cdot \mathrm{tr}(w_i^{(t)\top}x_p(y_p - \mathbf{V}^T\mathbf{W}x_p)^T\mathbf{H}_j \cdot \mathbf{1}_{\left|\left\langle w_i^{(t)}, x_p\right\rangle\right| \geq b_i^{(t)}})
\end{aligned} \tag{120}$$

## G.1 Alignment for $i \in S_{j,\text{sure}}$

This section analyzes the alignment growth for neurons $i \in S_{j,\text{sure}}$. Specifically, we show that when $t \leq T_C$, the alignment with the true feature $\mathbf{M}_j$ grows exponentially if $x_p$ contains the true feature $\mathbf{M}_j$. In contrast, the alignment with the spurious feature $\mathbf{M}_{j'}$ exhibits negligible growth, even for neurons $i \in S_{j,\text{sure}}$. Specially,

1. For the true feature $\mathbf{M}_j$, based on the result in Eq (96) and the bias threshold in Eq (115), the indicator functions are always activated. This ensures that the neuron can consistently increase its alignment in the direction of the true feature $\mathbf{M}_j$.

2. For the spurious feature $\mathbf{M}_{j'}$, based on the result in Eq (96) and the bias threshold in Eq (115), the indicator functions remain non-activated. This prevents the neuron from increasing its alignment in the direction of the spurious feature $\mathbf{M}_{j'}$.

The details of proof as follow:

Using Eq (95), we know

$$\|w_i^{(T_2)}\|_2^2 - \|w_i^{(T_1)}\|_2^2 \geq |\langle w_i^{(T_2)}, \mathbf{M}_j\rangle|^2 + |\langle w_i^{(T_2)}, \mathbf{M}_{j'}\rangle|^2 \geq \|w_i^{(T_2)}\|_2^2 - \|w_i^{(T_1)}\|_2^2 - o(\frac{\|w_i^{(T_1)}\|_2^2}{d})$$

$$\tag{121}$$

Using Eq (35) and Eq (36), we have

$$\langle w_i^{(t)}, \mathbf{M}_j\rangle - \langle w_i^{(t)}, \mathbf{M}_{j'}\rangle = \frac{(a+b-c)^t}{2}\left(\langle w_i^{(0)}, \mathbf{M}_j\rangle + \langle v_i^{(0)}, \mathbf{H}_j\rangle - \langle w_i^{(0)}, \mathbf{M}_{j'}\rangle - \langle v_i^{(0)}, \mathbf{H}_{j'}\rangle\right) + Err_t$$

$$\tag{122}$$

Using Eq (40) and $(a+b-c)^{T_1+T_2} \geq \Omega(d^2)$, with high probability $1 - O(\frac{1}{\sqrt{d}})$ we have,

$$|\langle w_i^{(T_2)}, \mathbf{M}_j\rangle|^2 - |\langle w_i^{(T_2)}, \mathbf{M}_{j'}\rangle|^2 \geq \Omega(\frac{\|w_i^{(T_1)}\|_2^2}{d}) \tag{123}$$

Therefore, with high probability $1 - O(\frac{1}{\sqrt{d}})$ we have

$$|\langle w_i^{(T_2)}, \mathbf{M}_j\rangle|^2 > \frac{\|w_i^{(T_2)}\|_2^2 - \|w_i^{(T_1)}\|_2^2}{2} > |\langle w_i^{(T_2)}, \mathbf{M}_{j'}\rangle|^2 \tag{124}$$

We set $b_i^{(0)} = \sqrt{\frac{\|w_i^{(T_2)}\|_2^2 - \|w_i^{(T_1)}\|_2^2}{2}}$, and using Eq (124), so similarly to the proof of Eq (86) we can prove:

1. For $i \in S_{j,\text{sure}}$ and $x_p$ contain the true feature $\mathbf{M}_j$, with high probability $1 - O(\frac{1}{\sqrt{d}})$ the indicator functions become consistently activated $0 \le t \le T_C$ such that:

$$\mathbf{1}_{\left|\left\langle w_i^{(t)}, x_p \right\rangle\right| \ge b_i^{(t)}} = 1 \tag{125}$$

2. For $i \in S_{j,\text{sure}}$ and $x_p$ contain the corresponding spurious aligned feature $\mathbf{M}_{j'}$, with high probability $1 - O(\frac{1}{\sqrt{d}})$ the indicator functions become consistently activated $0 \le t \le T_C$ such that:

$$\mathbf{1}_{\left|\left\langle w_i^{(t)}, x_p \right\rangle\right| \ge b_i^{(t)}} = 0 \tag{126}$$

3. For $i \notin S_{j,\text{pot}}$ and $\mathbf{M}_j^\perp$ where $j \in [d_1] \setminus [d]$, we have:

$$\mathbf{1}_{\left|\left\langle w_i^{(t)}, x_p \right\rangle\right| \ge b_i^{(t)}} = 0 \tag{127}$$

For the residual loss in Eq (119) and Eq (120), we bound the difference if $\mathbf{1}_{\left|\left\langle w_i^{(t)}, x_p \right\rangle\right| \ge b_i^{(t)}} = 1$:

$$
\begin{aligned}
\mathbf{H}_j z_{x_p}^j z_{y_p}^j &\overset{\Diamond}{\ge} (y_p - \mathbf{V}^T \mathbf{W} x_p) x_p^T \mathbf{M}_j \cdot \mathbf{1}_{|\langle w_i, x_p \rangle| \ge b_i^{(t)}} \\
&= (\mathbf{H}_j z_{x_p}^j z_{y_p}^j - \sum_{i=1}^m \langle v_i, \mathbf{H}_j \rangle \langle w_i, \mathbf{M}_j \rangle \mathbf{H}_j z_{x_p}^j z_{y_p}^j) \cdot \mathbf{1}_{\langle w_i, x_p \rangle \ge b} \\
&\overset{\heartsuit}{\ge} \mathbf{H}_j z_{x_p}^j z_{y_p}^j - O(d^{\gamma c_0}) \langle v_i, \mathbf{H}_j \rangle \langle w_i, \mathbf{M}_j \rangle \mathbf{H}_j z_{x_p}^j z_{y_p}^j
\end{aligned} \tag{128}
$$

In $\Diamond$, we employ the approximation $y_p x_p^\top \mathbf{M}_j \approx \mathbf{H}_j z_{x_p}^j z_{y_p}^j$, based on the observation that $z_{x_p}^j z_{y_p}^{j'} \ll z_{x_p}^j z_{y_p}^j$ when $j \neq j'$. In $\heartsuit$, we utilize Eq (38). There are at most $O(d^{\gamma c_0})$ neurons capable of learning $\mathbf{M}_j$, which satisfy the condition $\mathbf{1}_{\langle w_i, x_p \rangle \ge b}$.

For $i \in S_{j,\text{sure}}$ and for $x_p$ contain $\mathbf{M}_j$, using Eq (128), Eq (119) and Eq (126) we have:

$$
\begin{aligned}
\langle w_i^{(t+1)}, \mathbf{M}_j \rangle &\ge \langle w_i^{(t)}, \mathbf{M}_j \rangle + \eta \cdot \text{tr}\left( v_i^{(t)} \cdot (1 - \alpha_t^2) \mathbf{H}_j \mathbb{E}\left[ z_{x_p}^j z_{y_p}^j \right] \right) \\
&\ge \langle w_i^{(t)}, \mathbf{M}_j \rangle + \eta \frac{C_z (1 - \alpha_t^2)}{d} \langle v_i^{(t)}, \mathbf{H}_j \rangle,
\end{aligned} \tag{129}
$$

Similar to Eq (35), we have

$$|\langle w_i^{(t)}, \mathbf{M}_j \rangle| \ge \left( 1 + \eta \frac{C_z \cdot (1 - \alpha_t^2)}{d} \right)^t \left( \frac{\langle w_i^{(0)}, \mathbf{M}_j \rangle + \langle v_i^{(0)}, \mathbf{H}_j \rangle}{2} \right) \tag{130}$$

Similarly, for $i \in S_{j,\text{sure}}$ and $x_p$ contain the corresponding spurious aligned feature $\mathbf{M}_{j'}$, because $\Pr[\mathbf{1}_{\left|\left\langle w_i^{(t)}, x_p \right\rangle\right| \ge b_i^{(t)}} = 0] \ge 1 - O(\frac{1}{\sqrt{d}})$, we have

$$\langle w_i^{(t+1)}, \mathbf{M}_{j'} \rangle \le \langle w_i^{(t)}, \mathbf{M}_{j'} \rangle + O(\frac{\eta}{d^{1.5}}) \langle v_i^{(t)}, \mathbf{H}_{j'} \rangle \tag{131}$$

and

$$|\langle w_i^{(t)}, \mathbf{M}_{j'} \rangle| \le \left( 1 + O(\frac{\eta}{d^{1.5}}) \right)^t \left( \frac{\langle w_i^{(T_2)}, \mathbf{M}_{j'} \rangle + \langle v_i^{(T_2)}, \mathbf{H}_{j'} \rangle}{2} \right) \tag{132}$$

At $T_C = \Theta\left( \frac{d \log(d)}{\eta} \right)$, we have:

$$\frac{|\langle w_i^{(T_C)}, \mathbf{M}_j \rangle|}{|\langle w_i^{(T_C)}, \mathbf{M}_{j'} \rangle|} > \frac{\left( 1 + \eta \frac{C_z \cdot (1 - \alpha_t^2)}{d} \right)^{T_C}}{\left( 1 + O(\frac{\eta}{d^{1.5}}) \right)^{T_C}} \ge \Omega(d) \tag{133}$$

Therefore, we summarize that when $t = T_C$, the alignment with the true feature $\mathbf{M}_j$ dominates, satisfying:

$$\frac{|\langle w_i^{(T_C)}, \mathbf{M}_j \rangle|}{|\langle w_i^{(T_C)}, \mathbf{M}_{j'} \rangle|} \geq \Omega(d), \tag{134}$$

highlighting the significant separation between the true feature $\mathbf{M}_j$ and the spurious feature $\mathbf{M}_{j'}$ for neurons $i \in S_{j,\text{sure}}$. A similar result holds for $v_i$, where the alignment with the true feature $\mathbf{H}_j$ similarly dominates over the spurious feature $\mathbf{H}_{j'}$.

## G.2 Convergence

For $i \in S_{j,\text{sure}}$, when $x_p, y_p$ contains the true feature $j$, the indicator functions remain consistently activated. Consequently, the loss function $L_C$ becomes convex with respect to $w_i$ and $v_i$ independently. We verify that the following inequality holds

$$
\begin{aligned}
L_{C,j}(w_i^{(t+1)}, v_i^{(t+1)}) \leq {} & L_{C,j}(w_i^{(t)}, v_i^{(t)}) \\
& + \left\langle \nabla L_{C,j}(w_i^{(t)}, v_i^{(t)}), \left(w_i^{(t+1)} - w_i^{(t)}, v_i^{(t+1)} - v_i^{(t)}\right) \right\rangle \\
& + \frac{l_i}{2} \left\| \left(w_i^{(t+1)} - w_i^{(t)}, v_i^{(t+1)} - v_i^{(t)}\right) \right\|^2
\end{aligned}
\tag{135}
$$

where $l_i = O(C_z d^{2\gamma c_0})(\left\|v_i^{(t)}\right\|_2^2 \|x_p\|_2^2 + \left\|v_i^{(t)}\right\|_2^2 \|x_p\|_2^2) = \Theta(1)$. This means $L_{C,j}(w_i^{(t)}, v_i^{(t)})$ is $l_i$-smooth for all $i \in S_{j,\text{sure}}$ when $x_p, y_p$ contains the true feature $j$. Let $L = \max_{i \in m}(l_i) = \Theta(1)$

Let $\eta = \frac{1}{L}$ to ensure a monotonic decrease, plug Eq (117) and Eq (118) into Eq (135), we have

$$L_{C,j}(w_i^{(t+1)}, v_i^{(t+1)}) \leq L_{C,j}(w_i^{(t)}, v_i^{(t)}) - \frac{\eta}{2}\|\nabla L_{C,j}(w_i^{(t)}, v_i^{(t)})\|^2. \tag{136}$$

By the property of smoothness, we have

$$\|\nabla L_{C,j}(w_i^{(t)}, v_i^{(t)})\|_2^2 \geq \frac{2}{L}\left(L_{C,j}(w_i^{(t)}, v_i^{(t)}) - L_{C,j}(w_i^*, v_i^*)\right). \tag{137}$$

Take the telescope sum of from 0 to $T_C$, we have

$$
\begin{aligned}
\frac{1}{T_C} \sum_{t=0}^{T_C} L_{C,j}(w_i^{(t)}, v_i^{(t)}) & \leq L_{C,j}(w_i^*, v_i^*) + \frac{L^2 \Delta_0}{T_C} \\
& \overset{\diamond}{\leq} L_{C,j}(w_i^*, v_i^*) + \Theta(\frac{1}{d}) \\
& \overset{\heartsuit}{=} \Theta(\frac{1}{d})
\end{aligned}
\tag{138}
$$

where $\Delta_0 = L_{C,j}(w_i^{(0)}, v_i^{(0)}) - L_{C,j}(w_i^*, v_i^*)$. In $\diamond$, we use $T_C = \Theta(d)$, and $\|w_i^{(t)}\|_2^2 = \|v_i^{(t)}\|_2^2 = \Theta(1)$. In $\heartsuit$, we use $w_i^* = \alpha_{i,j}^* \mathbf{M}_j, V_i^* = \alpha_{i,j}^* \mathbf{H}_j$ and $L_{C,j}(w_i^*, v_i^*) = \Theta(\frac{1}{d})$ if $x_p$ contains the true feature $\mathbf{M}_j$.

Therefore, for all $j \in d$ and all $(x_p, y_p) \in S_h$, when $T_C = \Theta(d^2)$, we can ensure

$$L_C = \mathbb{E}_{(x_p, y_p) \in S_h} \left[ \frac{1}{2} \left\| \mathbf{V}^T \sigma(\mathbf{W} x_p) - y_p \right\|_2^2 \right] \leq \Theta(\frac{1}{d}) \tag{139}$$

## G.3 Summary

After $T_C$ iterations, the parameters $\mathbf{W}$ and $\mathbf{V}$ are updated to $\mathbf{W}^{T_C} = \hat{\mathbf{W}}$ and $\mathbf{V}^{T_C} = \hat{\mathbf{V}}$, respectively, using the dataset $S_h$. The generated caption is given by:

$$\hat{y}_p = \hat{\mathbf{V}}^T \sigma(\hat{\mathbf{W}} x_p), \tag{140}$$

where the expected loss satisfies:

$$\mathbb{E}\left[\frac{1}{2}\|\hat{y}_p - y_p\|_2^2\right] = L_C \leq \Theta\left(\frac{1}{d}\right). \tag{141}$$

1. For $i \in S_{j,\text{sure}}$, the alignment strength satisfies:

$$|\langle w_i^{(T_C)}, \mathbf{M}_j\rangle|^2 = \Theta(1)\left\|w_i^{(T_C)}\right\|_2^2 \tag{142}$$

and

$$|\langle w_i^{(T_C)}, \mathbf{M}_j'\rangle|^2 \leq O(\frac{1}{d})\left\|w_i^{(T_C)}\right\|_2^2 \tag{143}$$

where $j'$ represents the corresponding spurious alignment feature.

2. For $i \notin S_{j,\text{pot}}$, the alignment strength satisfies:

$$|\langle w_i^{(T_1)}, \mathbf{M}_j\rangle|^2 \leq O(\frac{1}{d})\left\|w_i^{(T_C)}\right\|_2^2 \tag{144}$$

3. For $\mathbf{M}_j^\perp$ where $j \in [d_1] \setminus [d]$, we have:

$$|\langle w_i^{(t+1)}, \mathbf{M}_j^\perp\rangle|^2 < O(\frac{1}{d_1})\left\|w_i^{(T_C)}\right\|_2^2 \tag{145}$$

## H  Filtering

During filtering, we sample the synthetic image-text pair $(x_p, \hat{y}_p)$ in $\hat{S}_w$ and the corresponding image-text pair $(x_p, y_p)$ in $S_w$. The image encoder $f$ and text encoder $h$ trained on raw data are employed to obtain the corresponding embeddings.

$$z'_{x_p} = f(x_p), \quad \hat{z}_{y_p} = h(\hat{y}_p), \quad z'_{y_p} = h(y_p) \tag{146}$$

Then, we calculate the cosine similarity of $\langle z'_{x_p}, \hat{z}_{y_p}\rangle$ and $\langle z'_{x_p}, z'_{y_p}\rangle$, and select the image-text pair with higher cosine similarity denoted as $(x, \tilde{y})$. In this way, we replace the noisy pairs in $S_w$ with synthetic pairs in $\hat{S}_w$. The resulting dataset is denoted as $\tilde{S} = \tilde{S}_w \cup S_h$.

The decoder generates synthetic captions $\hat{y}_p = \hat{\mathbf{V}}^T\sigma(\hat{\mathbf{W}}x_p)$. Using Eq (141), for each data pair $(x_p, y_p)$ which contain feature $(\mathbf{M}_j, \mathbf{H}_j)$ in $S_h$ we have

$$\mathbb{E}_{(x_p,y_p)}\left[\mathbb{E}_{j\in d}\left[\frac{1}{2}\left\|\mathbf{H}_j z^j_{\hat{y}_p} - \mathbf{H}_j z^j_{y_p}\right\|_2^2\right]|z^j_{y_p}| = 1\right] \leq \mathbb{E}_{(x_p,y_p)}\left[\frac{1}{2}\|\hat{y}_p - y_p\|_2^2\,|z^j_{y_p}| = 1\right] = L_C \leq \Theta(\frac{1}{d}) \tag{147}$$

Therefore, using $\|\mathbf{H}_j\|_2 = 1$ and $z_{x_p} = z_{y_p}$ in $S_h$, we have

$$\mathbb{E}_{x_p,j\in d}\left[z^j_{\hat{y}_p} z^j_{x_p}|z^j_{x_p}| = 1\right] \geq 1 - \Theta(\frac{1}{d}) \tag{148}$$

Base on Assumption B.1 $z^j_{x_p} \sim \text{Bernoulli}\left(\frac{C_z}{d}\right)$, we have

$$\Pr(z^j_{\hat{y}_p} = 1 \mid |z^j_{x_p}| = 1) \geq 1 - \Theta(\frac{1}{d}) \tag{149}$$

Using Eq (134) and Eq (149), we have

$$\Pr(z^{j'}_{\hat{y}_p} = 1 \mid |z^j_{x_p}| = 1) \leq \Theta(\frac{1}{d}) \tag{150}$$

Therefore, after replace all noisy text $y_p$ in $S_w$ by synthetic caption $\hat{y}_p$ in $\hat{S}_w$

1. for a positive pair $(x_p, y_p)$, we have

$$\mathbb{E}\left[z^j_{\tilde{x}_p} z^j_{\tilde{y}_p}\right] = \Theta(\frac{1}{d}), \quad \mathbb{E}\left[z^j_{\tilde{x}_p} z^{j'}_{\tilde{y}_p}\right] = \Theta\left(\frac{1}{d^2}\right), \quad \forall j' \neq j. \tag{151}$$

2. for negative pairs $(x_p, y_q)$, where $p \neq q$, we have:

$$\mathbb{E}\left[z^j_{x_p} z^{j'}_{y_q}\right] = \Theta\left(\frac{1}{d^2}\right), \quad \forall j, j' \in [d]. \tag{152}$$

# I  ITCP on Synthetic (Recaptioned) Data

During ITCP on Raw Data, we use a noisy dataset $S$. Based on SubSection B.2, we have $\mathbb{E}\left[z_x^j z_y^j\right]$ and $\mathbb{E}\left[z_x^j z_y^{j'}\right]$ both in $\Theta\left(\frac{1}{d}\right)$. In this scenario, for $i \in S_{j,\text{sure}}$, $w_i^{(t)}$ is jointly influenced by $\mathbf{M}_j$ and $\mathbf{M}_{j'}$, with both features contributing comparably to the updates. However, during ITCP on recaptioned data, we sample image-text pairs from the dataset $\tilde{S}$. Using Eq. (151), we find that $\mathbb{E}\left[z_{\tilde{x}_p}^j z_{\tilde{y}_p}^{j'}\right] = \Theta\left(\frac{1}{d^2}\right)$. In this case, for $i \in S_{j,\text{sure}}$, $w_i^{(t)}$ is influenced solely by $\mathbf{M}_j$, without interference from spurious features, ensuring purified representations.

The only difference between ITCP on Raw Data and Data lies in the $\mathbb{E}\left[z_{\tilde{x}_p}^j z_{\tilde{y}_p}^{j'}\right]$; all other training processes remain largely the same. Therefore, we simplify our proof accordingly.

## I.1  Phase I of ITCP on Synthetic Data

The Phase I of ITCP on Data is defined as the training iterations $t \leq T_1$, where $T_1 = \Theta\left(\frac{d \log d}{\eta}\right)$ is the iteration when all $\|w_i^{(T_2)}\|_2^2 = 2\|w_i^{(0)}\|_2^2$. Before $T_1$, we set $b_i^{(t)} = 0$. For every neuron $i \in [m]$, the weights $w_i, v_i$ will mostly ignore the noise features $\mathbf{M}^\perp, \mathbf{H}^\perp$ and learn to emphasize the features $\mathbf{M}, \mathbf{H}$.

If $\Pr(|z_{y_p}^j| = 1 \mid |z_{x_p}^{j'}| = 1) < 0.1$, we have $\mathbb{E}\left[z_x^j z_y^j\right] \gg \mathbb{E}\left[z_x^j z_y^{j'}\right]$ and $(a + b + c)^t \approx (a + b - c)^t$. In this case, $w_i^{(t+1)}$ is predominantly influenced by $\mathbf{M}_j$, with minimal contributions from $\mathbf{M}_{j'}$. The updates are thus primarily driven by the single feature $\mathbf{M}_j$, ensuring that spurious interactions from $\mathbf{M}_{j'}$ are negligible.

$$
\begin{aligned}
\|\mathbf{M}\mathbf{M}^\top w_i^{(t)}\|_2^2 &= \sum_{i=1}^d \left[\frac{(a+b+c)^t}{2}\left(\langle w_i^{(t)}, \mathbf{M}_j\rangle + \langle v_i^{(t)}, \mathbf{H}_j\rangle\right)\right]^2 \\
&= \left(1 + \frac{\eta C_z}{d}\right)^{2t} \frac{\|\mathbf{M}\mathbf{M}^\top w_i^{(0)}\|_2^2 + \|\mathbf{H}\mathbf{H}^\top v_i^{(0)}\|_2^2}{4}.
\end{aligned}
\tag{153}
$$

$i \in S_{j,\text{sure}}$:

$$
\begin{aligned}
|\langle w_i^{(T_1)}, \mathbf{M}_j\rangle|^2 &= \left(1 + \eta\frac{C_z}{d}\right)^{2T_1}\left(\frac{\langle w_i^{(0)}, \mathbf{M}_j\rangle + \langle v_i^{(0)}, \mathbf{H}_j\rangle}{2}\right)^2 \\
&\geq \left(1 + \eta\frac{C_z}{d}\right)^{2T_1} \cdot \frac{c_1 \log d}{d} \cdot \frac{\|\mathbf{M}\mathbf{M}^\top w_i^{(0)}\|_2^2 + \|\mathbf{H}\mathbf{H}^\top v_i^{(0)}\|_2^2}{4} \\
&= \frac{c_1 \log d}{d} \cdot \frac{\|\mathbf{M}\mathbf{M}^\top w_i^{(T_1)}\|_2^2 + \|\mathbf{H}\mathbf{H}^\top v_i^{(T_1)}\|_2^2}{2} \\
&\geq \frac{c_1 \log d}{d} \cdot \frac{\|w_i^{(T_1)}\|_2^2 + \|v_i^{(T_1)}\|_2^2 - \|w_i^{(0)}\|_2^2 - \|v_i^{(0)}\|_2^2}{2} \\
&\geq \frac{(1 + c_0) \log d}{d} \cdot \frac{\|w_i^{(T_1)}\|_2^2 + \|v_i^{(T_1)}\|_2^2}{2}
\end{aligned}
\tag{154}
$$

Because $\frac{\|w_i^{(T_1)}\|_2^2 + \|v_i^{(T_1)}\|_2^2}{2} = \|w_i^{(0)}\|_2^2 + \|v_i^{(0)}\|_2^2$ and $c_1 > 2(1 + c_0)$

$i \notin S_{j,\text{sure}}$:

$$|\langle w_i^{(T_1)}, \mathbf{M}_j \rangle|^2 = \left(1 + \eta \frac{C_z}{d}\right)^{2T_1} \left(\frac{\langle w_i^{(0)}, \mathbf{M}_j \rangle + \langle v_i^{(0)}, \mathbf{H}_j \rangle}{2}\right)^2$$

$$\leq \left(1 + \eta \frac{C_z}{d}\right)^{2T_1} \cdot \frac{c_2 \log d}{d} \cdot \frac{\|\mathbf{M}\mathbf{M}^\top w_i^{(0)}\|_2^2 + \|\mathbf{H}\mathbf{H}^\top v_i^{(0)}\|_2^2}{4} \tag{155}$$

$$= \frac{c_2 \log d}{d} \cdot \frac{\|\mathbf{M}\mathbf{M}^\top w_i^{(T_1)}\|_2^2 + \|\mathbf{H}\mathbf{H}^\top v_i^{(T_1)}\|_2^2}{2}$$

$$\leq \frac{\log d}{d} \cdot \frac{\|w_i^{(T_1)}\|_2^2 + \|v_i^{(T_1)}\|_2^2}{2}$$

$$|\langle w_i^{(t+1)}, \mathbf{M_j^\perp} \rangle|^2 \leq O(\tfrac{1}{d_1}) \frac{\|w_i^{(T_1)}\|_2^2 + \|v_i^{(T_1)}\|_2^2}{2}$$

## I.2   Phase II:

The Phase II of ITCP on Synthetic Data is defined as the training iterations $T_1 \leq t \leq T_2$, where $T_2 - T_1 = \Theta\left(\frac{d \log d}{\eta}\right)$ is the iteration.

We set $b_i^{(t)} = \sqrt{\frac{\log d}{d} \cdot \frac{\|w_i^{(T_1)}\|_2^2 + \|v_i^{(T_1)}\|_2^2}{2}}$ and $b_i^{(t+1)} = (1 + \frac{\eta}{d})b_i^{(t)}$ until all $\|\|w_i^{(T_2)}\|_2 \geq \Omega(d)\|w_i^{(T_1)}\|_2$. In this phase, the weights $(w_i, v_i)$ will mostly ignore the features $\mathbf{M}_j$, $\mathbf{H}_j$ if $i \notin S_{j,\text{sure}}$ and the noise features $\mathbf{M}^\perp$, $\mathbf{H}^\perp$, and learn to emphasize the features $\mathbf{M}_j$, $\mathbf{H}_j$ if $i \in S_{j,\text{sure}}$.

For $i \in S_{j,\text{sure}}$, using Lemma C.4, the following holds with high probability $1 - e^{-\Omega(d_1)}$ when $T_1 < t \leq T_2$:

$$\left|\langle w_i^{(t)}, \xi \rangle\right|^2 \leq O\left(\frac{\left\|w_i^{(t)}\right\|_2^2}{d^{1+c_0}}\right) < b_i^{(t)} \tag{156}$$

Under the assumption that, with high probability, the indicator function satisfies the condition when $t = T_1$:

$$\mathbf{1}_{\left|\left\langle w_i^{(t)}, x_p \right\rangle\right| \geq b_i^{(t)}} \cdot \mathbf{1}_{\left|\left\langle v_i^{(t)}, y_p \right\rangle\right| \geq b_i^{(t)}} = 1, \tag{157}$$

we can ensure that:

$$\mathbb{E}\left[z_x^j z_y^j \cdot \mathbf{1}_{\left|\left\langle w_i^{(t)}, x_p \right\rangle\right| \geq b_i^{(t)}} \cdot \mathbf{1}_{\left|\left\langle v_i^{(t)}, y_p \right\rangle\right| \geq b_i^{(t)}}\right] = \frac{C_z}{d}. \tag{158}$$

The weight dynamics for $|\langle w_i^{(t+1)}, \mathbf{M}_j \rangle|$ can be expressed as:

$$|\langle w_i^{(t+1)}, \mathbf{M}_j \rangle| = \left(1 + \eta \frac{C_z}{d}\right)\left(\frac{\langle w_i^{(t)}, \mathbf{M}_j \rangle + \langle v_i^{(t)}, \mathbf{H}_j \rangle}{2}\right). \tag{159}$$

Given that $\left(1 + \eta \frac{C_z}{d}\right) > \left(1 + \frac{\eta}{d}\right)$, and $\frac{\langle w_i^{(t)}, \mathbf{M}_j \rangle + \langle v_i^{(t)}, \mathbf{H}_j \rangle}{2} > b_i^{(t)}$, it follows that:

$$|\langle w_i^{(t+1)}, \mathbf{M}_j \rangle| > b_i^{(t+1)}. \tag{160}$$

Thus, with high probability, for $t \leq T_2$, we have:

$$\mathbf{1}_{\left|\left\langle w_i^{(t)}, x_p \right\rangle\right| \geq b_i^{(t)}} \cdot \mathbf{1}_{\left|\left\langle v_i^{(t)}, y_p \right\rangle\right| \geq b_i^{(t)}} = 1. \tag{161}$$

so for $T_1 < t \leq T_2$ we have

$$|\langle w_i^{(t+1)}, \mathbf{M}_j \rangle| = \left(1 + \eta \frac{C_z}{d}\right)^t \left(\frac{\langle w_i^{(T_1)}, \mathbf{M}_j \rangle + \langle v_i^{(T_1)}, \mathbf{H}_j \rangle}{2}\right) \tag{162}$$

For $i \notin S_{j,\text{sure}}$, the projection of weights onto a generic feature $\xi$ at iteration $T_1$ satisfies:

$$\Pr\left(\mathbf{1}_{\left|\langle w_i^{(t)}, x_p\rangle\right| \geq b_i^{(t)}} \cdot \mathbf{1}_{\left|\langle v_i^{(t)}, y_p\rangle\right| \geq b_i^{(t)}} = 1\right) \leq o\left(\frac{1}{d}\right). \tag{163}$$

We can ensure that:

$$\mathbb{E}\left[z_x^j z_y^j \cdot \mathbf{1}_{\left|\langle w_i^{(t)}, x_p\rangle\right| \geq b_i^{(t)}} \cdot \mathbf{1}_{\left|\langle v_i^{(t)}, y_p\rangle\right| \geq b_i^{(t)}}\right] = o\left(\frac{1}{d^2}\right). \tag{164}$$

The weight dynamics for $|\langle w_i^{(t+1)}, \mathbf{M}_j\rangle|$ can now be expressed as:

$$|\langle w_i^{(t+1)}, \mathbf{M}_j\rangle| = \left(1 + o\left(\frac{\eta}{d^2}\right)\right)\left(\frac{\langle w_i^{(t)}, \mathbf{M}_j\rangle + \langle v_i^{(t)}, \mathbf{H}_j\rangle}{2}\right). \tag{165}$$

Given that $\left(1 + o\left(\frac{\eta}{d^2}\right)\right) < \left(1 + \frac{\eta}{d}\right)$, and $\frac{\langle w_i^{(t)}, \mathbf{M}_j\rangle + \langle v_i^{(t)}, \mathbf{H}_j\rangle}{2} < b_i^{(t)}$, it follows that:

$$|\langle w_i^{(t+1)}, \mathbf{M}_j\rangle| < b_i^{(t+1)}. \tag{166}$$

If $|\langle w_i^{(T_1)}, \mathbf{M}_j\rangle| < b_i^{(T_1)}$, then $|\langle w_i^{(t)}, \mathbf{M}_j\rangle| < b_i^{(t)}$ for $t \leq T_2$. Thus, with high probability, for $t \leq T_2$, we have:

$$\mathbf{1}_{\left|\langle w_i^{(t)}, x_p\rangle\right| \geq b_i^{(t)}} \cdot \mathbf{1}_{\left|\langle v_i^{(t)}, y_p\rangle\right| \geq b_i^{(t)}} = 0. \tag{167}$$

$$|\langle w_i^{(t+1)}, \mathbf{M}_j\rangle| \leq \left(1 + o\left(\frac{\eta}{d^2}\right)\right)^t\left(\frac{\langle w_i^{(T_1)}, \mathbf{M}_j\rangle + \langle v_i^{(T_1)}, \mathbf{H}_j\rangle}{2}\right) \tag{168}$$

There exists $T_2 = \Theta\left(\frac{d \log d}{\eta}\right)$ such that the following conditions hold:

$$\left(1 + \eta\frac{C_z}{d}\right)^{T_2} = \Theta(d), \tag{169}$$

indicating that $|\langle w_i^{(t+1)}, \mathbf{M}_j\rangle|$ for $i \in S_{j,\text{sure}}$ increase iteratively until:

$$\|w_i^{(T_2)}\|_2 \geq \Omega(d)\|w_i^{(T_1)}\|_2 \tag{170}$$

while, for $i \notin S_{j,\text{sure}}$, the updates diminish, such that:

$$\left(1 + o\left(\frac{\eta}{d^2}\right)\right)^{T_2} \leq 1 + o\left(\frac{1}{d}\right), \tag{171}$$

indicating negligible growth in $|\langle w_i^{(t+1)}, \mathbf{M}_j\rangle|$.

Thus we have

$$\begin{aligned}
|\langle w_i^{(T_2)}, \mathbf{M}_j\rangle|^2 &= \|w_i^{(T_2)}\|_2^2 - \sum_{j\in[d], j\notin\mathcal{N}_i} \langle w_i^{(T_2)}, \mathbf{M}_j\rangle^2 - \sum_{j\in[d_1]\setminus[d]} \langle w_i^{(T_2)}, \mathbf{M}_j^\perp\rangle^2 \\
&\geq \|w_i^{(T_2)}\|_2^2 - (1 + o(1))\|w_i^{(T_1)}\|_2^2 - (1 + o(1))\|w_i^{(0)}\|_2^2 \\
&\geq (1 - o(1))\|w_i^{(T_2)}\|_2^2.
\end{aligned} \tag{172}$$

Finally, for $i \notin S_{j,\text{sure}}$, we have:

$$\|w_i^{(T_2)}, \mathbf{M}_j\|_2 \leq (1 + o(\frac{1}{d})) \cdot O\left(\frac{\|w_i^{(T_1)}\|_2}{\sqrt{d}}\right) \leq O\left(\frac{\|w_i^{(T_2)}\|_2}{\sqrt{d}}\right), \tag{173}$$

and for noise components:

$$|\langle w_i^{(T_2)}, \mathbf{M}_j^\perp\rangle|_2 \leq O\left(\frac{\|w_i^{(T_2)}\|_2}{\sqrt{d_1}}\right). \tag{174}$$

We summarize the results when $T_1 < t \leq T_2$ as follows:

1. For $i \in S_{j,\text{sure}}$, the alignment strength satisfies:

$$|\langle w_i^{(T_2)}, \mathbf{M}_j \rangle|^2 > (1 - o(1)) \frac{\|w_i^{(T_2)}\|_2^2 + \|v_i^{(T_2)}\|_2^2}{2} \tag{175}$$

without $j'$ that represents the corresponding spurious alignment feature.

2. For $i \notin S_{j,\text{pot}}$, the alignment strength satisfies:

$$|\langle w_i^{(T_1)}, \mathbf{M}_j \rangle|^2 \leq O(\frac{1}{d}) \cdot \frac{\|w_i^{(T_2)}\|_2^2 + \|v_i^{(T_2)}\|_2^2}{2} \tag{176}$$

3. For $\mathbf{M}_j^\perp$ where $j \in [d_1] \setminus [d]$, we have:

$$|\langle w_i^{(t+1)}, \mathbf{M}_j^\perp \rangle|^2 < O\left(\frac{1}{d_1}\right) \cdot \frac{\|w_i^{(T_2)}\|_2^2 + \|v_i^{(T_2)}\|_2^2}{2}. \tag{177}$$

Similar results also hold for $v_i$.

### I.3   Phase III Convergence of ITCP on Synthetic Data

Similarly to convergence Phase III in ITCP on Raw Data when $T_2 \leq t \leq T_3$, using Eq (27), Eq (30), and Eq (31), the loss function $L$ becomes convex with respect to $w_i$ and $v_i$ independently when $(x_p, y_p)$ and $(x_n, y_n)$ contain the true feature $j$.

We verify that the following inequality holds

$$
\begin{aligned}
L_j(w_i^{(t+1)}, v_i^{(t+1)}) \leq \ & L_j(w_i^{(t)}, v_i^{(t)}) \\
& + \left\langle \nabla L_j(w_i^{(t)}, v_i^{(t)}), \left(w_i^{(t+1)} - w_i^{(t)}, \ v_i^{(t+1)} - v_i^{(t)}\right) \right\rangle \\
& + \frac{l_{i,j}}{2} \left\| \left(w_i^{(t+1)} - w_i^{(t)}, \ v_i^{(t+1)} - v_i^{(t)}\right) \right\|^2
\end{aligned}
\tag{178}
$$

Let $L = \max_{i \in m}(l_{i,j}/(2\tau)) = \Theta(1)$ and $\eta = \frac{1}{L}$ to ensure a monotonic decrease, plug Eq (28) and Eq (29) into Eq (178), we have

$$L_j(w_i^{(t+1)}, v_i^{(t+1)}) \leq L_j(w_i^{(t)}, v_i^{(t)}) - \frac{\eta}{2}\|\nabla L_j(w_i^{(t)}, v_i^{(t)})\|^2. \tag{179}$$

Under our data assumptions for $S_w$ and conclusion in Eq (96), we define $w_i^* = \alpha_{i,j}^* \mathbf{M}_j, v_i^* = \alpha_{i,j}^* \mathbf{H}_j$. Thus, $L_j(w_i^*, v_i^*)$ captures both the alignment with the true feature $\mathbf{M}_j, \mathbf{H}_j$ and the spurious feature $\mathbf{M}_{j'}, \mathbf{H}_{j'}$, representing the minimal loss achievable under the influence of both true and spurious features in the optimization process. Using Eq (81), we know $w_i^{(T_2)} = \Theta(d)$, so $L_j(w_i^*, v_i^*) = -\Theta(d)$.

By the property of smoothness, we have

$$\|\nabla L_j(w_i^{(t)}, v_i^{(t)})\|_2^2 \geq \frac{2}{L}\left(L_j(w_i^{(t)}, v_i^{(t)}) - L_j(w_i^*, v_i^*)\right) \tag{180}$$

Take the telescope sum of from $T_2$ to $T_3$, we have

$$
\begin{aligned}
\frac{1}{T_3 - T_2} \sum_{t=T_2}^{T_3} L_j(w_i^{(t)}, v_i^{(t)}) &\leq L_j(w_i^*, v_i^*) + \frac{L^2 \Delta_0}{T_3 - T_2} \\
&\overset{\diamond}{\leq} L_j(w_i^*, v_i^*) + \Theta(1)
\end{aligned}
\tag{181}
$$

where $\Delta_0 = L_j(w_i^{(T_1)}, v_i^{(T_1)}) - L_j(w_i^*, v_i^*) = \Theta(1)$. In $\diamond$, we use $T_2 = \Theta(d)$, and $L = \Theta(\frac{1}{d})$.

Generalized to every $j \in d$, the same convergence holds for all $i \in S_{j,\text{sure}}$ when $(x_p, y_p)$ and $(x_n, y_n)$ contain feature $j, j'$. For all $(x_p, y_p)$ and $(x_n, y_n)$ in $S_w$, the following inequality holds:

$$\frac{1}{T_3 - T_2} \sum_{t=T_2}^{T_3} L(f^{(T_3)}, h^{(T_3)}) \leq L(f^*, h^*) + \Theta(1). \tag{182}$$

### I.4 Summary

ITCP trained on recaptioned data $\tilde{S}$ proceeds according to Eq. (1). After $T = \Theta(d^2 \log d)$ SGD iterations with batch size $B = \Omega(d)$ and learning rate $\eta = O(1)$, the returned weights $(\widetilde{\mathbf{W}}, \widetilde{\mathbf{V}})$ achieve a contrastive loss that is asymptotically optimal:

$$\frac{\tilde{L}(f_{\widetilde{\mathbf{W}}}, h_{\widetilde{\mathbf{V}}}) - \tilde{L}^*}{\left|\tilde{L}^*\right|} \leq o(1). \tag{183}$$

Each neuron pair $(\tilde{w}_i, \tilde{v}_i)$ in $(\widetilde{\mathbf{W}}, \widetilde{\mathbf{V}})$, for $i \in [m]$, primarily encodes a single aligned feature indexed by a set $\tilde{N}_i \subseteq [d]$, with $|\tilde{N}_i| = 1$. Specifically, we have:

$$\begin{aligned}
\tilde{w}_i &= \sum_{j \in \tilde{N}_i} \tilde{\alpha}_{i,j} \mathbf{M}_j + \sum_{j \in [d] \setminus \tilde{N}_i} \tilde{\beta}_{i,j} \mathbf{M}_j + \sum_{j \in [d_1] \setminus [d]} \tilde{\gamma}_{i,j} \mathbf{M}_j^\perp, \\
\tilde{v}_i &= \sum_{j \in \tilde{N}_i} \tilde{\alpha}_{i,j} \mathbf{H}_j + \sum_{j \in [d] \setminus \tilde{N}_i} \tilde{\beta}_{i,j} \mathbf{H}_j + \sum_{j \in [d_1] \setminus [d]} \tilde{\gamma}_{i,j} \mathbf{H}_j^\perp,
\end{aligned} \tag{184}$$

where $\tilde{\alpha}_{i,j}^2 = \Theta(\|\tilde{w}_i\|_2^2 + \|\tilde{v}_i\|_2^2)$, and the residual terms satisfy $\tilde{\beta}_{i,j}/\tilde{\alpha}_{i,j} \leq O(1/\sqrt{d})$, $\tilde{\gamma}_{i,j}/\tilde{\alpha}_{i,j} \leq O(1/\sqrt{d_1})$.

Moreover, for every feature index $j \in [d]$, there exists an $\Omega(1)$ many of neurons $i \in [m]$ such that $\tilde{N}_i = \{j\}$, indicating that each semantic concept is distinctly captured by dedicated neuron pairs.

## J  Downstream Task

We consider the same zero-shot classification task as in Section 3.4, where the image $x$ and the class-wise text prompts $\{y_k\}_{k=1}^K$ are given. Each prompt $y_k$ corresponds to one of $K$ class labels, and the goal is to classify $x$ into the class with the best matching prompt.

Each text prompt $y_k$ is generated as:

$$y_k = \mathbf{H} z'_{y_k} + \xi_{y_k}, \quad \|z'_{y_k}\|_0 = \Theta(1), \quad \|z'_{y_k}\|_{\max} = \Theta(1). \tag{185}$$

Each test image $x$ is generated as:

$$x = \mathbf{M}' z'_x + \xi_x, \quad \|z'_x\|_0 = \Theta(1), \quad \|z'_x\|_{\max} = \Theta(1), \tag{186}$$

where $\mathbf{M}' = \mathbf{M}\mathbf{P}_1$, and

$$\max_{i,j} |(\mathbf{P}_1)_{ij} - \delta_{ij}| \leq O(1/\sqrt{d}). \tag{187}$$

If $x$ belongs to class $k$, then:

$$\left\|(z'_x)^\top z'_{y_k}\right\|_2 > \left\|(z'_x)^\top z'_{y_{k'}}\right\|_2, \quad \forall k' \neq k. \tag{188}$$

Using Eq. (96) and Eq. (144), let $f(x)$ and $h(y)$ represent the image encoder and text encoder of ITCP on raw data, respectively. Given a data sample $x$ containing $\mathbf{M}_j$ and $y$ containing $\mathbf{H}_{j'}$, where $j'$ is the spurious feature corresponding to $j$, it holds with high probability that:

$$\left\langle \frac{f(x)}{\|f(x)\|_2}, \frac{h(y)}{\|h(y)\|_2} \right\rangle = \Theta(1). \tag{189}$$

This result implies that the image and text encoders of ITCP on raw data struggle to distinguish between features $j$ and $j'$, leading to misclassification caused by spurious correlations.

However, using Eq. (175) and Eq. (176), let $\tilde{f}(x)$ and $\tilde{g}(y_k)$ denote the image and text encoders of ITCP on recaptioned data. Given $x$ containing $\mathbf{M}_j$ and $y$ containing spurious $\mathbf{H}_{j'}$, it holds with high probability $1 - \Theta\left(\frac{1}{d}\right)$ that:

$$\left\langle \frac{\tilde{f}(x)}{\|\tilde{f}(x)\|_2}, \frac{\tilde{g}(y)}{\|\tilde{g}(y)\|_2} \right\rangle \leq \Theta\left(\frac{1}{d}\right). \tag{190}$$

This result implies that the image and text encoders of ITCP on synthetic data are capable of effectively distinguishing the true feature from the spurious feature.

Because $K = \Theta(1)$ and $\|z_{y_k}\|_0 = \Theta(1)$, we only have constant class classification and constant features in images. Thus, we have:

1. For the image encoder $f(x)$ and text encoder $h(y_k)$ of ITCP on raw data:

$$\Pr\left(\arg\max_k \langle f(x), h(y_k)\rangle = k_x\right) = 1 - \Theta(1), \tag{191}$$

2. For the image encoder $\tilde{f}(x)$ and text encoder $\tilde{g}(y_k)$ of ITCP on synthetic data:

$$\Pr\left(\arg\max_k \langle \tilde{f}(x), \tilde{g}(y_k)\rangle = k_x\right) = 1 - o(1). \tag{192}$$

