# OpenReview forum: "Contrastive Learning with Data Misalignment: Feature Purity, Training Dynamics and Theoretical Generalization Guarantees"
_NeurIPS.cc/2025/Conference — NeurIPS 2025 poster_

### Official Review · Reviewer_kXL9 · 2025-06-21

**Clarity:** 2
**Significance:** 2
**Originality:** 3
**Rating:** 4
**Confidence:** 3

**Summary:**

This paper studies the following text-image contrastive learning setting:
- We are given a clean dataset $S_h$ and a noisy dataset $S_w$.
- We first train a text embedding $h(y)$ and an image embedding $f(x)$ using the union dataset $S = S_h \cup S_w$.
- Meanwhile, we train a text generator $G: x \mapsto \hat{y}$ only using the clean dataset $S_h$. Here $x$ is an image and $\hat{y}$ is the generated text caption.
- We filter the data as follows: For each $x$, if the cosine similarity between $f(x)$ and $g(\hat{y})$ is greater than that between $f(x)$ and $g(y)$, we replace $y$ with $\hat{y}$. Then, we train two new embeddings $h'$ and $f'$ on this new filtered dataset.

I suppose that one can continue this process iteratively, but it is not considered in the paper.

The authors consider a very specific setting of data and model. Under this very specific setting, they prove that the embeddings trained on the original union dataset learn spurious features, whereas the embeddings trained on the new filtered dataset are better. They empirically validate their theoretical result with synthetic experiments.

**Questions:**

I have some questions about the assumptions made in this paper:
1. Why do you assume that "the noise magnitude can dominate the signal" (line 162)? Why is it a reasonable assumption? Will your results still hold if the noise magnitude is much lower?
2. In line 172 you mentioned that $|S_h| = \Theta (d^2)$ is much smaller than $|S_w|$. Where do you state this in your assumptions or results? I couldn't see how your results depend on the relative size of the two datasets. Or is it true that your results are independent of the relative size?
3. In assumption 3.4, what is $z_{y_p}^j$ if $z_{x_p}^j = 0$? And why do you only consider a single pair of $(j, j')$ (line 186)? Is it true that you are assuming that $z_{y_p}$ and $z_{x_p}$ differ by at most two dimensions?
4. In line 195, why should $W$ and $V$ be initialized at $\bar{W}$ and $\bar{V}$? Why is this initialization good provided that $\bar{W}$ and $\bar{V}$ are learned on the noisy union dataset? What would happen if one uses standard initialization?

**Ethical Concerns:**

["NO or VERY MINOR ethics concerns only"]

**Final Justification:**

The rebuttal addresses most of my questions. I have raised my score from 3 to 4.

**Limitations:**

Limitations are not discussed in the conclusion, though the authors claimed to do so in the checklist.

**Quality:**

2

**Strengths And Weaknesses:**

Overall, the setting of this paper is clear, though the paper is not particularly easy to follow. If we have a clean dataset and a noisy dataset, then it is reasonable to use the clean dataset to purify the noisy dataset before using it. Proving that such purification helps learn better features is useful but not extremely significant. I am more interested in the mechanism behind how such purification helps. Instead of the theorems, I am more interested in the "key step" that explains why the filtering reduces the spurious features in the dataset. I couldn't find it in the main body of the paper. I might have missed it but I suggest the authors highlight this key step if it is already in the main body.

The authors only compare the embeddings trained on the noisy dataset with the embeddings trained on the filtered dataset, but I think there are more embeddings that should be compared:
- Trained on the clean dataset $S_h$ alone
- Trained on the clean dataset $S_h$ plus the images $S_w$ with generated text only; that is, we never use the text in $S_w$

The second setting is a typical semi-supervised learning setting, and there are a bunch of theoretical papers on the effectiveness of semi-supervised learning. This setting is also not limited to vision-language contrastive learning. Pseudo-labeling is a very common technique in semi-supervised learning. Regarding the setting in this paper, I am especially curious whether the text information in $S_w$ is useful at all, since the authors consider "high-noise conditions" (line 165).

Moreover, the setting considered in this paper is very specific. There are a number of very strong assumptions, making it questionable whether the proved results still hold for a slightly more general setting. I have some specific questions on the assumptions (see below).

In summary, I lean towards rejection because I am not sure how useful the proved results are, given that the setting studied in this paper is very specific and limited. Moreover, constructing a specific setting where using a generator trained on clean data (to generate pseudo labels, filter noisy data, etc.) helps improve the performance is not really hard, and has in fact been studied a lot in the context of semi-supervised learning. Instead, how such a generator could help leverage the weak signal in the noisy text captions from the noisy dataset is a more interesting question.

I am willing to raise my score if the AC or the other reviewers find the results in this paper meaningful, though personally I am not a big fan of "theoretical results" proved by constructing extremely specific examples.

---

> ### Author Rebuttal · Authors · 2025-07-31
>
> We thank the reviewer for the valuable time in the evaluation.
> ## Response to Weakness 1: Clarification on the specific setting and semi-supervised learning
>
> We clarify the data model adopted in this paper is standard and not designed for our specific setting. Both our learning framework and theoretical analysis are different from those in semi-supervised learning and cannot be regarded as extensions of existing work in that literature.
>
> **The sparse-coding model we use is standard in theory and widely used to represent images and texts [R1, R2, R3].** Shallow networks remain SOTA for analyzing training dynamics and generalization [R4, R5, R6], as deeper models are harder to study. The one-hidden-layer model is still SOTA for theoretical analysis of single-modal contrastive learning.
>
> **Our contrastive setting follows the four-stage structure used by many frameworks.** E.g., BLIP applies the recaption–filter cycle iteratively; LaCLIP does it once. We do not aim to propose a new training method but instead give theoretical foundations for these existing strategies. This paper focuses on theory, not algorithmic novelty.
>
> **Our setting differs from semi-supervised learning [R7, R8, R9], which assumes labeled data is noiseless or nearly clean, and uses unimodal classifiers.** Our “clean” pairs are well-aligned but can be highly noisy (lines 162–169). We clarify that "clean" means aligned features, not noise-free. So standard semi-supervised theory cannot provide guarantees in our setting. In addition, contrastive learning aligns two modalities without labels, unlike semi-supervised learning with true or pseudo labels. This alignment requires analyzing coupled encoders under a non-convex contrastive loss, which is far harder than the single-encoder supervised case. As far as we know, no existing work gives full dynamic analysis in this multimodal setting. Should the reviewer request it, we are happy to provide further comparison or elaboration in the next stage of discussion.
>
> ## Response to Weakness 2: Clarification of the Key Step in Removing Spurious Features
>
> To avoid confusion in terminology: in our paper, *recaptioning* means generating synthetic text using \( G(x) \), and *filtering* refers to replacing some raw text with this synthetic text.
>
> **Both recaptioning and filtering are essential.** Appendix B.1 (line 626) shows that fine-tuning the decoder \( G(x) \) on clean data \( S_h \) amplifies alignment with true features exponentially while keeping alignment with spurious features negligible. Thus, the generated captions retain relevant semantics and suppress spurious correlations.
>
> Filtering uses these purified captions to replace misaligned ones, further eliminating spurious content. As shown in Theorem 4.3, the resulting dataset contains significantly fewer spurious features and more well-aligned pairs than the original data.
>
> ## Response to Weakness 3: On Embeddings from Noisy vs. Filtered Data
>
> Training only on $S_h$ reflects limited supervision in practice—$S_h$ is small and high-quality, while $S_w$ provides scale. High-capacity VLMs cannot be trained effectively on $S_h$ alone. Empirically, using only $S_h$ yields downstream accuracy 0.86, far below the $>0.90$ when including $S_w$ (Fig. 1(c)), so we exclude this baseline.
>
> For the setting $S_h \cup \{(x, \hat{y}) \mid x \in S_w\}$, our theory directly applies. As shown in Lines 1053–1054, we compare cosine similarity between filtered ($\langle z'_x, \hat{z}_y \rangle$) and fully replaced ($\langle z'_x, z'_y \rangle$) captions, selecting the higher similarity. We prove filtering is at least as effective as full replacement, and empirically performs better.
>
> Prior work ([R10], Fig. 2) studies how varying the ratio of synthetic/raw captions affects performance. Rather than repeating such ablations, our focus is to explain theoretically why recaptioning plus filtering improves feature purity and downstream results.
>
> ## Response to Question 1: Noise Dominates Signal
>
> **The assumption that noise dominates the signal (line 160) is not necessary for our main theorems, but it helps to demonstrate that our main theorems work even in some extreme high-level noise cases.** Our results remain valid in the low-noise regime without any significant modification.
>
> We highlight the high-noise case to emphasize that even under severe noise, our one-hidden-layer contrastive model can still learn useful features. The lower bound in Eq(160) is only to show robustness in this worst-case regime.
>
> This assumption is motivated by prior work on supervised learning [R6], to justify the improved performance of nonlinear networks over linear models under high-level noise. Since our goal is not to compare linear and nonlinear models, we do not expand on this in the main text. To avoid confusion, we will remove the lower bound in (160) and add a footnote that our guarantees apply in both low- and high-noise regimes.
>
> ## Response to Question 2: Clarification on Whether the Results Depend on the Relative Size of $S_h$ and $S_w$
>
> In our theoretical analysis, we characterize the required batch size $B$ and the required number of training iterations for the model to converge. As we assume independence of the samples in different batches, the sample size requirement is at least $T \times B$ (iterations times batch size).
>
> (i) Pretraining ITCP with contrastive loss (Theorem 4.1, line 241) uses $T = \Theta(d^2 \log d)$ and $B = \Omega(d)$. This is Stage (S1) on raw data (line 102) and Stage (S4) on filtered data (line 118) with $S = S_h \cup S_w$ and $\tilde{S} = S_h \cup \tilde{S}_w$.
>
> (ii) Finetuning the decoder $G$ for recaptioning (Theorem 4.3, line 259) uses $T = \Theta(d \log d)$ and $B = \Omega(d)$. This is Stage (S2) (line 107), where only pairs in $S_h$ are used to finetune the image-grounded decoder.
>
> VLM pretraining typically needs far more samples than decoder finetuning, so we model $|S_w| \gg |S_h|$ to reflect practice, where human-annotated $S_h$ is small and web data $S_w$ is abundant.
>
> The results will depend on sample size; specifically, larger samples will lead to faster convergence and smaller generalization error. Our intention was to study the worst-case scenario, aiming to identify the minimum number of samples needed for successful training. Therefore, we did not specifically analyze cases with varying sample sizes.
>
> ## Response to Question 3: Assumption 3.4
>
> When $z_{x_p}^j = 0$, $z_{y_p}^j$ can still equal 1 if a different feature $j' \neq j$ is active in the image and spuriously mapped to index $j$ in text. So:
>
> $$\Pr(z_{y_p}^j = 1 \mid z_{x_p}^j = 0) = \Pr(z_{x_p}^{j'} = 1)\\Pr(z_{y_p}^j = 1 \mid z_{x_p}^{j'} = 1) = \frac{C_z}{d}C_s$$
>
> where \( C_z/d \) is the activation probability of any other feature \( j' \) in the image and \( C_s \) is the spurious alignment rate. Thus,
>
> $$\Pr(z_{y_p}^j = 0 \mid z_{x_p}^j = 0) = 1 - \frac{C_z}{d} C_s$$
>
> We assume a single spurious feature $j'$ per $j$ (forming $d/2$ disjoint pairs) only to simplify presentation. This can be generalized to $K-1$ spurious correlates per $j$, making Eq. (34) a $2K \times 2K$ matrix (line 701) without altering the insight.
>
> Theorem 4.2 and its guarantees rely only on the total spurious mass (bounded by $C_s$), not on how many $j'$ exist per $j$. So the core mechanism is robust to such generalization.
>
> Both $z_x$ and $z_y$ are $d$-dimensional sparse latent vectors (line 161). Spurious misalignment can occur on multiple dimensions, not just one pair. If $z_x$ has $k$ nonzeros, then $z_x$ and $z_y$ can differ on up to $2k$ coordinates.
>
> ## Response to Question 4: Clarification on Initialization at $\bar{W}, \bar{V}$
>
>  Although $\bar{W}, \bar{V}$ are learned on the noisy dataset $S = S_h \cup S_w$, Theorem 4.2 shows that each neuron pair learns a mixed representation containing both relevant and spurious features. Despite the contamination, such weights preserve partial structure and are substantially better than random initialization.
>
> When fine-tuning the decoder $G$ on clean data $S_h$, initializing from $(\bar{W}, \bar{V})$ allows the model to purify relevant features, yielding accurate synthetic captions. This practice is standard in state-of-the-art VLMs such as BLIP [R11], where the decoder is initialized from a pretrained VLM and then fine-tuned on human-annotated image-text pairs.
>
> It is possible to use standard random initialization, but that would require a larger set of clean data to achieve the same level of representation purity of the learned decoder.
>
> ## References
> [R1] Yang et al., "Linear Spatial Pyramid Matching Using Sparse Coding for Image Classification," CVPR 2009.
> [R2] Yang et al., "Robust Sparse Coding for Face Recognition," CVPR 2011.
> [R3] Arora et al., "A Latent Variable Model Approach to PMI-based Word Embeddings," TACL 2016.
> [R4] HaoChen et al., "Provable Guarantees for Self-Supervised Contrastive Learning," NeurIPS 2021.
> [R5] Wen and Li, "Toward Understanding the Feature Learning Process of Self-Supervised Contrastive Learning," ICML 2021.
> [R6] Allen-Zhu and Li, "Feature Purification: How Adversarial Training Performs Robust Deep Learning," FOCS 2022.
> [R7] D.-H. Lee, "Pseudo-label: The Simple and Efficient Semi-Supervised Learning Method for Deep Neural Networks," ICML 2013.
> [R8] Zhang, Shuai, et al., "How Does Unlabeled Data Improve Generalization in Self-Training? A One-Hidden-Layer Theoretical Analysis," ICLR 2022.
> [R9] Wei, Colin, et al., "Theoretical Analysis of Self-Training with Deep Networks on Unlabeled Data," ICLR 2021.
> [R10] Nguyen, et al., "Improving Multimodal Datasets with Image Captioning," NeurIPS 2023.
> [R11] Li, Junnan, et al., "BLIP: Bootstrapping Language-Image Pre-training for Unified Vision-Language Understanding and Generation," ICML 2022.

---

> > ### Comment · Reviewer_kXL9 · 2025-08-02
> >
> > I thank the authors for the rebuttal. I don't have further questions at this point. I have raised my score to 4 and will discuss with my fellow reviewers.

---

> > > ### Author Response · Authors · 2025-08-07
> > >
> > > We sincerely thank you for your time, thoughtful evaluation, and positive rating. We are glad that our rebuttal addressed your concerns. We appreciate your support and look forward to further discussion during the decision process.

---

### Official Review · Reviewer_CyKG · 2025-06-30

**Clarity:** 3
**Significance:** 3
**Originality:** 4
**Rating:** 5
**Confidence:** 2

**Summary:**

Contrastive learning has been very successful for learning discriminative representations from image-text pairs. However, how contrastive learning can improve discriminative representations has not been well investigated, especially from theoretical perspectives. This paper theoretically and empirically analyzes contrastive learning with misaligned image-text pairs, showing that spurious alignment leads to feature impurity and generalization degradation. Under a sparse coding model, the authors demonstrate that filtering out misaligned data improves purified representations, and provide SGD convergence guarantees under such noisy supervision.

**Questions:**

Extension to Deeper/Transformer Architectures:
Your model uses a 2-layer ReLU network with symmetric activation for tractability.
→ Do you expect similar dynamics in deeper or transformer-based encoders? Can empirical observations support this? What are the bottlenecks in theoretical extension?

Clarification on Assumptions and Presentation Structure:
The paper starts with Assumption 3.2 without presenting Assumption 3.1, which raises concerns about completeness.
→ Could you clarify if Assumption 3.1 was omitted intentionally or inadvertently? Moreover, the overall structure of the paper may benefit from better separation between theoretical analysis, empirical validation, and broader discussion. Have you considered re-structuring to better align motivation, analysis, and conclusions?

Feature Purity – Monitoring and Conceptual Clarity:
The notion of feature purity plays a central role in both the theory and experiments.
→ Could you provide more precise criteria or visualizations for how feature purity is measured and interpreted in practice? Would it be beneficial to isolate this concept in a dedicated section (e.g., a definition box or diagnostic figure)
to aid clarity and reproducibility?

Impact on Fine-tuning and Transfer Learning:
Your results emphasize zero-shot generalization.
→ Do you expect feature purity to remain beneficial in downstream fine-tuning or domain adaptation scenarios? How might it influence representations during gradient-based adaptation?

Estimability and Practical Relevance of Cs (Spurious Alignment Rate)
Your theory hinges on a spurious alignment probability Cs that determines the noise level in supervision.
→ Can Cs be estimated or approximated in real-world settings, e.g., via proxy metrics on semantic similarity or caption quality? If not, how sensitive are your theoretical guarantees to misspecification of this parameter?

**Ethical Concerns:**

["NO or VERY MINOR ethics concerns only"]

**Final Justification:**

Sorry for the delayed response, and thank you for the detailed and concrete review. I have carefully read the authors’ rebuttal and considered the clarifications provided. I appreciate the effort the authors put into addressing the concerns raised, and I am satisfied with the explanations. I have engaged in the reviewer discussion and documented my reasoning here. I also acknowledge that my review and conduct are subject to the Responsible Reviewing initiative, and I submit this Final Justification accordingly.
That said, I would like to note that this submission is mathematically demanding and outside my primary area of expertise. While I did my best to evaluate the work, I could not carefully verify all technical details, and therefore I assign a lower confidence level to my assessment.

**Limitations:**

- Real-world robustness: Clarify how the proposed theory holds under real-world caption noise distributions, especially those involving semantic ambiguity rather than total irrelevance.
- Estimating spurious alignment rate (Cs): Since Cs is central to theoretical guarantees, discussing methods to estimate Cs or proxy metrics in practice would enhance the applicability of the work.

**Paper Formatting Concerns:**

There is no formatting issues in this paper.

**Quality:**

3

**Strengths And Weaknesses:**

Strengths:
- First theoretical treatment of data misalignment in contrastive learning for VLMs.
- Provides provable guarantees of SGD convergence under noisy supervision. Introduces the concept of feature purity and connects it to generalization performance.
- Experimental results strongly support the theoretical claims, including on real-world datasets.

Weaknesses:
- The theoretical model uses a simplified 2-layer network and does not directly extend to real-world architectures like transformers.
- Recaptioning mechanism G(x) is not theoretically modeled or analyzed — despite its empirical impact.
- The notion of spurious alignment (Cₛ) is theoretically assumed but not estimated or measured in real datasets.
- Random shuffling texts in image-text pairs in the experiments may not reflect real situations since image-text pairs may not that far as random associations. More realistic settings would be to shuffle with similar texts probabilictically.

---

> ### Author Rebuttal · Authors · 2025-07-31
>
> We thank the reviewer for the valuable time in the evaluation.
>
> ## Response to Weakness 1: The theoretical model uses a simplified 2-layer network and does not directly extend to real-world architectures like transformers
>
> While the model studied in this paper is a simplified version of real-world applications, we believe that analyzing the training dynamics of a one-hidden-layer network with nonlinear activation represents the most advanced setting currently allowing a comprehensive theoretical analysis of contrastive learning. Table 1 in the paper shows that our two-encoder architecture achieves the highest modelling fidelity among existing theoretical works published in recent top ML conferences:
>
> **Contrastive-learning theory**
> [R3] analyses only linear encoders.
> [R4] extends to one hidden layer but still studies a single encoder for one modality.
> Our work is the first to handle two nonlinear encoders, misaligned image–text multi-modal data, and zero-shot evaluation.
>
> **Existing theory for transformers is limited to supervised tasks or linear approximations:**
> [R1, R2] studies model editing in one-hidden-layer transformers under supervised objectives.
>
> In summary, we believe our work has addressed more complicated and realistic models compared to prior studies. Extending the analysis beyond one-hidden-layer ReLU networks would require fundamentally different tools to handle the highly non-convex loss landscape. Therefore, we leave this direction for future works.
>
> **Empirical validation on transformer VLMs**
> Notably, our empirical experiments are all conducted using 12-layer ViT-based encoders, and their training behavior aligns well with the theoretical predictions, providing strong empirical support for the theoretical insights:
>
> 1. **Recaption improves performance.** LaCLIP achieves higher accuracy and silhouette scores than CLIP (Table 2).
>
> 2. **Purified neurons enhance performance.** Lowest-similarity neurons yield best accuracy; highest-similarity worst (Figure 2a–c,e–g).
>
> 3. **Misalignment reduces purity.** Caption shuffling increases neuron similarity proportionally, matching Theorem 4.2 (Figure 2d,h).
>
> These observations show that conclusions proved for a simplified network transfer qualitatively to state-of-the-art transformer VLMs such as CLIP and LaCLIP.
>
> Extending our analysis to deeper or transformer-based architectures would require substantial technical advances. Beyond the nonlinearity introduced by ReLU activations, such extensions must address the multiplicative interactions of the attention mechanism and the highly nontrivial behavior of the softmax function. These components introduce dense, dynamic coupling across features, posing significant challenges for any rigorous analytical framework.
>
> ## Response to Weakness 2: Recaptioning mechanism is not theoretically modelled or analysed
>
> **We theoretically analyze a simplified recaptioning mechanism in Appendix G and prove it enhances feature purity.** A simplified recaptioning mechanism $G(x)$ is actually theoretically analyzed in Appendix G, beginning at line 973. We could not include that in the main paper due to space limit, resulting in this confusion. We actually theoretically prove that fine-tuning the decoder on clean data $S_h$ can make the neurons become aligned with true features, i.e., their alignment is increased exponentially see Eq.(130), while spurious alignments remain negligible see Eq.(132). This theoretical analysis of a simplified captioning step serves as a prototype explanation, offering insight into the underlying mechanisms of recaptioning in practice. It supports the effectiveness of $G(x)$ in preserving relevant features and suppressing spurious ones, thereby enhancing feature purity.
>
> ## Response to Weakness 3: The notion of spurious alignment (Cₛ) is theoretically assumed but not estimated or measured in real datasets.
>
> Same as Response to Question 5.
>
> ## Response to Weakness 4: Additional Experiment - Similarity-Based Shuffling
>
> Random shuffling may exaggerate misalignment. We therefore ran a more realistic variant: for each image we located the ten captions with highest similarity and randomly selected one of them for shuffling. Feature purity was measured by average absolute cosine similarity of all neuron pairs, identical to Figure 2(d). Results are shown below:
>
> |$C_s$|Random shuffle|Similarity shuffle|
> |---|---|---|
> |0.0|0.0562|0.0562|
> |0.1|0.0762|0.0693|
> |0.3|0.0875|0.0760|
> |0.5|0.1013|0.0847|
> |0.8|0.1270|0.0999|
>
> Although similarity-based shuffling yields lower absolute cosine similarity than fully random shuffling, feature purity still degrades as $C_s$ increases. Hence realistic misalignment produces the same qualitative trend, reinforcing our theoretical conclusions.
>
> ## Response to Question 1: Extension to Deeper/Transformer Architectures
>
> Same as Response to Weakness 1.
>
> ## Response to Question 2: Clarification on Assumptions and Structure
>
> Sorry for the confusion. The perceived omission of Assumption 3.1 appears to stem from a misunderstanding. Due to the latex numbering system used in this paper, Definitions and Assumptions are numbered together. Assumption 3.2 appears after Definition 3.1, thus obtaining this numbering. No assumption has been omitted.
>
> We will provide a clearer separation in the indexing and organization of the theoretical framework and empirical validation to improve the paper's clarity.
>
> ## Response to Question 3: Feature Purity – Monitoring and Conceptual Clarity
>
> We thank the reviewer for raising this point. The formal definition for measuring feature purity can be found in line 340. Specifically, we compute the absolute cosine similarity between every pair of learned neurons $v_j$ and $v_{j'}$ as $|\langle v_j, v_{j'} \rangle| / (\|v_j\| \|v_{j'}\|)$ for all $j, j' \in \{1,2,\dots,512\}$. A lower average value means neurons are more orthogonal whereas a larger value indicates feature mixing.
>
> Figures 2 apply this metric: (a–c,e–g) show that selecting high-purity (lower average absolute cosine similarity) neurons maximises zero-shot scores, whereas (d,h) confirm that higher shuffle rates reduce purity (average absolute cosine similarity increases)  as theory predicts.
>
> ## Response to Question 4: Impact on Fine-tuning and Transfer Learning
>
> Purified representations, which emphasise true semantic features and suppress spurious correlations, offer a stronger starting point for gradient-based adaptation. In principle, this lets fine-tuning specialise relevant features more efficiently instead of spending capacity to correct entangled ones, thereby reducing sample complexity and accelerating convergence.
>
> A systematic empirical study of fine-tuning and domain-adaptation benefits is beyond the scope of the current work; we leave a detailed exploration of these transfer-learning effects as future research.
>
> ## Response to Question 5: Estimability and Practical Relevance of $C_s$
>
> **While estimating $C_s$ in real datasets remains challenging, our theory provides meaningful qualitative insights across a broad regime.** The spurious alignment rate $C_s$ plays a central role in our theoretical framework by quantifying the likelihood that an image feature appears spuriously in low-quality image-text pairs. While our theorems assume that $C_s \in \left( \omega(1/\log d), 1/2 \right)$, they remain valid throughout this broad regime—from low spurious alignment levels $\omega(1/\log d)$, up to moderate noise levels approaching $1/2$. Across this range, our analysis guarantees that recaptioning significantly improves alignment and downstream performance.
>
> That said, we acknowledge that estimating or approximating $C_s$ in real-world datasets remains a challenging open problem. It may be possible to develop proxy metrics based on semantic similarity scores or caption relevance to approximate the degree of spurious feature alignment, but we leave this for future empirical work. Despite the absence of a direct estimator, the qualitative implications of our theory remain meaningful and broadly applicable in practical VLM training scenarios.
>
> ## Response to Limitation 1: Real-world robustness
>
> Our theory considers the worst-case noise of totally irrelevant misalignment because semantic ambiguity is difficult to model analytically, and, to our knowledge, no existing theory fully addresses this form of ambiguity.
>
> Despite this simplification, we validated the key predictions on real datasets such as CC12M and LAION-400M and LaCLIP, which naturally contain ambiguous captions. The empirical results in Table 2 and Figure 2 align with our theory.
>
> ## Response to Limitation 2: Estimating spurious alignment rate
>
> Same as Response to Question 5.
>
> ## References
> [R1] Li et al., "When Is Task Vector Provably Effective for Model Editing? A Generalization Analysis of Nonlinear Transformers," NeurIPS 2023.
> [R2] Huang, Yu, et al. "A Theoretical Analysis of Self-Supervised Learning for Vision Transformers," ICLR 2025.
> [R3] HaoChen et al., "Provable Guarantees for Self-Supervised Contrastive Learning," NeurIPS 2021.
> [R4] Wen and Li, "Toward Understanding the Feature Learning Process of Self-Supervised Contrastive Learning," ICML 2021.

---

> > ### Author Response · Authors · 2025-08-07
> >
> > Thank you again for your original positive rating. We understand you may have a busy schedule; however, as the rebuttal phase deadline approaches, we wanted to check in to see if you had any remaining concerns or comments regarding our response. Please let us know if there’s anything you’d like us to clarify. We appreciate your time and consideration.

---

### Official Review · Reviewer_t22h · 2025-07-02

**Clarity:** 3
**Significance:** 3
**Originality:** 3
**Rating:** 5
**Confidence:** 3

**Summary:**

This paper develops a theoretical framework for understanding contrastive learning on noisy, misaligned image–text data and proves how “recaptioning” fixes it. Specifically, it improves zero-shot performance in vision–language models by targeting the data misalignment where texts can include spurious or omit true visual features. Under a sparse-coding data model, it analyzes SGD dynamics for two one-hidden-layer ReLU encoders trained with a spectral contrastive loss, showing that spurious text features entangle representations and degrade generalization. It then proves that a simple ReLU decoder trained on a small high-quality set can generate captions that suppress spurious correlations, yielding purified features and probably better out-of-domain zero-shot accuracy—results are validated by experiments on CLIP and LaCLIP.

**Questions:**

Q1: How do contrastively pre-trained VLMs align modalities, extract feature representations, and achieve zero-shot capabilities? How does text recaptioning on noisy image-text pairs provably enhance generalization performance?


Q2: How well do the theoretical insights extend to deeper or transformer-based encoder architectures commonly used in VLMs, and what modifications (if any) would be needed in your dynamics or misalignment models to capture their behavior?



Q3: Definition 3.1 uses a ReLU with form σ(⟨w,x⟩−b)−σ(−⟨w,x⟩−b), which differs from standard  ReLU networks. What is the technical necessity of this symmetric activation choice, and can your SGD-dynamics analysis extend to vanilla max⁡(0,⟨w,x⟩)?


Q4: Assumption 3.2 models both images and texts via the same sparse coding with orthonormal dictionaries M and H. In practice, text representations (e.g.\ token embeddings, LLM hidden states) live in very different spaces than dense image features. Could correlated or overcomplete text dictionaries qualitatively break the convergence or purity guarantees?

**Ethical Concerns:**

["NO or VERY MINOR ethics concerns only"]

**Final Justification:**

This paper is recommended to be accepted.

**Limitations:**

Yes

**Quality:**

3

**Strengths And Weaknesses:**

This paper demonstrates several strengths across the dimensions of originality, quality, clarity, and significance:




Quality:


This paper gives rigorous theoretical proof of the training dynamics and generalization analysis of nonlinear VLMs, showing the impact of misaligned image-text pairs on pre-training performance. Moreover, they give theoretical justification of enhanced out-of-domain generalization through pre-training with text recaptioning.



Clarity:


This paper gives clear proof and illustrations to set up their theory and training framework. The proof and assumptions are explained clearly.




Significance:




This paper is significant because it rigorously links feature purity—the degree to which each neuron encodes a single semantic direction—to downstream generalization in contrastive models. Theorems 4.1–4.2 establish that, although SGD reliably optimizes the nonconvex contrastive objective, training on raw, misaligned pairs inevitably entangles true and spurious features. Theorems 4.3–4.4 then show that recaptioning and filtering purify these representations so each neuron encodes only a single feature, and Theorem 4.5 guarantees that this purification yields vanishing failure rates in zero-shot, out-of-domain classification.





Originality:
The originality is good in this paper.

---

> ### Author Rebuttal · Authors · 2025-07-30
>
> We thank the reviewer for the valuable time in the evaluation.
>
> ## Response to Question 1: How do contrastively pre-trained VLMs align modalities, extract feature representations, and achieve zero-shot capabilities? How does text recaptioning on noisy image-text pairs provably enhance generalization performance?
>
> During contrastive pretraining, stochastic gradient descent (SGD) initially causes neurons to align with both true and spuriously correlated features, due to comparable gradient magnitudes (Appendix D). As training progresses, this spurious alignment is reinforced—neurons entangled with mixed features amplify both components simultaneously (Appendix E). Without intervention, the final representation encodes multiple features per neuron, preventing clean separation (Appendix F, Theorem 4.2).
>
> To improve the pre-trained model, a decoder is fine-tuned on clean data to generate synthetic captions. Theorem 4.3 (Appendix G) proves that neurons aligned with true features enhance their alignment exponentially, while suppressing spurious activations to negligible levels. Appendix H further shows that replacing original captions with synthetic ones filters out spurious correlations. Consequently, purified alignment is restored, and Theorem 4.5 (Appendix J) establishes that the zero-shot error drops to $o(1)$, demonstrating improved downstream generalization. A more detailed description can be found in Appendix B.1 Proof Scratch.
>
> ## Response to Question 2: How well do the theoretical insights extend to deeper or transformer-based encoders commonly used in VLMs?
>
> Extending our analysis to deep or transformer-based encoders would require major technical advances.
>
> **Attention and softmax complexity**
> Transformers rely on multi-head attention, where each layer multiplies token similarities by a softmax weight matrix. This induces dense, data-dependent coupling across all feature dimensions, far beyond the local interactions in a one-hidden-layer ReLU network. Formal convergence proofs must track the evolution of these weights and the resulting non-linear dynamics, a problem still open even in supervised settings.
>
> **Tokenized input and pairwise structure**
> Vision–language transformers operate on sequences of image patches or subword tokens. A rigorous treatment would need a mathematical model for tokenization and a similarity metric for positive pairs that captures spatial or linguistic structure. Such formulations are not yet available in a form amenable to SGD dynamics analysis.
>
> **Current theoretical frontier**
> Existing theory for transformers is limited to supervised tasks or linear approximations: [R1, R2] studies model editing in nonlinear transformers under supervised objectives. For contrastive learning, the most expressive theoretical results remain at the one-hidden-layer level, as in [R3].
>
> Thus our two-encoder, one-hidden-layer nonlinear contrastive model represents the state of the art for provable training dynamics in VLMs. Generalising to full transformers is an important direction for future work.
>
> **Empirical verification on large transformer VLMs**
> Our theoretical insights, derived from a one-hidden-layer model, are borne out by experiments on deep ViT-based systems:
>
> 1. **Purified neurons enhance generalisation.** LaCLIP, which incorporates the recaption–filter stage predicted to raise feature purity, attains higher accuracy and a larger silhouette score than its CLIP counterpart (Table 2).
>
> 2. **Data misalignment reduces feature purity.** When we shuffle captions to increase misalignment, the average neuron cosine similarity grows in proportion to the shuffle rate, matching Theorem 4.2.
>
> These observations show that conclusions proved for a simplified network transfer qualitatively to state-of-the-art transformer VLMs such as CLIP and LaCLIP.
>
> ## Response to Question 3: Motivation for the Symmetric ReLU Activation in Definition 3.1
>
> We adopt the symmetric ReLU $f_i(x)=\sigma(\langle w_i,x\rangle-b_i)-\sigma(-\langle w_i,x\rangle-b_i)$. This form lets us take $b_i>0$ without loss of generality and greatly simplifies analysis. The positive bias makes a neuron activate when $\langle w_i,x\rangle>b_i$ and also when $\langle w_i,x\rangle<-b_i$, yielding $f_i(x)=-f_i(-x)$. With this symmetry a single indicator function covers both signs of the inner product, streamlining gradient–alignment calculations while preserving the expressive power of the standard ReLU. All results carry over to the usual ReLU $\max(0,\langle w_i,x\rangle)$ after a linear rescaling of weights and biases.
>
> The same symmetric activation is common in recent theory [R4, R5].
>
> ## Response to Question 4: Could correlated or overcomplete text dictionaries break the guarantees?
>
> **Practical incoherence suffices**
> This is an interesting question. Since orthogonal or incoherent bases are standard assumptions in learning theory, we follow existing works in adopting them for our analysis. Exact orthogonality is not required; our theoretical guarantees remain valid even if the text dictionary is somewhat correlated, as long as distinct features are sufficiently incoherent—meaning their pairwise inner products are upper bounded by a smaller value. This bounded incoherence limits cross-feature interference, allowing our convergence and feature-purity results to hold. The weaker incoherence yields larger residual terms $r_i$ in Eq.(12) and requires more iterations to achieve convergence.
>
> On the other hand, if the dictionary is highly coherent, it can lead to non-unique representations of the data. This makes it difficult to clearly define positive pairs and learned features, posing challenges that fall outside the scope of our current analysis. We leave a deeper exploration of this case for future work.
>
> **Extension to overcomplete bases**
> An overcomplete dictionary can be reduced to a complete basis by a suitable linear transformation. The latent code $z_x$ therefore need not be limited to the discrete set $\{0,\pm1\}$; any real-valued sparse representation suffices. Allowing arbitrary magnitudes in $z_x$ changes only the scale of the coefficients $\alpha_{i,j}$ and $\alpha_{i,j'}$ in Eq.(12), but the qualitative conclusions regarding feature mixing and eventual purification are unchanged.
>
> **Standard theoretical assumption**
> Orthogonal or incoherent bases are standard in neural-network theory, such as transferability analysis in CLIP embeddings [R6] and sparse-modelled contrastive learning dynamics [R3]. Sparse coding is in fact a standard modelling choice for both images and texts and is widely used in recent neural-network theory:
>
> 1. **Image evidence**
> [R7] shows that Caltech101 images admit sparse codes over learned dictionaries and that sparse coding outperforms vector quantisation.
> [R8] demonstrates that face images can be reconstructed by a small set of dictionary atoms and that sparsity improves recognition accuracy.
>
> 2. **Text evidence**
> [R9] models word co-occurrence statistics with sparse latent factors, providing a theoretical basis for skip-gram embeddings.
>
> 3. **Sparse coding in neural-network theory**
> [R6] analyses sparse text features in multimodal models such as CLIP.
> [R4] proves adversarial robustness under a sparse latent image model.
> [R3] tracks how contrastive learning separates true and spurious features using a sparse data model.
>
> These studies confirm that assuming sparse codes for both modalities is a common and practical basis for rigorous analysis.
>
> ## References
> [R1] Li et al., "When Is Task Vector Provably Effective for Model Editing? A Generalization Analysis of Nonlinear Transformers," NeurIPS 2023.
> [R2] Huang, Yu, et al. "A Theoretical Analysis of Self-Supervised Learning for Vision Transformers," ICLR 2025.
> [R3] Wen and Li, "Toward Understanding the Feature Learning Process of Self-Supervised Contrastive Learning," ICML 2021.
> [R4] Allen-Zhu and Li, "Feature Purification: How Adversarial Training Performs Robust Deep Learning," FOCS 2022.
> [R5] HaoChen et al., "Provable Guarantees for Self-Supervised Contrastive Learning," NeurIPS 2021.
> [R6] Chen et al., "Understanding Transferable Representation Learning and Zero-Shot Transfer in CLIP," ICLR 2023.
> [R7] Yang et al., "Linear Spatial Pyramid Matching Using Sparse Coding for Image Classification," CVPR 2009.
> [R8] Yang et al., "Robust Sparse Coding for Face Recognition," CVPR 2011.
> [R9] Arora et al., "A Latent Variable Model Approach to PMI-based Word Embeddings," TACL 2016.

---

> > ### Comment · Reviewer_t22h · 2025-08-03
> >
> > I thank the authors for the rebuttal. The authors have addressed my concerns.

---

> > > ### Author Response · Authors · 2025-08-07
> > >
> > > We sincerely thank you for your time and effort in reviewing our rebuttal and are pleased that all of your concerns have been addressed. Should any further questions arise during the subsequent discussion stage, please do not hesitate to let us know.

---

### Official Review · Reviewer_XKgJ · 2025-07-03

**Clarity:** 4
**Significance:** 3
**Originality:** 3
**Rating:** 5
**Confidence:** 3

**Summary:**

The paper proposes a theoretical framework to analyze the impact of data misalignment on multimodal contrastive learning models (e.g., CLIP). This framework theoretically characterizes the effects of spurious correlations, recaptioning, filtering, etc., on model performance. The experimental results from simulated experiments and experiments on Food-101, CIFAR-100, and Caltech-101 support the framework.

**Questions:**

1. In Equation 2, why is $f_i(x)$ defined in the form presented in the paper rather than a simpler form like $f_i(x)=\sigma(<w_i,x>+b_i)$?
2. Could you supplement the results of Table 2 on ImageNet?

**Ethical Concerns:**

["NO or VERY MINOR ethics concerns only"]

**Quality:**

4

**Strengths And Weaknesses:**

strength:
- The issue analyzed in this paper is of great significance.
- The theoretical analysis of this paper is solid.
- Real-world experiments have demonstrated the effectiveness of their framework.

weakness:

1. In the theoretical analysis, the model adopted in the paper is somewhat simplified (a one-hidden-layer ReLU network).
2. The assumptions introduced in the paper are overly strong. For instance, Assumption 3.2 assumes that image-text pairs are under sparse coding.

---

> ### Author Rebuttal · Authors · 2025-07-30
>
> We thank the reviewer for the valuable time in the evaluation.
>
> ## Response to Weakness 1: Model simplicity
> While the model studied in this paper is a simplified version of real-world applications, we believe that analyzing the training dynamics of a one-hidden-layer network with nonlinear activation represents the most advanced setting currently allowing a comprehensive theoretical analysis of contrastive learning. Table 1 in the paper shows that our two-encoder architecture achieves the highest modelling fidelity among existing theoretical works.
>
> **Contrastive-learning theory**
> [R1] analyses only linear encoders.
> [R2] extends to one hidden layer but still studies a single encoder for one modality.
> Our work is the first to treat two nonlinear encoders, misaligned image–text multi-modal data, and zero-shot evaluation.
>
> **Supervised and adversarial theory**
> [R3] establishes feature purification with one hidden layer under adversarial training.
> [R4] studies nonlinear transformers, but only for supervised model editing, and did not consider contrastive or multimodal objectives.
>
> **Semi-supervised pseudo-labeling**
> [R5] and [R6] analyse self-training with one-hot class pseudo labels. Generating a single class label is far simpler than producing a full synthetic caption, and pseudo-label methods involve a single encoder with explicit labels. By contrast, our setting requires two nonlinear encoders, high-dimensional text generation, and a non-convex contrastive loss, making the theoretical analysis substantially more challenging.
>
> ## Response to Weakness 2: The sparse-coding assumption may be overly strong
> Thanks for the observation. Sparse coding is in fact a standard modelling choice for both images and texts and is widely used in recent neural-network theory, rather than a special assumption of our work.
>
> **Image evidence**
> [R7] shows that Caltech101 images admit sparse codes over learned dictionaries and that sparse coding outperforms vector quantisation.
> [R11] demonstrates that face images can be reconstructed by a small set of dictionary atoms and that sparsity improves recognition accuracy.
>
> **Text evidence**
> [R8] models word co-occurrence statistics with sparse latent factors, providing a theoretical basis for skip-gram embeddings.
>
> **Sparse coding in neural-network theory**
> [R9] analyses sparse text features in multimodal models such as CLIP.
> [R10] proves adversarial robustness under a sparse latent image model.
> [R2] tracks how contrastive learning separates true and spurious features using a sparse data model.
>
> These studies confirm that assuming sparse codes for both modalities is a common and practical basis for rigorous analysis.
>
> ## Response to Question 1: Why is $f_i(x)$ defined as in Equation (2)?
> We adopt the symmetric ReLU $f_i(x)=\sigma(\langle w_i,x\rangle-b_i)-\sigma(-\langle w_i,x\rangle-b_i)$. This form lets us take $b_i>0$ without loss of generality and greatly simplifies analysis. The positive bias makes a neuron activate when $\langle w_i,x\rangle>b_i$ and also when $\langle w_i,x\rangle<-b_i$, yielding $f_i(x)=-f_i(-x)$. With this symmetry a single indicator function covers both signs of the inner product, streamlining gradient–alignment calculations while preserving the expressive power of the standard ReLU. All results carry over to the usual ReLU $\max(0,\langle w_i,x\rangle)$ after a linear rescaling of weights and biases.
>
> The same symmetric activation is common in recent theory [R3, R2].
>
> ## Response to Question 2: Supplementary ImageNet Results
>
> |Model|Top-1 (%)|Top-5 (%)|Silhouette|
> |---|---|---|---|
> |CC12M CLIP|35.04|62.10|-0.014639|
> |CC12M LaCLIP|**42.62**|**70.17**|**-0.008141**|
> |LAION-400M CLIP|58.34|84.73|-0.029893|
> |LAION-400M LaCLIP|**62.27**|**86.34**|**-0.056593**|
> |RedCaps CLIP|37.66|63.31|-0.022045|
> |RedCaps LaCLIP|**39.66**|**66.06**|**-0.012269**|
>
> The LaCLIP variants consistently surpass their CLIP counterparts on both Top-1 and Top-5 accuracy. Higher silhouette scores further indicate cleaner feature separation after recaptioning, in line with our theoretical predictions.
>
> ## References
> [R1] HaoChen et al., "Provable Guarantees for Self-Supervised Contrastive Learning," NeurIPS 2021.
> [R2] Wen and Li, "Toward Understanding the Feature Learning Process of Self-Supervised Contrastive Learning," ICML 2021.
> [R3] Allen-Zhu and Li, "Feature Purification: How Adversarial Training Performs Robust Deep Learning," FOCS 2022.
> [R4] Li et al., "When Is Task Vector Provably Effective for Model Editing? A Generalization Analysis of Nonlinear Transformers," NeurIPS 2023.
> [R5] D.-H. Lee, "Pseudo-Label: The Simple and Efficient Semi-Supervised Learning Method for Deep Neural Networks," 2013.
> [R6] Zhang, Shuai, et al., "How Does Unlabeled Data Improve Generalization in Self-Training? A One-Hidden-Layer Theoretical Analysis," ICLR 2022.
> [R7] Yang et al., "Linear Spatial Pyramid Matching Using Sparse Coding for Image Classification," CVPR 2009.
> [R8] Arora et al., "A Latent Variable Model Approach to PMI-based Word Embeddings," TACL 2016.
> [R9] Chen et al., "Understanding Transferable Representation Learning and Zero-Shot Transfer in CLIP," ICLR 2023.
> [R10] Allen-Zhu and Li, "Feature Purification: How Adversarial Training Performs Robust Deep Learning," FOCS 2022.
> [R11] Yang et al., "Robust Sparse Coding for Face Recognition," CVPR 2010.

---

> > ### Comment · Reviewer_XKgJ · 2025-08-03
> >
> > I thank the authors for the rebuttal. The authors have basically addressed my concerns.

---

> > > ### Author Response · Authors · 2025-08-07
> > >
> > > We sincerely thank you for your time and effort in reviewing our rebuttal and are pleased that the major concerns have been addressed. Should any questions requiring further clarification arise, please do not hesitate to let us know.

---

### Decision · Program_Chairs · 2025-09-17

**Decision:**

Accept (poster)

**Comment:**

This paper develops a theoretical framework for understanding contrastive learning on noisy, misaligned image–text data, and shows that misalignment leads to mixed representations and reduced generalization, while improving alignment through recaptioning and filtering enhances feature purity and model performance. Although the paper considers strong assumptions and simplified settings (two-layer networks), all reviewers agree that this is a good paper. Therefore, I recommend accepting the paper.